# The Cosmological CPT Theorem

Harry Goodhew[1,a,b], Ayngaran Thavanesan[2,a], and Aron C. Wall[3,a,c]

a *Department of Applied Mathematics and Theoretical Physics, University of Cambridge,*
*Wilberforce Road, Cambridge, CB3 0WA, UK.*
b *Leung Center for Cosmology and Particle Astrophysics, National Taiwan University,*
*Taipei 10617, Taiwan.*
c *School of Natural Sciences, Institute for Advanced Study, Princeton, NJ 08540 USA.*

## Abstract

The CPT theorem states that a unitary and Lorentz-invariant theory must also be invariant under a discrete symmetry **CRT** which reverses charge, time, and one spatial direction. In this article, we study a $\mathbb{Z}_2 \times \mathbb{Z}_2$ symmetry group, in which two of the nontrivial symmetries ("Reflection Reality" and a 180 degree rotation) are implied by Unitarity and Lorentz Invariance respectively, while the third is **CRT**. (In cosmology, Scale Invariance plays the role of Lorentz Invariance.) This naturally leads to converses of the CPT theorem, as any two of the discrete $\mathbb{Z}_2$ symmetries will imply the third one. Furthermore, in many field theories, the Reflection Reality $\mathbb{Z}_2$ symmetry is actually sufficient to imply the theory is fully unitary, over a generic range of couplings. Building upon previous work on the Cosmological Optical Theorem, we derive non-perturbative reality conditions associated with bulk Reflection Reality (in all flat FLRW models) and **CRT** (in de Sitter spacetime), in arbitrary dimensions. Remarkably, this **CRT** constraint suffices to fix the phase of all wavefunction coefficients at future infinity (up to a real sign) — without requiring any analytic continuation, or comparison to past infinity — although extra care is required in cases where the bulk theory has logarithmic UV or IR divergences. This result has significant implications for de Sitter holography, as it allows us to determine the phases of arbitrary $n$-point functions in the dual CFT.

[1] goodhewhf@ntu.edu.tw
[2] at735@cantab.ac.uk
[3] aroncwall@gmail.com

# 1 Introduction

In quantum field theory, symmetries and other physical principles impose fundamental constraints on dynamical systems. This philosophy of "bootstrapping" has proven to be powerful in removing the need for repetitive computations, since typical Lagrangian methods lead to a huge amount of redundancy when going from theory to observables. In particle physics, the role of fundamental principles is hard to observe when calculating scattering amplitudes from Lagrangians in perturbation theory with many redundancies, for example different Lagrangians can be related to each other by gauge transformations or field redefinitions. As a result, the on-shell method was developed to describe all physically consistent $S$-matrices [1–6]. Recently, this approach has been used to tackle the difficult task of computing late-time correlators which live at the boundary at the end of inflation through the program known as the Cosmological Bootstrap (see e.g. [7–20] as well as a recent Snowmass white paper [21]). The goal of the Cosmological Bootstrap program is to circumvent the need for standard perturbation theory techniques to compute cosmological correlators, which, even at tree-level, require nested time integrals to be performed, and instead use symmetries and physical principles to directly infer the form and properties of the cosmological correlators.

However, this approach raises a new question: what are the symmetries that underpin the physical processes within the universe as well as our universe as a whole? One of the most universal and important symmetries in flat spacetime is **CRT** (the simultaneous reversal of time, **T**, reflection across a plane, **R**, and charge conjugation, **C**). This symmetry ensures, for example, that particles and antiparticles have the same mass. Famously, the CPT theorem ensures that this symmetry is always valid in any QFT which is Lorentz invariant and unitary.

What are the implications of this symmetry for our universe as a whole? At first sight one might think that **CRT** is simply no longer a useful symmetry in cosmology (at least over distances longer than the Hubble scale), because the arrow of expansion breaks time reversal symmetry. Applying **CRT** would relate an expanding universe to a (hypothetical) contracting universe, and would therefore (seemingly) not place any interesting constraints on the expanding cosmology alone. On the other hand, in global de Sitter, which has a both a contracting phase and an expanding phase, **CRT** symmetry maps operators at future infinity ($\mathcal{I}^+$) to operators past infinity ($\mathcal{I}^-$). But this formulation of **CRT** also does not appear to be very useful, if we only have access to observables at $\mathcal{I}^+$, as is typical for applications to inflationary cosmology, or dS/CFT.[1] If this were the only implication of **CRT**, it would not be of much use in cosmology.

Suppose we have a spacetime which can be well approximated by the Bunch-Davies vacuum state in de Sitter spacetime (e.g. during slow roll inflation). In this setup, we will show that by doing a suitable analytic continuation in the conformal time parameter $\eta$, we can access the past dS-Poincaré patch, and hence place nontrivial constraints on correlators and wavefunction coefficients in the future Poincaré patch. At future infinity ($\eta = 0$) the constraints on the wavefunction coefficient become particularly simple, as the $\eta$ continuation becomes trivial for IR finite correlators. In this case, we will be able to determine the phase of wavefunction coefficients without any analytic continuation.

Similarly, in any FLRW cosmology with spatially flat time slices, we will show that one can also continue to a contracting cosmology by continuing the spatial coordinates **y** (or equivalently the momenta **k**). This constraint on the wavefunction coefficients, which comes from Unitarity rather than **CRT**, gives a nonperturbative form of the Cosmological Optical Theorem [13–15], which agrees with the existing tree level results. (Here, we are referring to the hermitian analyticity reality conditions that apply to single diagrams, not to the "cutting rules" which relate different diagrams.)

The reason we can obtain these stronger constraints, is that we are using the additional conditions

---

[1]In the usual formulations of dS/CFT (see e.g. [22–24]), the holographic field theory lives on a single conformal boundary, and is dual to the Hartle-Hawking saddle extended to $\mathcal{I}^+$ only. One could alternatively extend the saddle to $\mathcal{I}^-$, but this would be described by a distinct, **CRT**-reversed dual field theory.

that (i) we are considering the vacuum state only, and (ii) the dynamics of the theory come from a local Lagrangian that has CRT symmetry around *every* point. This combination of discrete symmetries with locality turns out to give results much more powerful than those which can be obtained by simply thinking of discrete symmetries as acting globally on correlators, without considering the etiology of the correlators, i.e. the manner in which they arise from interactions, starting from a Bunch-Davies boundary condition.

## 1.1 The CPT Theorem and its Converses

In this paper we also hope to clarify the structure of the CPT theorem in flat spacetime. The usual CPT theorem proves the following implication:

$$\text{Lorentz Invariance} + \text{Unitarity} \implies \textbf{CRT} \text{ invariance .}$$

In this paper we derive a new formulation of the CPT theorem for integer spin fields in which we consider two discrete transformations, Reflection Reality ($\textbf{RR}$) and a 180° rotation ($\textbf{R}_\pi$), that are implied by Unitarity and Lorentz Invariance respectively. These form a $\mathbb{Z}_2 \times \mathbb{Z}_2$ symmetry group with $\textbf{CRT}$ and the identity.[2] This allows us to weaken the premises of the CPT theorem to:

$$\text{180° rotation} + \text{Reflection Reality} \implies \textbf{CRT} \text{ invariance .}$$

The existence of a 180° rotation follows from an analytic continuation of the existence of a single $SO^+(1,1)$ Lorentz Invariance symmetry, which is itself a subgroup of the full Lorentz group $SO^+(d,1)$. On the other hand, Reflection Reality ($\textbf{RR}$) is a discrete symmetry which holds for all theories with *real* (but not necessarily positive) norm on states, and hence *all* unitary theories have to satisfy non-perturbatively.[3]

This symmetry group structure enables us to make converse statements to the usual CPT theorem, as any two of the transformations implies the third. So we also have:

$$\textbf{CRT} \text{ invariance} + \text{180° rotation} \implies \text{Reflection Reality ;}$$

$$\textbf{CRT} \text{ invariance} + \text{Reflection Reality} \implies \text{180° rotation .}$$

It is important to note that a reflection real theory is not necessarily unitary, e.g. a theory with negative norm states or spin-statistics violating fields. However, this situation typically arises only in cases where one makes a bad discrete choice as to the matter content of a theory. In many cases (e.g. where you start with a unitary theory and continuously deform the coupling constants in a way that doesn't change the number of degrees of freedom), $\textbf{RR}$ is enough to accidentally imply Unitarity. We will argue that this happens for a generic (codimension 0) set of couplings in Section 5.1.

While this fact, that $\textbf{RR}$ is often enough to imply Unitarity, is not commonly discussed, it plays a structural role in many theoretical physics constructions. For example, it explains why field theorists working in Euclidean signature are very careful to ensure their theories are covariant, but don't spend much time worrying about Unitarity — this is because the discrete symmetries in a covariant Euclidean signature theory are usually sufficient to automatically ensure Unitarity upon continuation to Minkowski

---

[2]While in this paper we only treat integer spin carefully, if we allow spinors a 360° rotation introduces a factor of $(-1)^{2s}$, requiring that we pass to a double cover: the dihedral group $D_4$. See the discussion at the end of Section 2.2.

[3]It is easy to find theories which are not invariant under $SO^+(1,1)$, but do satisfy the discrete 180° rotation implied by the boost and hence are $\textbf{CRT}$-invariant provided that they satisfy $\textbf{RR}$. For example, consider the interaction $\tilde{\lambda}(\dot{\phi})^4$, where $\tilde{\lambda}$ is a real coupling, $\phi$ is a real scalar field and dots denote derivatives with respect to time. This is clearly not invariant under the $SO^+(1,1)$ boost (since only covariant derivatives preserve boosts) but is both Unitary (as it has a real coupling) and $\textbf{CRT}$ invariant as it invariant under the discrete 180° rotation $\textbf{R}_\pi$ due to the even number of time derivatives.

signature. Another example of this phenomenon was studied in [25], where the covariant and canonical formulations of the Maxwell field were compared in curved spacetime. In this case, covariance suffices to imply Unitarity, but not vice versa. In our current language, this is because Unitarity accidentally follows from **CRT** together with a discrete subgroup of covariance.

## 1.2 Applications to Cosmology

Furthermore, we identify an isomorphic $\mathbb{Z}_2 \times \mathbb{Z}_2$ group structure in de Sitter, relating discrete symmetries arising from Scale Invariance, Unitarity and **CRT** in global de Sitter.

Using the global structure of de Sitter as a guide, we are able to also identify the set of discrete symmetries pertaining to the Poincaré patch of de Sitter through analytic continuations and use this to derive an analogous theorem in cosmology. In global de Sitter, the analogue of a Lorentz boost is an $SO^+(1,1)$ subgroup of the full de Sitter group $SO^+(d+1,1)$. Any such $SO^+(1,1)$ boost defines a future Poincaré patch, in which it acts as a scaling symmetry $\mathbf{D}_\lambda$ on the conformal coordinates: $(\eta, \mathbf{y}) \to (\lambda\eta, \lambda\mathbf{y})$, for $\lambda > 0$.[4] The 180° rotation thus becomes an analytic continuation of the scaling symmetry that sends $(\eta, \mathbf{y}) \to (-\eta, -\mathbf{y})$, which we call Discrete Scale Invariance ($\mathbf{D}_{-1}$). This is particularly interesting in the context of inflation where we often break de Sitter boosts (these act as special conformal transformations at the boundary) and so we lose the full set of de Sitter isometries. Here we need only to invoke an analytic continuation of Dilatations. This symmetry should therefore be valid in realistic models of slow-roll inflation, up to terms proportional to the tilt $n_s - 1$.

The Cosmological CPT theorem now says that:

$$\text{Scale Invariance} + \text{Unitarity} \implies \textbf{CRT} \text{ invariance.}$$

while the weakened form and converse results are:

$$\text{Discrete Scale Invariance} + \text{Reflection Reality} \implies \textbf{CRT} \text{ invariance;}$$

$$\textbf{CRT} \text{ invariance} + \text{Discrete Scale Invariance} \implies \text{Reflection Reality;}$$

$$\textbf{CRT} \text{ invariance} + \text{Reflection Reality} \implies \text{Discrete Scale Invariance.}$$

We remind the reader that **RR** is a discrete symmetry which all unitary theories have to satisfy non-perturbatively, as well as to all orders in perturbation theory. This will become important when we discuss the constraints of these discrete symmetries on cosmological observables.[5] Besides the spacetime manifold itself, we will study the action of these symmetries on two basic types of objects: cosmological correlators in the in-in formalism, and wavefunction coefficients.

The first structure are cosmological $n$-pt correlators $B(\eta_1 \ldots \eta_n; \mathbf{y}_1 \ldots \mathbf{y}_n)$. Here, we find in section 3.4 the following symmetry transformations for equal time correlators (the cosmological correlators are all on an equal time at the end of inflation) in the Poincaré patch:

$$\textbf{CRT}: \quad B_n(\eta; \mathbf{y}) = B_n^*(-\eta; -\mathbf{y}); \tag{1.1}$$

$$\mathbf{D}_{-1}^{\pm}: \quad B_n(\eta; \mathbf{y}) = B_n(-\eta; -\mathbf{y}); \tag{1.2}$$

$$\textbf{RR}: \quad B_n(\eta; \mathbf{y}) = B_n^*(\eta; \mathbf{y}), \tag{1.3}$$

---

[4]Really this is a Killing symmetry which takes the form of a translation of cosmic time together with a rescaling of space, i.e. $\mathbf{D} = x^\mu \partial_\mu \neq x^i \partial_i$ which differs from the usual CFT literature. But it looks like a scaling of the future conformal boundary $\mathcal{I}^+$, so cosmologists call it "Scale Invariance". Hence we will use the term "Scale Invariance" interchangeably with dilations/dilatations at various points to connect with the terminology used in the cosmology literature.

[5]We plan a follow-up paper to discuss how the discrete symmetries manifest themselves as well as their general constraints in the static patch of de Sitter [26].

e.g. **RR** implies the simple constraint that equal-time correlators are real. However, we will find that we can derive a more powerful constraint when working with the wavefunction coefficients considered in the cosmology literature.

**Non-perturbative constraints on wavefunction coefficients.** Secondly, we also study the application of these symmetries to the coefficients of the Wavefunction of the Universe (WFU) $\Psi$ (see e.g. [27–36]). Note that the WFU $\Psi$ we work with here is the QFT in curved spacetime limit of the Wheeler-DeWitt states considered in the canonical quantum gravity literature (see e.g. [24, 37–53]). By Taylor expanding $\Psi$, one obtains the wavefunction coefficients $\psi_n$, which have proved useful in understanding Unitarity in the inflationary context through the Cosmological Optical Theorem [13–15], although until now these constraints have only been able to be expressed in a perturbative way. However, as a result of our approach through the Cosmological CPT theorem which has no reliance on perturbation theory, we are, for the first time, able to make non-perturbative statements of bulk Unitarity.

It is important to realise that we *cannot* simply directly extend the equations (1.1)–(1.3) from $B_n$ to $\psi_n$, as this method fails to produce interesting constraints on the phase of the wavefunction in the future Poincaré patch. (The problem is that $\psi_n$ is generated by a single sheet of the path integral, instead of two sheets like $B_n$.) Instead, we have to re-analyse the problem by directly considering the effect of the symmetries on the local Lagrangian, as stated above. This also identifies the reason for why the authors of [54] found some examples of *non-local* interactions which did not satisfy (1.9) at tree-level.

(In our analytic continuation of possible wavefunction coefficients $\psi_n$, we find that we can no longer assume *a priori* that a 360° complex rotation of the scaling acts trivially on $\psi_n$. As often happens with complex functions, there is a possible monodromy when going around the singularity at the origin. Thus, we must distinguish between the two signs of the 180° rotation, which we call $\mathbf{D}_{-1}^{\pm}$. This extends the $\mathbb{Z}_2 \times \mathbb{Z}_2$ to a larger group isomorphic to the automorphisms of the integers: $\mathrm{Aut}(\mathbb{Z})$.[6])

In section 6.1 we find that the local Lagrangian symmetries enforce the following relationships between bulk wavefunction coefficients, $\psi_n$:

$$\mathbf{CRT}: \quad [\psi_n(\eta; \mathbf{y}; \Omega)]^* \;=\; \psi_n(e^{-i\pi}\eta; -\mathbf{y}; e^{-i\pi}\Omega)\,; \tag{1.4}$$

$$\mathbf{D}_{-1}^{\pm}: \quad \psi_n(\eta; \mathbf{y}; \Omega) \;=\; \psi_n(e^{\pm i\pi}\eta; e^{\pm i\pi}\mathbf{y}; \Omega)\,; \tag{1.5}$$

$$\mathbf{RR}: \quad [\psi_n(\eta; \mathbf{y}; \Omega)]^* \;=\; \psi_n(\eta; -e^{i\pi}\mathbf{y}; e^{-i\pi}\Omega)\,. \tag{1.6}$$

In the above expressions, the factors of $e^{\pm i\pi}$ indicate the direction of analytic continuation in the complex plane. $\Omega$ is the Weyl factor of the bulk metric, which ends up being rotated by the non-perturbative analytic continuations of $\eta$ or $\mathbf{y}$.

**Perturbative constraints.** The above results can be expanded to arbitrary order in perturbation theory. Importantly, the rotation of the Weyl factor $\Omega$ leads to a dependence on the number of loops $L$ in the Feynman-Witten diagram. However, the effects of this $\Omega$ rotation end up being *trivial* in $d = $ odd spatial dimensions, for all tree level diagrams in the bulk (and more generally, loop diagrams that are free from divergences).

In sections 6.2 and 7.2 we find the following perturbative constraints on the bare (UV-finite) wavefunc-

---

[6]Of course, any $\psi_n$ that is *in fact* scale-invariant is by definition invariant under $\mathbf{D}_\lambda$ for *all* complex values of $\lambda$, so *a posteriori* the two symmetries do agree since, precisely because they are symmetries, they do nothing to a physical valued $\psi_n$. But, we need to consider how they act on the space of all possible $\psi_n$'s to pick out which values are physically allowed.

tion coefficients, valid for general real $d$:

$$\mathbf{CRT}: \quad \left[\psi_n^{(L)}(\eta_0; \mathbf{k})\right]^* = e^{i\pi(d+1)(L-1)}\psi_n^{(L)}(e^{-i\pi}\eta_0; \mathbf{k}), \tag{1.7}$$

$$\mathbf{D}_{-1}^{\pm}: \quad \psi_n^{(L)}(\eta_0; \mathbf{k}) = e^{\pm i\pi d}\psi_n^{(L)}(e^{\pm i\pi}\eta_0; e^{\mp i\pi}\mathbf{k}), \tag{1.8}$$

$$\mathbf{RR}: \quad \left[\psi_n^{(L)}(\eta_0; \mathbf{k})\right]^* = e^{i\pi((d+1)L-1)}\psi_n^{(L)}(\eta_0; e^{-i\pi}\mathbf{k}). \tag{1.9}$$

Here we have taken the Fourier Transform to enable a more direct comparison to the Cosmology literature (our results are given in both position and momentum space in the body of the paper). The last symmetry, **RR**, holds for correlators in any flat FLRW spacetime and has been previously observed for both contact [13, 14] and exchange [54] tree-level diagrams. The corrections for UV divergent amplitudes are discussed in Sections 6.2 and 7.5.

**Phase formula.** One particularly powerful application of (1.7) is to push the wavefunction coefficients to the future boundary where, for light fields, the time dependence factorises straightforwardly. The resulting boundary wavefunction coefficient, $\overline{\psi}_n^{(L)}$, no longer depends on $\eta$, and so (in Section 7.4) we find it is possible to deduce the phase of an arbitrary coefficient:

$$\arg(\overline{\psi}_n^{(L)}) = -\frac{\pi}{2}\left((d+1)(L-1) + dn - \sum_\alpha \Delta_\alpha\right) + \pi\mathbb{N}, \tag{1.10}$$

where the last term indicates that **CRT** can only fix the phase up to an overall $\pm$ sign. $\Delta$ is the conformal dimension of the $n$ operators $\overline{\phi}_+$ used to differentiate the boundary wavefunction $\overline{\Psi}[\overline{\phi}_-]$, which is written in the basis of the conjugate operators $\overline{\phi}_-$ with dimension $d - \Delta$.[7] This result holds for any boundary wavefunction coefficients that UV and IR-finite (i.e. free from logarithmic divergences in $\eta$ or the UV cutoff $\epsilon$), as well as the coefficient in front of the leading $\log(\eta)$ divergence for any IR-divergent wavefunction coefficients, or the leading $\log(\epsilon)$ divergence for any UV-divergent diagram.[8]

If we restrict attention to the case where all particles have $\Delta = d$, as for the stress-tensor and massless scalar fields, we find the phase is independent of $n$:

$$\arg(\overline{\psi}_n^{(L)}) = -\frac{\pi}{2}(d+1)(L-1) + \pi\mathbb{N}. \tag{1.11}$$

This then tells us that massless scalar fields and gravitons have real boundary wavefunction coefficients to all loop order in even spacetime dimension provided they don't diverge in the UV. When UV divergences are present one finds in dimensional regularisation (dim-reg) a generic complex phase proportional to the dim-reg parameter $\delta$, i.e. $\arg(\overline{\psi}_n^{(L)}) \propto \pi\delta$. For tree-level, IR-finite and scale-invariant Feynman-Witten diagrams in cosmology, this was recently found independently [54] where the authors proved this reality condition by direct analysis of the bulk time integrals.[9] The implications of this for cosmology in terms of no-go theorems for parity violations has previously been explored for the specific case of the tree-level scalar trispectrum in [65, 66], and will be treated for more general tree-level and loop diagrams in [67]. This

---

[7]From a dS/CFT perspective, the objects with dimension $d-\Delta$ are the "sources" of the CFT partition function $Z_{\mathrm{CFT}} = \overline{\Psi}$, while the ones with dimension $\Delta$ are the "operators". But from a cosmology perspective, both are operators in the bulk Hilbert space, and we have to choose a basis. In cosmology, one normally chooses this basis by writing $\overline{\Psi}$ as a function of whichever field component falls off more slowly, which would imply that $\Delta \geq d/2$, but the phase formula above would still be valid if we make the opposite choice. We do not attempt in this paper to extend our results to heavy fields in the principal series, where $\Delta = d/2 + i\nu$. Doing so would require making the unpleasant decision to either sacrifice self-adjointness, or conformal invariance of the boundary conditions. See e.g. [55–64] for discussions regarding the principal series.

[8]There is also an extra correction for diagrams with spinor propagators, see footnote 39.

[9]AT would like to express his gratitude to David Stefanyszyn, Xi Tong and Yuhang Zhu for various extensive and helpful discussions on reality in cosmology.

is also consistent with what is found in flat-space amplitudes which can be shown to be real at tree-level, see e.g. [68][10]

## 1.3   Implications for dS/CFT

These constraints not only have implications for cosmology [67] but may also illuminate the role of Unitarity in de Sitter quantum gravity (dS/CFT), where previous attempts to identify the imprints of bulk unitary time evolution in the boundary theory had proven to be elusive (see e.g. [22, 23]). Given the close relationship between **CRT** and Unitarity, and the fact that **CRT** simplifies at the future boundary $\mathcal{I}^+$, our work gives some reason to think that bulk Unitarity corresponds to some equally natural restriction on the boundary CFT side. At the very least, if somebody claims to have an example of dS/CFT holography, the phase formula would give a simple way to rule out examples where the bulk theory violates Reflection Reality (**RR**), without any need to do calculations that probe the bulk interior. Then, to ensure that the bulk theory is fully unitary, the remaining task is to check by hand that that the bulk Lagrangian has no negative norm fields. We expect that success at this task would not require any additional fine-tuning of continuous parameters, although it might require making good discrete choices.[11]

A very simple check of the phase formula is to calculate the required number of degrees of freedom $c_{\mathrm{dS}}$ of the holographic CFT in de Sitter. There are a number of possible definitions of $c_{\mathrm{dS}}$, including:

1. various coefficients of the trace anomaly $\langle T^i_i \rangle$ in $\overline{\psi}^{(L)}_1$, when $d = $ even, or

2. the coefficient of the 2-point stress tensor correlator $\langle T_{ij}(\mathbf{k}) T_{lm}(-\mathbf{k}) \rangle$ in $\overline{\psi}^{(L)}_2$ for general $d$,

where $L$ is the loop order in the bulk theory, not the boundary CFT. These definitions coincide with the Virasoro central charge $c$ when $d = 2$, but differ in arbitrary $d$. Happily, the phase formula agrees for all these indicators,[12] so regardless of the definition of $c_{\mathrm{dS}}$ we find that

$$\arg(c_{\mathrm{dS}}) = \arg(\overline{\psi}^{(L)}_n) = -\frac{\pi}{2}(d+1)(L-1) + \pi\mathbb{N}\,. \tag{1.12}$$

From (1.12) we can infer that for $d = 2, 4, \ldots$, we expect the central charge to be imaginary at tree level (but real at odd-loop order), while for $d = 1, 3, 5 \ldots$ it is real at all orders (section 7.5). This matches what has been found in the literature:

- For $d = 2$ at tree-level, $c = i\frac{3\ell}{2G_N}$ (e.g. from applying Friedel's analysis to dS [41]); while for the case of pure gravity at 1-loop order we have $c = i\frac{3\ell}{2G_N} + 13$ [70];[13]

- For $d = 3$ at tree-level, $c_{\mathrm{dS}} \propto -\frac{\ell^2}{G_N}$ (e.g. in higher-spin dS/CFT [24] or Maldacena [27]);

- For $d = 4$ at tree level, $c_{\mathrm{dS}} \propto -i\frac{\ell^3}{G_N}$ (e.g. Maldacena [27]).

- For $d = 5$ we have $c_{\mathrm{dS}} \propto +\frac{\ell^4}{G_N}$, the same sign as a normal unitary CFT [71].

---

[10]AT thanks Carlos Duaso Pueyo for making him aware of this result in amplitudes.

[11]Of course, there are some other consistency conditions that dS/CFT should obey besides bulk Unitarity. This includes normalisability conditions on the wavefunction $\Psi$, which can be used to fix the sign of the real part of typical $\psi_2$ coefficients. A particularly strong constraint is the absence of bulk tachyons, which seems to require the absence of irrelevant single-trace operators $\Delta > d$ in the boundary field theory.

[12]The phase also agrees with the sphere free energy $F = \log Z$ in $d = $ odd (see [69] for a pedagogical review.

[13]We thank Jordan Cotler and Kristan Jensen for making us aware of their previous result for the 1-loop contribution to the central charge in $dS_3$ as well as upcoming work for loop calculations in $dS_2$ and $dS_4$. The more observant reader will notice that the real 1-loop contribution is positive in the convention chosen by the authors and the same sign as the tree-level contribution as predicted from the combination of the normalisability constraint and the phase formula.

Although our phase formula does not determine the real sign of $c_{\mathrm{dS}}$, this can be fixed by the stability requirement that $G_N > 0$, and one finds that the sign of the tree-level $c_{\mathrm{dS}}$ has a repeating pattern in $d$ mod 4 [71], as represented by the four cases above. This compatibility with previous results, as well as the results from more non-trivial explicit checks (forthcoming in [72] and summarised in Section 7.4 below) confirm the validity of the phase formula (1.10).

## 1.4 Plan of Paper

The outline of the paper is as follows:

We begin in Section 2 by reviewing the CPT theorem in flat space. We identify three discrete symmetries which form a $\mathbb{Z}_2 \times \mathbb{Z}_2$ group and are implied by the usual ingredients of the CPT theorem, namely Lorentz Invariance, Unitarity and **CRT**. We highlight the constraints these discrete symmetries impose on the observables and use the power of the group structure to provide a proof of the CPT theorem as well as converse statements. Then, in Section 3, we use the embedding formalism to extend these symmetries to de Sitter space, in both global and Poincaré coordinates. We describe how the $\mathbb{Z}_2 \times \mathbb{Z}_2$ group acts on cosmological correlators, with the 180° rotation taking the form of a Discrete Scale Invariance symmetry. We also define the wavefunction coefficients, leaving their symmetry transformations to later sections.

To lay the groundwork for further results, in Section 4, we discuss the important distinction between symmetries of the spacetime, and symmetries of the local Lagrangian, two concepts which do not quite overlap with each other. (Only the latter is useful for constraining wavefunction coefficients in a single Poincaré patch.) In Section 5 we show that, in many generic scenarios, the Reflection Reality symmetry **RR** suffices to imply that the theory is fully Unitary. (This greatly increases the importance of the converse CPT theorem that concludes to **RR**.)

In Section 6, we sketch how to implement discrete symmetries in cosmology, by analytically continuing either the $\eta$ coordinate (to implement the local Lagrangian form of **CRT**) or the **k** coordinate (to implement the local Lagrangian form of **RR**), with these two transformations related by a discrete scale transform (180° rotation) of the spacetime. This result can be expressed in both non-perturbative and perturbative forms. In Section 7 we provide a more concrete and detailed derivation of the perturbative conditions. By taking the limit as $\eta \to 0$, we derive the phase formula for the wavefunction coefficients at future infinity, and we check the formula by comparing to calculations of specific Feynman-Witten amplitudes in the literature. Finally, we conclude in Section 8 with a discussion of the implications of these results for dS/CFT and inflation, as well as some directions for possible future work.

## 2  CPT in Flat space

The CPT theorem in flat space states that a Unitary and Lorentz-invariant theory will also be **CRT** invariant. However, it is not possible to make converse statements. In this section we will present a new perspective on the proof of the CPT theorem, highlighting the group structure of a set of discrete symmetries that are consequences of **CRT**, Unitarity and Lorentz Invariance. This group structure necessitates the existence of converse statements and thus highlights which aspects of Unitarity or Lorentz Invariance are missed in these converse statements; however in Section 5 we will discuss how in many field theories, the full principle of Unitarity is *accidentally* implied up to some discrete choices.

## 2.1 Lorentz Invariance

A common confusion in the literature is interchangeable labelling of the proper orthochronous Lorentz group (also known as the restricted Lorentz group) $SO^+(d,1)$ and the proper Lorentz group $SO(D,1)$. The precise statement of Lorentz Invariance is invariance under the proper orthochronous Lorentz group $SO^+(d,1)$ which consists of those Lorentz transformations that preserve both the orientation of space and the direction of time. Table 1 provides a classifcation of all the subgroups of the Lorentz group which contain **at least one** of the four connected components $\{1, P, T, PT\}$. [14] The collection of the four interconnected parts can be organised into a group as the quotient group $O(d,1)/SO^+(d,1)$ which is also isomorphic to the $\mathbb{Z}_2 \times \mathbb{Z}_2$ group. Any Lorentz transformation can be defined through a correct, proper orthochronous Lorentz transformation combined with two additional pieces of data, which distinguish one of the four interconnected parts. This structure is characteristic of finite-dimensional Lie groups.

| Connected components | Name | Group label |
|---|---|---|
| 1 | Proper Orthochronous Lorentz group | $SO^+(d,1)$ |
| 1, PT | Proper Lorentz group | $SO(d,1)$ |
| 1, P | Orthochronous Lorentz group | $O^+(d,1)$ |
| 1, T | Orthochiral Lorentz group | $O^{\times}(d,1)$ |
| 1, P, T, PT | Lorentz group | $O(d,1)$ |

Table 1: Subgroups of the Lorentz group which contain **at least one** of the four connected components $\{1, P, T, PT\}$, where $d$ is the number of spatial dimensions. In applications to de Sitter, one should replace $d$ with $D = d + 1$.

## 2.2 Outline of CPT Theorem

The CPT theorem states that any QFT that is:

1. Unitary,

2. Lorentz invariant, and has

3. Energy bounded from below,

must also be invariant under the combined **CRT** symmetry. We will now dissect each assumption one by one.

Assumption 3 allows us to analytically continue to Euclidean signature, using the fact that $e^{-H\Delta\tau}$ is suppressed for positive Euclidean times $\Delta\tau > 0$. Let us adopt coordinates in Euclidean signature where $\tau = it$ and $x_I$, $I = 1 \ldots d$ are the spatial coordinates. We highlight here (as it will become important in the cosmological case) that the **R** in **CRT** reflects a single spatial coordinate, say $x_1 \to -x_1$.

The $SO^+(d,1)$ Lorentz Invariance in $d+1$-dimensional Minkowski spacetime analytically continues to $SO(d+1)$ rotational invariance in Euclidean signature. On the other hand, Unitarity implies that Euclidean correlators satisfy Reflection Positivity, which we will define in the following section. However, it is not necessary to enforce either of these relationships in their entirety to ensure **CRT** symmetry. We do not require full $SO^+(d,1)$ Lorentz Invariance; we simply need the subgroup $SO^+(1,1)$ boost (this will

---

[14]In [73] the authors refer to the $O^{\times}(d,1)$ subgroup which contains the connected components $\{1, T\}$ as the "orthochorous" group, after the Greek word χωρικοσ meaning a spatial territory, but we feel that the name "orthochiral" is more intuitive and distinct, given that this is the group of transformations which preserve the spatial orientation, i.e. the chirality.

become important when studying the implications of the CPT theorem for cosmology). In fact, all that is required is a discrete, $\mathbb{Z}_2$ symmetry from each of them:

$$
\begin{array}{ccccc}
\text{Unitarity} & \longleftrightarrow & \text{Reflection Positivity} & \Longrightarrow & \text{Reflection Reality} \\
\text{Lorentz-Invariance} & \longleftrightarrow & \text{Rotational Invariance} & \Longrightarrow & 180° \text{ Rotation}
\end{array}
$$

where in each row, the first arrow represents the Wick rotation to Euclidean Signature and the second arrow is an implication. The CPT theorem is then simply the statement that the combination of these two symmetries is a **CRT** transformation. Therefore, requiring the symmetry of correlators under both Reflection Reality and this 180° rotation enforces their invariance under the **CRT** transformation.

Suppose, for concreteness, we start by considering vacuum correlators of the form

$$
B_n = \big\langle \phi_1(\tau, x)\phi_2(\tau, x)\dots\phi_n(\tau, x) \big\rangle, \tag{2.1}
$$

where $\phi$ is some generic, real, scalar field and we have suppressed the dependence on the other spatial coordinates that we do not transform. Without loss of generality, we will choose to work in a real basis for all fields, so that the transformation **CT** just becomes **T** and we have an **RT** theorem (this will become important when discussing heavy fields in cosmology).[15] Then the $\mathbb{Z}_2$ transformations of interest act in the following way on Euclidean field theory correlators:

- As we will show in Sections 2.3 and 2.4, Euclidean Reflection Positivity implies a weaker principle know as Reflection Reality. We can enforce this by insisting that our correlators are invariant under the transformation:

$$
\mathbf{RR} : B(\tau, x) \to B^*(-\tau, x), \tag{2.2}
$$

$$
\mathbf{RR}(B_n) = \big\langle \phi_1(-\tau, x)\phi_2(-\tau, x)\dots\phi_n(-\tau, x) \big\rangle^*; \tag{2.3}
$$

- A 180° ($= \pi$) rotation in the $\tau$-$x$ plane corresponds to the transformation:[16]

$$
\mathbf{R}_\pi : B(\tau, x) \to B(-\tau, -x), \tag{2.4}
$$

$$
\mathbf{R}_\pi(B_n) = \big\langle \phi_1(-\tau, -x)\phi_2(-\tau, -x)\dots\phi_n(-\tau, -x) \big\rangle; \tag{2.5}
$$

- Finally, **CRT** is the transformation:

$$
\mathbf{CRT} : B(\tau, x) \to B^*(\tau, -x), \tag{2.6}
$$

$$
\mathbf{CRT}(B_n) = \big\langle \phi_1(\tau, -x)\phi_2(\tau, -x)\dots\phi_n(\tau, -x) \big\rangle^*. \tag{2.7}
$$

It might seem surprising that there is no transformation of the $\tau$ coordinate here. However, this is simply a reflection of the fact that these correlators are a function of Euclidean time, $\tau = it$, so simultaneously complex conjugating and sending $t \to -t$ leaves $\tau$ unchanged.

---

[15]To obtain the discrete transformations for complex scalar fields, one must additionally change all fields/operators $\phi$ to $\phi^\dagger$ whenever complex conjugating.

[16]Some authors refer to this transformation as "CPT". The difference in convention goes back to the earliest papers on the CPT theorem, see [74] for a nice review. In the language of [74], we are defining **T** as a "Wigner time reversal" (which involves a complex conjugation) while other authors use "Schwinger time reversal" (which instead reverses the order of operators. For example, the recent paper by Harlow and Numasawa [75] on gauging "CRT" is actually gauging what we call the 180° rotation, at least in their Euclidean signature construction. It is unclear whether it makes sense to gauge our Wignerian **CRT**, as this would raise the question of what it means to have a nontrivial complex conjugation of amplitudes when going around a non-contractible cycle. (Perhaps it just means the whole amplitude vanishes.)

It is straightforward to see that combining any two of these transformations produces the third and therefore (together with the identity) they form a group which is isomorphic to $\mathbb{Z}_2 \times \mathbb{Z}_2$ (or the Klein 4-group). The consequence of this is that a theory respecting any two of these discrete symmetries will automatically respect the third.[17]

If we exploit this group structure to derive **CRT** from invariance under the other two transformations we recover the standard CPT theorem. However, there also exist two converse directions, where we use

$$\mathbf{CRT} + \mathbf{RR} \implies \mathbf{R}_\pi \,,$$

or else

$$\mathbf{CRT} + \mathbf{R}_\pi \implies \mathbf{RR} \,.$$

This last implication is particularly interesting as, in many field theories, Reflection Reality is strong enough to *accidentally* imply the full principle of Unitarity, despite being weaker that it. This will be discussed further in Section 5.

To generalise to correlators of operators that carry spin, it is of course also necessary to specify how the fields transform under the coordinate transformations above, e.g. by defining them as tensor fields. In the case of a tensor field with an odd number of $\tau$ indices, the transformation $\tau \to -\tau$ will reverse its sign; this is to ensure that e.g. the time component of a real vector field $V_t$ remains self adjoint: $(V_t)^\dagger = V_t \implies (V_\tau)^\dagger = -V_\tau$. (The spatial polarisations are unaffected by this transformation.)

For theories involving spinor fields, it is also necessary to consider projective representations of $\mathbb{Z}_2 \times \mathbb{Z}_2$. In this paragraph, let us assume for simplicity that the theory is fully rotationally covariant. Then a $360°$ rotation gives a factor of $(-1)^{2s}$ (where $s$ is the spin); this is of course the same as the fermion parity $(-1)^F$ for fields that satisfy spin-statistics. Hence, the $180°$ rotation $R_\pi$ has order 4 when acting on half-integer spin fields, whilst Unitarity requires that $\mathbf{RR}^2 = +1$ and hence in a rotationally covariant theory we also need $\mathbf{CRT}^2 = +1$. This is only possible if the group anticommutes in the sense that $[A, B] := ABA^{-1}B^{-1} = -1$ when $A, B$ are any two distinct elements of $\{\mathbf{RR}, \mathbf{CRT}, \mathbf{R}_\pi\}$. One thus finds that the appropriate double cover of $\mathbb{Z}_2 \times \mathbb{Z}_2$ for spinors is the dihedral group $D_4$.[18] We will focus on fields with integer spin representations and not consider half-integer spin fields in detail this paper, but a similar formalism should also provide a much easier approach to obtain a Unitarity constraint for correlators involving spinors in flat space.

## 2.3 Reflection Positivity

In this section we review how the Reflection Positivity of correlators follows from the assumption of Unitarity, as it is an important element in the proof of the CPT theorem presented in the previous section, and may be unfamiliar to a reader.

Consider a ket state defined on the $\tau = 0$ slice of a Euclidean field theory. This may be defined by (an arbitrary sum/integral of) a time ordered Euclidean correlator, in which all operators are inserted on the

---

[17]One might at first glance think that it is required to have a $90°$ rotation $r : \tau \to x, x \to -\tau$ to relate Reflection Reality to **CRT**. This would be true if the proof required a conjugation $r(\mathrm{RR})r^{-1} = \mathbf{CRT}$. While this would be necessary to establish an isomorphism between RR and **CRT**, proving the CPT theorem requires only taking the group product with the $180°$ rotation, as the product of any 2 symmetries is another symmetry.

[18]With the addition of one additional magical fact that Unitarity requires $\mathbf{CRT}^2 = (-1)^{2s}(-1)^F$, which requires a close inspection of the transformation properties of 2-pt correlators to prove, the considerations in this paragraph provide all the necessary ingredients to prove the spin-statistics theorem. But spin-statistics requires at least covariance under $90°$ rotations, not just $180°$ rotations, so it is less robust than the CPT theorem.

side $\tau < 0$:

$$|\psi\rangle = \sum\!\!\!\!\!\!\int f\, \mathcal{O}_n(\tau_n)\dots\mathcal{O}_2(\tau_2)\mathcal{O}_1(\tau_1)\,|0\rangle \tag{2.8}$$

with $\tau_1 \leq \tau_2 \leq \dots \leq \tau_n < 0$. We have suppressed writing the spatial coordinates for notational conciseness as they are left invariant by this transformation. Note that $f$ has an explicit dependence on $\tau$[19] and is an arbitrary distribution over the spatial coordinates and choices of fields, such that the resulting $n$-point correlator is finite. Similarly, the corresponding bra is[20]

$$\langle\psi| = \sum\!\!\!\!\!\!\int f^*_{\tau\to-\tau}\, \langle 0|\, \mathcal{O}_1^\dagger(-\tau_1)\mathcal{O}_2^\dagger(-\tau_2)\dots\mathcal{O}_n^\dagger(-\tau_n)\,. \tag{2.9}$$

The positivity of the inner product $\langle\psi|\psi\rangle \geq 0$ therefore implies the principle of Reflection Positivity:

$$\left(\sum\!\!\!\!\!\!\int f^*_{\tau\to-\tau}\, \langle 0|\, \mathcal{O}_1^\dagger(-\tau_1)\mathcal{O}_2^\dagger(-\tau_2)\dots\mathcal{O}_n^\dagger(-\tau_n)\right)\left(\sum\!\!\!\!\!\!\int f\, \mathcal{O}_n(\tau_n)\dots\mathcal{O}_2(\tau_2)\mathcal{O}_1(\tau_1)\,|0\rangle\right) \geq 0\,, \tag{2.10}$$

a principle which is much simpler in spirit than the number of symbols it takes to write it out. Note that even in a unitary theory, it is possible for the RHS to vanish if the correlator vanishes, for example if we insert the equation of motion in a free theory, or a pure gauge state in a gauge theory. Such null states may be projected out of the theory, leaving a Hilbert space with a positive-definite inner product.

For a fixed number, $N$, of insertions on each side, we would obtain the positivity of some $2N$-point function. However, it is also allowed to consider states coming from quantum superpositions of different values of $N$. Because the sums are taken independently on both sides of the $\tau = 0$ surface, Reflection Positivity also places constraints on the correlations with odd numbers of points. It is not, however, allowed for the operators which define the ket to stray "offside" onto the wrong side of the $\tau = 0$ surface, even if you include the corresponding reflected operator.

In Lorentzian signature, Unitarity consists of two statements: (i) there exists a positive norm on the state space, and (ii) this norm is preserved under time evolution. Strictly speaking, Reflection Positivity is the Wick rotation of (i), not (ii).[21] But (ii) follows if we additionally use the $\tau$-translation invariance of the Euclidean path integral. This implies that if we insert an empty strip of width $\Delta\tau$ between two correlators, it doesn't matter which side we insert it on:

$$|\chi\rangle^\dagger\left(e^{-H\Delta\tau}|\psi\rangle\right) = \left(e^{-H\Delta\tau}|\chi\rangle\right)^\dagger|\psi\rangle\,. \tag{2.11}$$

from which it follows that $H = H^\dagger$, guaranteeing that the Lorentzian evolution $e^{iHt}$ preserves the norm. (More generally, if one assumes the Euclidean partition function is a covariant functional of the metric, one could deduce that Lorentzian evolution with respect to arbitrary lapse and shift functions will also be unitary.) Some constraints from only having states with positive norm were explored in [76, 77], and explained in a particularly clear way in [78].

---

[19] To avoid cluttered equations in the discussion to follow we will only highlight the dependence on $\tau$, but the correlators continue to depend on the other spatial coordinates.

[20] In these and subsequent expressions, we define $\mathcal{O}^\dagger(\tau)$ as the operator obtained by *first* acting on $\mathcal{O}(0)$ with the $\dagger$, and *then* translating it to $\tau \neq 0$. These operations do not commute, because $\tau^\dagger = -\tau$ in Euclidean field theory (in equal-time quantisation). It is therefore reasonable to use notation in which $\mathcal{O}(\tau)^\dagger = \mathcal{O}^\dagger(-\tau)$, in order to distinguish cases in which the $\dagger$ acts on the argument, from those in which it does not. To avoid confusion, in the main body of the paper we will always place the $\dagger$ on the operator, meaning that any reversal of $\tau$ coordinate will be stated explicitly in the $\tau$-argument of the operator.

[21] Unitarity also says that pure states evolve to pure states, but we take this for granted here.

## 2.4 Reflection Reality

Since every positive number is real, Reflection Positivity obviously implies the weaker principle that:

$$\left( \sum\!\!\!\!\!\!\int f^*_{\tau \to -\tau} \langle 0 | \, \mathcal{O}_1^\dagger(-\tau_1)\mathcal{O}_2^\dagger(-\tau_2)\ldots\mathcal{O}_n^\dagger(-\tau_n) \right) \left( \sum\!\!\!\!\!\!\int f \, \mathcal{O}_n(\tau_n)\ldots\mathcal{O}_2(\tau_2)\mathcal{O}_1(\tau_1) \, | 0 \rangle \right) \in \mathbb{R} \,. \tag{2.12}$$

This is simply the assertion that the inner product is *real* (but not necessarily positive). This is the analogue of (i) in the previous section. The further statement corresponding to (ii), that this (possibly indefinite) real product is preserved by Lorentzian time evolution, follows from the same argument in (2.11).

Eq. (2.12) does not yet manifestly take the form of a $\mathbb{Z}_2$ symmetry, because of the annoying restrictions that (a) the amplitude must be a product of two conjugate correlators, (b) each of which is restricted to one side of $\tau = 0$. However, it turns out that when we only care about deriving a reality condition, neither of these restrictions are actually required. To see this, simply consider a state of the form

$$|\beta\rangle = |0\rangle + \beta|\psi\rangle \,, \tag{2.13}$$

where $\beta$ is an arbitrary complex number. Now Eq. (2.12) tells us that

$$\langle \beta | \beta \rangle = \langle 0|0\rangle + \beta\langle 0|\psi\rangle + \beta^*\langle \psi|0\rangle + |\beta|^2\langle \psi|\psi\rangle \in \mathbb{R} \,. \tag{2.14}$$

But the first and last terms are also real by Eq. (2.12). And the only way the sum of the middle terms can be real for all values of $\beta$, is if

$$\langle \psi|0\rangle = \langle 0|\psi\rangle^* \,, \tag{2.15}$$

from which it follows that

$$\left( \sum\!\!\!\!\!\!\int f^*_{\tau \to -\tau} \langle 0 | \, \mathcal{O}_1^\dagger(-\tau_1)\mathcal{O}_2^\dagger(-\tau_2)\ldots\mathcal{O}_n^\dagger(-\tau_n) \right) |0\rangle = \langle 0 | \left( \sum\!\!\!\!\!\!\int f \, \mathcal{O}_n(\tau_n)\ldots\mathcal{O}_2(\tau_2)\mathcal{O}_1(\tau_1) \, | 0 \rangle \right)^* \,. \tag{2.16}$$

Similarly, at this point the restriction that the correlators remain on one side of the $\tau = 0$ surface can be lifted, either by using translation invariance, or by analyticity of the Euclidean position space correlators. It follows that an arbitrary correlator satisfies the following Reflection Reality (**RR**) condition:

$$\big\langle \mathcal{O}_1(\tau_1;\mathbf{x}_1)\mathcal{O}_2(\tau_2;\mathbf{x}_2)\ldots\mathcal{O}_n(\tau_n;\mathbf{x}_n) \big\rangle = \big\langle \mathcal{O}_1^\dagger(-\tau_1;\mathbf{x}_1)\mathcal{O}_2^\dagger(-\tau_2;\mathbf{x}_2)\ldots\mathcal{O}_n^\dagger(-\tau_n;\mathbf{x}_n) \big\rangle^* \tag{2.17}$$

which now takes the form of a $\mathbb{Z}_2$ symmetry that reverses the sign of Euclidean time $\tau$ and also complex conjugates the amplitude. This symmetry is therefore required to hold in any unitary field theory. But it is somewhat weaker than Unitarity as it is compatible with the existence of states with negative-norm.

## 2.5 Lorentzian Correlators

In Lorentzian signature, **RR** is simply a hermitian conjugation, and implies that self-adjoint fields have real expectation values. As $\phi = \phi^\dagger$, the action of taking the adjoint simply reverses the order of operators, and (in operator ordering) **RR** implies:

$$\big\langle \phi_1(t_1;x_1)\phi_2(t_2;x_2)\ldots\phi_n(t_n;x_n) \big\rangle = \big\langle \phi_n(t_n;x_n)\ldots\phi_2(t_2;x_2)\phi_1(t_1;x_1) \big\rangle^* \tag{2.18}$$

but without any positivity conditions. We also now introduce time ordered correlators as the object of consideration in Lorentzian signature,

$$G_n = \big\langle \mathcal{T}\phi_1(t_1;x_1)\phi_2(t_2;x_2)\ldots\phi_n(t_n;x_n) \big\rangle \,. \tag{2.19}$$

The same group structure then emerges from the transformations:

- Reflection Reality is the statement that:

$$\mathbf{RR} : G(t; x) \to \overline{G}^{*}(t; x) \,, \tag{2.20}$$

$$\mathbf{RR}(G_n) = \left\langle \overline{\mathcal{T}} \phi_1(t_1; x_1) \phi_2(t_2; x_2) \ldots \phi_n(t_n; x_n) \right\rangle^{*} \,, \tag{2.21}$$

where the bar indicates that time ordering has been exchanged with anti-time ordering (in other words, the order would have been reversed if we had been using operator ordering). For equal time correlation functions this relationship therefore enforces that these correlators are real. In other words, $\mathbf{RR}$ implies the existence of a real norm on the Hilbert space (which need not be positive). As we shall see later, this norm is also preserved under Lorentzian time evolution.

- The $180°$ rotation for time-ordered correlators becomes:

$$\mathbf{R}_\pi : G(t; x) \to G(-t; -x) \,, \tag{2.22}$$

$$\mathbf{R}_\pi(G_n) = \left\langle \mathcal{T} \phi_1(-t_1; -x_1) \phi_2(-t_2; -x_2) \ldots \phi_n(-t_n; -x_n) \right\rangle \,, \tag{2.23}$$

where time ordering has been preserved notwithstanding the reversal of the sign of $t$.

- Finally, $\mathbf{CRT}$ is the transformation:

$$\mathbf{CRT} : G(t; x) \to \overline{G}^{*}(-t; -x) \,, \tag{2.24}$$

$$\mathbf{CRT}(G_n) = \left\langle \overline{\mathcal{T}} \phi_1(-t_1; -x_1) \phi_2(-t_2; -x_2) \ldots \phi_n(-t_n; -x_n) \right\rangle^{*} \,. \tag{2.25}$$

# 3 CPT in de Sitter

We previously showed that in flat space the CPT theorem in Euclidean signature can be understood through a $\mathbb{Z}_2 \times \mathbb{Z}_2$ symmetry group that relates the discrete symmetries of Reflection Reality $\mathbf{RR}$, the $180°$ Euclidean rotation $\mathbf{R}_\pi$, and $\mathbf{CRT}$. The first two of these symmetries are guaranteed by the Lorentzian properties of Unitarity and $SO^+(1, 1)$ invariance.

Analogous properties also exist in de Sitter, leading to an isomorphic $\mathbb{Z}_2 \times \mathbb{Z}_2$ group. In this section we will determine the effect of the three discrete symmetries $\mathbf{RR}$, $\mathbf{R}_\pi$, and $\mathbf{CRT}$ on the coordinates by considering the embedding of de Sitter in a $D + 1$-dimensional Minkwoski spacetime.

## 3.1 Global de Sitter

Similar to how a Lorentz invariant theory in a $D + 1$-dimensional (where $D = d + 1$) Minkowski spacetime is invariant under the restricted Lorentz group $SO^+(D + 1, 1)$, a de Sitter invariant theory in a $d + 1$-dimensional de Sitter spacetime is invariant under $SO^+(d + 1, 1)$, which consists of those de Sitter transformations that preserve both the orientation of space and the direction of time.

In Euclidean signature, the $SO^+(d + 1, 1)$ de Sitter invariance in $d + 1$-dimensional de Sitter spacetime becomes $SO(d + 2)$ rotational invariance of a $S^{d+1}$ sphere of radius $\ell$ embedded in $\mathbb{R}^{D+1}$:

$$\delta_{AB} x^A x^B = \tau^2 + x_I x^I = \ell^2 \,, \tag{3.1}$$

and hence its metric is given by,

$$\mathrm{d}s^2 = \mathrm{d}\Theta^2 + \ell^2 \cos^2\left(\frac{\Theta}{\ell}\right) \left(\mathrm{d}\Phi_d^2 + \sin^2(\Phi_d)\,\mathrm{d}\Omega_{d-1}^2\right) \,. \tag{3.2}$$

The embedding space and the coordinates of the Euclidean dS sphere are related by

$$\tau = \ell \sin\left(\frac{\Theta}{\ell}\right), \tag{3.3}$$

$$x^I = \ell \cos\left(\frac{\Theta}{\ell}\right) \sin(\Phi_d) z^i, \tag{3.4}$$

$$x^D = \ell \cos\left(\frac{\Theta}{\ell}\right) \cos(\Phi_d), \tag{3.5}$$

where the $z^i$ describe a unit $S^{d-1}$ satisfying $z_i z^i = 1$ and are expressible in terms of $d-1$ angular variables, $\Phi_i$.

We now act with the same flat space transformations described in Section 2, but now in the $\tau$-$x^D$ plane of the embedding space. In the embedding space the **CRT** transformation and rotations act on these coordinates as

$$\mathbf{CRT} : (\tau, x^1, \ldots, x^d, x^D) \to (\tau, x^1, \ldots, x^d, -x^D), \tag{3.6}$$

$$\mathbf{R}_\theta : (\tau, x^1, \ldots, x^d, x^D) \to (\tau \cos(\theta) + i x^D \sin(\theta), x^1, \ldots, x^d, x^D \cos(\theta) + i\tau \sin(\theta)). \tag{3.7}$$

The reason these are identical for $\theta = \pi$, despite representing different transformations when acting on correlators, is that **CRT** additionally includes complex conjugation, which does not affect the coordinates. We also note that in this formulation this rotation can be understood as an analytic continuation of the Lorentz boost to imaginary values of the rapidity, $\theta = i\xi$. The action of the $\mathbb{Z}_2$ transformations from Section 2 on the Euclidean dS correlators is the following:

- Reflection Reality requires invariance under the transformation:

$$\mathbf{RR} : B(\Theta; \Phi) \to B^*(-\Theta; \Phi), \tag{3.8}$$

$$\mathbf{RR}(B_n) = \left\langle \phi_1(-\Theta_1; \Phi_1) \phi_2(-\Theta_2; \Phi_2) \ldots \phi_n(-\Theta_n; \Phi_n) \right\rangle^*, \tag{3.9}$$

where Reflection Reality maps an insertion on the $S^{d+1}$ across the $\Theta = 0$ plane and a global complex conjugation acting on both complex fields and amplitudes/correlators;

- A 180° rotation corresponds to the transformation:

$$\mathbf{R}_\pi : B(\Theta; \Phi) \to B(-\Theta; \pi - \Phi), \tag{3.10}$$

$$\mathbf{R}_\pi(B_n) = \left\langle \phi_1(-\Theta_1; \pi - \Phi_1) \phi_2(-\Theta_2; \pi - \Phi_2) \ldots \phi_n(-\Theta_n; \pi - \Phi_n) \right\rangle, \tag{3.11}$$

where the rotation amounts to a reflection from a pole on one axis of the sphere to the opposite pole;

- Finally, **CRT** is the transformation:

$$\mathbf{CRT} : B(\Theta; \Phi) \to B^*(\Theta; \pi - \Phi), \tag{3.12}$$

$$\mathbf{CRT}(B_n) = \left\langle \phi_1(\Theta_1; \pi - \Phi_1) \phi_2(\Theta_2; \pi - \Phi_2) \ldots \phi_n(\Theta_n; \pi - \Phi_n) \right\rangle^*, \tag{3.13}$$

where the transformation is a reflection from a pole on one axis of the sphere to the opposite pole as well as a global complex conjugation acting on both complex fields and amplitudes/correlators.

When we do a Wick rotation $\tau = i x^0$ to Lorentzian signature, this converts the embedding Euclidean space into a Minkowski space:

$$\mathrm{d}s^2 = \eta_{\mu\nu} \, \mathrm{d}x^\mu \, \mathrm{d}x^\nu = -(\mathrm{d}x^0)^2 + \mathrm{d}x^I \, \mathrm{d}x^I, \quad I = 1, \ldots, D. \tag{3.14}$$

De Sitter spacetime then corresponds to a hyperboloid in these coordinates [79] satisfying,

$$\eta_{\mu\nu}x^\mu x^\nu = -(x^0)^2 + x^I x^I = \ell^2 = \frac{1}{H^2} = \frac{d(d-1)}{2\Lambda}\,, \tag{3.15}$$

where $\ell$ is the de Sitter radius which is inversely related to the usual Hubble constant $H$ parameterising the expansion rate set by the positive cosmological constant $\Lambda$. This hyperboloid can be sliced up in many ways; the slicings we will consider are the closed slicing of global de Sitter (this section) and the Poincaré flat slicing (Section 3.2).[22] The flat slicing coordinates of interest to inflationary cosmology cover only the region with $x_0 + x_D > 0$, whilst the closed slicing of global de Sitter covers the whole hyperboloid.

In Lorentzian global de Sitter coordinates (also referred to as closed slicing), the metric w.r.t. the proper time coordinate $T$ is given by

$$\mathrm{d}s^2 = -\mathrm{d}T^2 + \ell^2 \cosh^2\left(\frac{T}{\ell}\right)\left(\mathrm{d}\Phi_d^2 + \sin^2(\Phi_d)\,\mathrm{d}\Omega_{d-1}^2\right)\,, \tag{3.16}$$

where $T \in \mathbb{R}$ and $\mathrm{d}\Omega_{d-1}^2$ is the round metric on the unit $d-1$-sphere. We can see from the metric that constant $T$ surfaces are $d$-spheres which shrink from the past asymptotic boundary $\mathcal{I}^-$ at $T = -\infty$ to $T = 0$ and grow from $T = 0$ to the future asymptotic boundary $\mathcal{I}^+$ at $T = +\infty$. The embedding space and global de Sitter coordinates are related as

$$x^0 = \ell \sinh\left(\frac{T}{\ell}\right)\,, \tag{3.17}$$

$$x^i = \ell \cosh\left(\frac{T}{\ell}\right)\sin(\Phi_d)z^i\,, \tag{3.18}$$

$$x^D = \ell \cosh\left(\frac{T}{\ell}\right)\cos(\Phi_d)\,. \tag{3.19}$$

The **CRT** transformation and rotations act on the embedding space coordinates as

$$\mathbf{CRT} : (x^0, x^1, \ldots, x^d, x^D) \to (-x^0, x^1, \ldots, x^d, -x^D)\,, \tag{3.20}$$

$$\mathbf{R}_\theta : (x^0, x^1, \ldots, x^d, x^D) \to (x^0\cos(\theta) + ix^D\sin(\theta), x^1, \ldots, x^d, x^D\cos(\theta) + ix^0\sin(\theta))\,. \tag{3.21}$$

The **CRT** transformation in Lorentzian global de Sitter is then straightforwardly given by

$$\mathbf{CRT} : (T, \Phi_1, \ldots, \Phi_d) \to (-T, \Phi_1, \ldots, \pi - \Phi_d)\,, \tag{3.22}$$

$$\mathbf{CRT} : G(T; \Phi_d) \to \overline{G}^*(-T; \pi - \Phi_d)\,, \tag{3.23}$$

$$\mathbf{CRT}(G_n) = \left\langle \overline{\mathcal{T}}\phi_1(-T_1; \pi - \Phi_{d1})\phi_2(-T_2; \pi - \Phi_{d2})\ldots\phi_n(-T_n; \pi - \Phi_{dn})\right\rangle^*\,. \tag{3.24}$$

Unfortunately, an arbitrary rotation doesn't have such a simple expression; the reason for this is that this rotation in the embedding space formalism misaligns the axis $x_0$ from the axis of the hyperboloid, so that the slices are now ellipsoids not spheres and attempting to parameterise the resulting slices using spherical coordinates fails. However, for the specific 180° rotation in Section 2.5 we have

$$\mathbf{R}_\pi : (T, \Phi_1, \ldots, \Phi_d) \to (-T, \Phi_1, \ldots, \pi - \Phi_d)\,, \tag{3.25}$$

$$\mathbf{R}_\pi : G(T; \Phi_d) \to G(-T; \pi - \Phi_d)\,, \tag{3.26}$$

$$\mathbf{R}_\pi(G_n) = \left\langle \mathcal{T}\phi_1(-T_1; \pi - \Phi_{d1})\phi_2(-T_2; \pi - \Phi_{d2})\ldots\phi_n(-T_n; \pi - \Phi_{dn})\right\rangle\,, \tag{3.27}$$

which as before, acts on the coordinates in the same way as the **CRT** transformation, with the difference between the two reappearing when we consider a complex object. Finally we can infer that the **RR**

---

[22]We leave the implications for the de Sitter static patch to future work [26].

transformation in these coordinates is given by

$$\mathbf{RR} : (T, \Phi_1, \ldots, \Phi_d) \to (T, \Phi_1, \ldots, \Phi_d) \,, \tag{3.28}$$

$$\mathbf{RR} : G(T; \Phi_d) \to \overline{G}^*(T; \Phi_d) \,, \tag{3.29}$$

$$\mathbf{RR}(G_n) = \left\langle \, \overline{\mathcal{T}} \phi_1(T_1; \Phi_{d1}) \phi_2(T_2; \Phi_{d2}) \ldots \phi_n(T_n; \Phi_{dn}) \, \right\rangle^* \,, \tag{3.30}$$

where, once again, we see that for equal time correlation functions this relationship enforces that these correlators are real, i.e. $\mathbf{RR}$ implies the existence of a real norm on the Hilbert space, which need not be positive.[23]

## 3.2   Poincaré de Sitter

Having shown in Section 3.1 that global de Sitter inherits the three discrete symmetries of interest in the CPT theorem from the embedding space, we can now consider these transformations in the Poincaré patch, where we perform inflationary calculations. However, things will become much less trivial here, since a *single* Poincaré patch explicitly chooses a specific orientation of time from the perspective of the global time coordinate. Thus, the discrete transformations will each require an analytic continuation outside of the physical domain of the original Poincaré patch. In this section, we will establish the behaviour of this analytic continuation at the level of the coordinates and in Sections 3.3-3.5 we will discuss the implications for the inflationary observables known as the cosmological correlators and the Wavefunction of the Universe.

In the flat slicing of the Poincaré patch we have the metric,

$$\mathrm{d}s^2 = -\,\mathrm{d}t^2 + e^{2Ht}\,\mathrm{d}y^i\,\mathrm{d}y^i \,, \quad i = 1, \ldots, d \,, \tag{3.31}$$

where $t \in \mathbb{R}$ is generally referred to as cosmological time in the literature. These coordinates are related to the embedding space coordinates by

$$x^0 = \frac{1}{H}\sinh\left(Ht\right) + \frac{H}{2}y^i y^i e^{Ht} \,, \tag{3.32}$$

$$x^D = \frac{1}{H}\cosh\left(Ht\right) - \frac{H}{2}y^i y^i e^{Ht} \,, \tag{3.33}$$

$$x^i = e^{Ht}y^i \,, \quad 1 \le i \le d \,, \tag{3.34}$$

The transformation $t \to t + \frac{i\pi}{H}$ recovers both parts of the $\mathbf{CRT}$ transform in these coordinates but it naturally reflects all spatial coordinates rather than just one.[24] It is possible to recover just the transformation in (3.20) by additionally transforming all $y^i \to -y^i$. Unfortunately, analytically continuing the time variable is not easily interpreted as a time reversal. However, if we go to conformal time[25], where the metric becomes

$$\mathrm{d}s^2 = \left(-\frac{1}{H\eta}\right)^2 \left(-\,\mathrm{d}\eta^2 + \mathrm{d}y^i\,\mathrm{d}y^i\right) \,, \quad i = 1, \ldots, d \,, \tag{3.36}$$

---

[23]The constraint from $\mathbf{RR}$ that equal time correlators are real is consistent with what has been found in the literature when exploring the constraints that enforcing a positive norm Hilbert space places on de Sitter correlators [76–78].

[24]Some authors [80, 81] claim that a transformation which involves flipping both $H$ and $t$ is "CPT" and that this can be used to obtain a unitary formulation of QFT in curved spacetime. However it is clear that this would not map on to the $\mathbf{CRT}$ transformation defined by the embedding space formalism and thus it is not clear that this transformation has any relation to bulk Unitarity.

[25]In cosmology conformal time and cosmological time are related by $\mathrm{d}t = a\,\mathrm{d}\eta$ where for de Sitter we have $a(t) = e^{Ht}$ and thus

$$\int_{-\infty}^{\eta} \mathrm{d}\eta' = \int_{-\infty}^{t} e^{-Ht'}\,\mathrm{d}t' \implies \eta = -\frac{1}{H}e^{-Ht} \,. \tag{3.35}$$

then our new coordinates are related to those in the embedding space by

$$x^0 = \frac{1}{H}\sinh(Ht) + \frac{H}{2}y_i y_i e^{Ht} = \frac{H^2\eta^2 - 1 - H^2 y^i y^i}{2H^2\eta} = \frac{\eta^2 - 1 - y^i y^i}{2H\eta}\,, \tag{3.37}$$

$$x^D = \frac{1}{H}\cosh(Ht) - \frac{H}{2}y^i y^i e^{Ht} = \frac{H^2 y^i y^i - 1 - H^2\eta^2}{2H^2\eta} = \frac{y^i y^i - 1 - \eta^2}{2H\eta}\,, \tag{3.38}$$

$$x^i = e^{Ht}y^i = -\frac{y^i}{H\eta}\,, \quad 1 \le i \le d\,, \tag{3.39}$$

$$\eta = y^0 = \frac{-1}{H(x^0 + x^D)}\,, \quad y^i = \frac{x^i}{x^0 + x^D}\,. \tag{3.40}$$

In the final equality for (3.37) and (3.38) we have used the fact that we can rescale $\eta \to \eta/H$ and $y^i \to y^i/H$ without loss of generality. In order to be consistent with the cosmology literature we will work with the parametrisation in which $\eta \le 0$ (with $\eta = 0^-$ being the same future asymptotic boundary $\mathcal{I}^+$ in global de Sitter) and $y^i \in \mathbb{R}$, i.e. we will pick the Poincaré patch covering $x^0 + x^D \ge 0$. Such foliations only cover half of de Sitter space; however **CRT** or the $180°$ rotation will take you from one Poincaré patch to the other.

We see that the **CRT** transformation acts on the coordinates as:

$$\mathbf{CRT} : (\eta, y_1, \ldots, y_d) \to -(\eta, y_1, \ldots, y_d) = (-\eta, -y_1, \ldots, -y_d)\,. \tag{3.41}$$

Therefore, in order to recover a **CRT** transformation in a single spatial and time coordinate in the embedding space, or equivalently in global de Sitter slicing, it is necessary to transform all spatial coordinates in the Poincaré patch in addition to analytically continuing our conformal time variable.[26] The analytic continuation in conformal time $\eta$ is just an artefact of the fact that **T** (or **CRT**) is a transformation within the full global de Sitter which is not captured by a single Poincaré patch. Thus **CRT** involves an analytic continuation since the observables are defined for one Poincaré patch where $\eta \le 0$. We will discuss in Sections 3.3-3.5 how this analytic continuation manifests itself at the level of observables.

We can similarly consider how the Lorentzian $SO^+(1,1)$ boost in the embedding space manifests in the Poincaré patch, since it is this boost which, when analytically continued, becomes $SO(2)$ in Euclidean signature and gives us the necessary $\mathbf{R}_\pi$ rotation in the $\mathbb{Z}_2 \times \mathbb{Z}_2$ group. From the perspective of the Poincaré patch the generator of the $SO^+(1,1)$ boost becomes:

$$L_{0D} \equiv x_0 \frac{\partial}{\partial x^D} - x_D \frac{\partial}{\partial x^0} \tag{3.42}$$

$$= \eta \frac{\partial}{\partial \eta} + y^i \frac{\partial}{\partial y^i} \equiv D\,, \tag{3.43}$$

where we have used the fact that

$$\frac{\partial}{\partial x^0} \equiv \frac{\partial y^\mu}{\partial x^0}\frac{\partial}{\partial y^\mu} = H\eta^2 \frac{\partial}{\partial \eta} + H\eta y^i \frac{\partial}{\partial y^i} \tag{3.44}$$

$$\frac{\partial}{\partial x^D} \equiv \frac{\partial y^\mu}{\partial x^D}\frac{\partial}{\partial y^\mu} = H\eta^2 \frac{\partial}{\partial \eta} + H\eta y^i \frac{\partial}{\partial y^i}\,, \tag{3.45}$$

Hence we find that the $SO^+(1,1)$ boost acts as spacetime dilatations (which we will refer to as just Dilatations for the sake of simplicity) on the Poincaré patch, which in the cosmology literature is generally

---

[26]From the perspective of flat space, global de Sitter, and any spacetimes with the full $SO(1,1)$ symmetry, rather than the restricted $SO^+(1,1)$, **CRT** is a trivial crossing symmetry which involves no analytic continuation.

referred to as bulk Scale Invariance.[27] The action of the rotation operator on the coordinates in embedding space is:

$$(x_0, x_1, \ldots, x_d, x_D) = \left( -\frac{1}{H^2\eta} + \eta - \frac{y_i y_i}{2\eta}, -\frac{y_1}{H\eta}, \ldots, -\frac{y_d}{H\eta}, -\frac{1}{H^2\eta} - \eta + \frac{y_i y_i}{2\eta} \right), \tag{3.46}$$

$$\mathbf{R}_\theta : (x_0, x_1, \ldots, x_d, x_D) \to (x_0 \cos(\theta) + ix_D \sin(\theta), x_1, \ldots, x_D \cos(\theta) + ix_0 \sin(\theta)) \tag{3.47}$$

$$= \left( -\frac{e^{i\theta}}{H^2\eta} + \frac{\eta}{e^{i\theta}} - \frac{y_i y_i}{2\eta e^{i\theta}}, -\frac{y_1}{H\eta}, \ldots, -\frac{y_d}{H\eta}, -\frac{e^{i\theta}}{H^2\eta} - \frac{\eta}{e^{i\theta}} + \frac{y_i y_i}{2\eta e^{i\theta}} \right). \tag{3.48}$$

From this we can read off that this rotation is equivalent to the Poincaré patch coordinate transformation,

$$\mathbf{R}_\theta : (\eta, y_1, \ldots, y_d) \to e^{-i\theta}(\eta, y_1, \ldots, y_d). \tag{3.49}$$

Just as the flat space transformation was an analytic continuation of Lorentz boosts to imaginary rapidity, this rotation has become the analytic continuation of the Dilatation operator to complex scales. In particular the 180° rotation acts on the coordinates as a negative Dilatation:

$$\mathbf{D}_{-1} : (\eta, y_1, \ldots, y_d) \to -(\eta, y_1, \ldots, y_d) = (-\eta, -y_1, \ldots, -y_d). \tag{3.50}$$

Note that the action on coordinates is the same as for **CRT**. This is expected as the same thing happens in flat space; the key differences for the 180° rotation are i) it does not involve any complex conjugation, and ii) the correlator continues to be time-ordered. This is particularly interesting in the context of inflation where we often break de Sitter boosts (these act as special conformal transformations at the boundary) and so we lose the full set of de Sitter isometries. Here we need only to invoke an analytic continuation of Dilatations, which comes naturally from the embedding space formalism.

In momentum space we would have almost the same relations:

$$\mathbf{CRT} : (\eta, k_1, \ldots, k_d) \to -(\eta, -k_1, \ldots, -k_d) = (-\eta, k_1, \ldots, k_d), \tag{3.51}$$

$$\mathbf{D}_{-1} : (\eta, y_1, \ldots, y_d) \to -(\eta, k_1, \ldots, k_d) = (-\eta, -k_1, \ldots, -k_d). \tag{3.52}$$

apart from the fact that the complex conjugation in **CRT** in position space reverses the sign of $k$, due to the Fourier transform, cancelling out the reflection of each spatial coordinate.

We will now move on to discussing the constraints these discrete symmetries impose on the inflationary observables known as cosmological correlators.

## 3.3 Inflationary Observables

In the cosmological context the purpose of studying de Sitter spacetime is to understand the behaviour of quantum fluctuations at the end of inflation. These perturbations then grow over the history of the universe to produce all the structure that we see. Due to the randomness inherent to the quantum nature of these fluctuations and the single universe that we have to observe we study them through their correlation functions. In this section we focus on cosmological correlators calculated using the in-in formalism, since constraints for them only require the *spacetime symmetries* considered in Section 3.2. In this paper, we will focus only on observables involving fields with integer spin representations and will not consider spinor

---

[27]Note that this is different from the usual notion of Scale Invariance in the CFT literature which is invariance under the generator $D = x^i \partial_i$. However we see that in the limit of $\eta \to 0^-$, $\lim_{\eta \to 0^-} D_{\eta,y} = D$, i.e. they are equivalent at the asymptotic time boundary of de Sitter. In actual fact it is well known in the literature that the group of **ALL** the bulk Killing symmetries (for which the corresponding Ward identities are sometimes referred to as the anomalous Ward identities in the literature) of de Sitter becomes the Euclidean conformal group $SO(d+1, 1)$ at the asymptotic time boundary $\eta = 0^-$. This is the fundamental intuition for a "dS/CFT".

fields in detail. However, a similar formalism should provide a much easier approach to deriving symmetry and Unitarity constraints for observables involving half-integer spin fields in cosmology (see e.g. [65, 82–94] for discussions of spinors in de Sitter space and cosmology), which has yet to be done in the cosmology literature.

(Later, we will revisit these same observables using the Wavefunction of the Universe (WFU) which introduces a new type of $n$-pt functions known as the wavefunction coefficients. However, as we will discuss in Section 3.5 this requires consideration of a different type of symmetry, namely the *local Lagrangian symmetries* discussed in Section 4, as it is those which will impose non-trivial constraints on the WFU and wavefunction coefficients.)

The correlators of interest in cosmology are equal time correlation functions evaluated on some time slice, usually taken to be the asymptotic future of de Sitter or some (quasi-)de Sitter inflationary spacetime,

$$\langle \mathcal{O}(\eta) \rangle = \langle \Psi(\eta) | \mathcal{O}(\eta) | \Psi(\eta) \rangle \,. \tag{3.53}$$

We do not (a priori) know the state $|\Psi(\eta)\rangle$. However, we assume that the universe was in a vacuum of the full theory, $|\Psi\rangle = |\Omega\rangle$. Just like in flat space it will prove very helpful to move to the interaction picture where operators evolve according to the free Hamiltonian whilst the evolution of states is dictated by the interacting Hamiltonian. Then these correlators are

$$\langle \mathcal{O}(\eta) \rangle = \langle \Omega(\eta_i) | U_I^\dagger(\eta_i, \eta) \mathcal{O}_I(\eta) U_I(\eta_i, \eta) | \Omega(\eta_i) \rangle \,, \tag{3.54}$$

which are known as "in-in" correlators due to the fact that both vacuum states are defined in the past at $\eta_i$. This is in opposition to flat space amplitudes where the ket state corresponds to the future "out" vacuum.

We could perform a Dyson expansion for this expression, using the fact that this time evolution operator is

$$U(\eta_i, \eta) = \mathcal{T} \exp\left[ -i \int_{\eta_i}^{\eta} \mathrm{d}\eta H_{\text{int}}[\phi, \phi'] \right] \,. \tag{3.55}$$

However, $|\Omega(\eta_i)\rangle$ is the in vacuum of the full theory (despite being expressed in the interaction picture). This is relevant as the operators, $\mathcal{O}$, will be constructed from the interaction picture creation and annihilation operators, which act straightforwardly on the free (Bunch-Davies) vacuum, $|0\rangle$. By taking $\eta_i$ to the infinite past we can turn off the interactions adiabatically so that the free and interacting vacua coincide. We will discuss the procedure by which this is done more extensively in Section 7.1.

## 3.4 Symmetries of the Cosmological Correlators

In this section we explore how the symmetry transformations **CRT** and $\mathbf{D}_{-1}$ from Section 3.2 act on in-in cosmological correlators and thus infer the constraint from Reflection Reality (**RR**). The cosmological correlators can be expressed as

$$B_n(\eta_0; \mathbf{y}_1 \ldots \mathbf{y}_n) = \langle \Omega(\eta_0) | \phi_1(\eta_0; \mathbf{y}_1) \ldots \phi_n(\eta_0; \mathbf{y}_n) | \Omega(\eta_0) \rangle \tag{3.56}$$

$$= \langle \Omega(\eta_i) | U^\dagger(\eta_i, \eta_0) \phi_1(\eta_0; \mathbf{y}_1) \ldots \phi_n(\eta_0; \mathbf{y}_n) U(\eta_i, \eta_0) | \Omega(\eta_i) \rangle \,, \tag{3.57}$$

where $\eta_i$ is some time at which we specify the initial conditions at the start of inflation and $\eta_0$ is the reheating surface at the end.

Importantly, for a correlator to be invariant under a transformation, we require several different properties of the theory and background in which it lives. To illustrate this, let us consider the action of each

of the symmetries individually. For **CRT** we find

$$\textbf{CRT} : B_n(\eta; \mathbf{y}_1 \dots \mathbf{y}_n) \rightarrow B_n^*(-\eta; -\mathbf{y}_1, \dots, -\mathbf{y}_n) \tag{3.58}$$

$$= \textbf{CRT}(\langle\Omega(\eta_{\text{i}})|)U^\dagger(\eta_{\text{i}}, \eta)\phi_1^*(-\eta; -\mathbf{y}_1) \dots \phi_n^*(-\eta; -\mathbf{y}_n)U(\eta_{\text{i}}, \eta)\textbf{CRT}(|\Omega(\eta_{\text{i}})\rangle), \tag{3.59}$$

where we have used the fact that the time evolution operator defined in (3.55) is invariant under **CRT**. Thus we recover our original correlator if:

- The state is **CRT**-invariant, $\textbf{CRT}(|\Omega(\eta_{\text{i}})\rangle) = |\Omega(\eta_i)\rangle$, which is true for the Bunch-Davies vacuum in which the in-in correlators are usually defined;

- The theory is **CRT**-invariant, $\textbf{CRT} : H \rightarrow H$ which ensures that $U(\eta_i, \eta)$ is also **CRT** 0invariant;

- The fields are **CRT**-invariant, $\textbf{CRT}(\phi(\eta; \mathbf{y})) \equiv \phi^*(-\eta; -\mathbf{y}) = \phi(\eta; \mathbf{y})$. This is actually ensured by the fact that their equation of motion and the initial Bunch-Davies vacuum state is **CRT**-invariant. However we will see in Section 7.4 that at the boundary the **CRT** transformation acts more non-trivially on the fields.

For the Dilatation operator, we will use the intuition from Section 3.2 to assume we can access $\lambda = -1$ despite the fact that it usually defined with $\lambda \in \mathbb{R} \geq 0$ and then address this analytic continuation in more detail in Section 6,

$$\textbf{D}_\lambda : B_n(\eta; \mathbf{y}_1, \dots, \mathbf{y}_n) \rightarrow B_n(\lambda\eta; \lambda\mathbf{y}_1, \dots, \lambda\mathbf{y}_n) \tag{3.60}$$

$$= \langle\Omega(\lambda\eta_{\text{i}})|\, U^\dagger(\lambda\eta_{\text{i}}, \lambda\eta)\phi_1(\lambda\eta; \lambda\mathbf{y}_1) \dots \phi_n(\lambda\eta; \lambda\mathbf{y}_n)U(\lambda\eta_{\text{i}}, \lambda\eta)\,|\Omega(\lambda\eta_{\text{i}})\rangle. \tag{3.61}$$

Thus we recover our original correlator if:

- The state is scale-invariant, $|\Omega(\lambda\eta_{\text{i}})\rangle = |\Omega(\eta_i)\rangle$, which is true for the Bunch-Davies vacuum in which the in-in correlators are usually defined, since the Bunch-Davies vacuum (as well as any other $\alpha$-vacua) is invariant under all the de Sitter isometries;

- The theory is scale-invariant, $\textbf{D}_\lambda : H \rightarrow H$ which ensures that $U(\lambda\eta_i, \lambda\eta) = U(\eta_i, \eta)$;

- The fields are scale-invariant, $\textbf{D}_\lambda(\phi(\eta; \mathbf{y})) \equiv \phi(\lambda\eta; \lambda\mathbf{y}) = \phi(\eta; \mathbf{y})$. This is actually ensured by their scaling properties, if they were not scale invariant themselves then this transformation would also involve transforming the fields as we will see in Section 7.4.

We can then use **CRT** and $\textbf{D}_{-1}$ to infer **RR**, and thus find in analogy with Sections 2.4 and 2.5, that **RR** constrains $B_n$ to be real by ensuring that the time evolution operator is unitary, $U^\dagger U = \mathbb{1}$. Hence we are justified in labelling this property and its associated transformation **RR**.

We are now equipped to construct a symmetry group from the action of these transformations,

$$\textbf{CRT} : \quad B_n(\eta; \mathbf{y}) \quad = \quad B_n^*(-\eta; -\mathbf{y}); \tag{3.62}$$

$$\textbf{D}_{-1}^\pm : \quad B_n(\eta; \mathbf{y}) \quad = \quad B_n(-\eta; -\mathbf{y}); \tag{3.63}$$

$$\textbf{RR} : \quad B_n(\eta; \mathbf{y}) \quad = \quad B_n^*(\eta; \mathbf{y}), \tag{3.64}$$

Just as in flat space these three transformations form a closed group and therefore, any two of them imply the last.

Thus we are not asserting that these are guaranteed to be symmetries of an inflationary theory, just that one of them will come for free if the other two are satisfied. Similar to how a theory can be invariant under

the discrete 180° rotation $\mathbf{R}_\pi$ but not invariant under $SO^+(1,1)$ flat space, it is relatively easy to come up with theories in de Sitter that accidentally satisfy this discrete ($\lambda = -1$) scaling transformation without actually respecting continuous scale transformations ($\lambda > 0$). For example, consider the interaction $ga^2\phi^4$ in $D = 3 + 1$-spacetime dimensions, where $g$ is a real coupling. This is clearly not scale invariant (as a scale-invariant non-derivative interaction would contain four powers of $a$ rather than two) but is both Unitary (since it has a real coupling) and **CRT** invariant as it does have Discrete Scale Invariance due to the even number of extra scale factors.

The focus here has been on equal time correlators due to their importance to cosmology. However, Unitarity also has implications for unequal time correlators. In this case the result is almost identical to the flat space results expressed in Lorentzian signature:

$$\mathbf{CRT} : \quad \mathcal{T}B_n(\eta; \mathbf{y}) \quad = \quad \overline{\mathcal{T}}B_n^*(-\eta; -\mathbf{y}) \, ; \tag{3.65}$$

$$\mathbf{D}_{-1}^{\pm} : \quad \mathcal{T}B_n(\eta; \mathbf{y}) \quad = \quad \mathcal{T}B_n(-\eta; -\mathbf{y}) \, ; \tag{3.66}$$

$$\mathbf{RR} : \quad \mathcal{T}B_n(\eta; \mathbf{y}) \quad = \quad \overline{\mathcal{T}}B_n^*(\eta; \mathbf{y}) \, , \tag{3.67}$$

where the bar indicates that time ordering has been exchanged with anti-time ordering. The main difference between flat space and here being that it is all the spatial coordinates that are inverted rather than just one. This is the case in any dimension due to the way the global de Sitter transformation is projected onto the Poincaré patch.

## 3.5 Wavefunction of the Universe

The same cosmological correlation functions discussed in Sections 3.3 and 3.4 can be calculated by using a basis of $\phi$ field eigenstates for both the bra and the ket:

$$\langle \mathcal{O}(\eta) \rangle = \int \mathrm{D}\phi \mathrm{D}\tilde{\phi} \langle \Psi(\eta) | \phi; \eta \rangle \langle \phi; \eta | \mathcal{O}(\eta) | \tilde{\phi}; \eta \rangle \langle \tilde{\phi}; \eta | \Psi(\eta) \rangle \tag{3.68}$$

$$= \int \mathrm{D}\phi \mathrm{D}\tilde{\phi} \Psi^*[\phi; \eta] \Psi[\tilde{\phi}; \eta] \langle \phi; \eta | \mathcal{O}(\eta) | \tilde{\phi}, \eta \rangle \tag{3.69}$$

$$= \int \mathrm{D}\phi \Psi^*[\phi; \eta] \Psi[\phi; \eta] \mathcal{O}(\phi; \eta) \, , \tag{3.70}$$

where $\mathcal{O}(\phi; \eta)$ is obtained by replacing instances of $\hat{\phi}$ with the corresponding eigenvalue, $\phi$, of the state $|\phi; \eta\rangle$, and the Wavefunction of the Universe (WFU) is defined as [28]

$$\Psi[\phi; \eta] \equiv \langle \phi; \eta | \Psi(\eta) \rangle = \langle \phi; \eta | \Omega \rangle \, . \tag{3.71}$$

From (3.71) it alreay becomes apparent that the same analysis used for the cosmological correlators in Section 3.4 will not apply to the WFU, since the WFU is not an overlap of two vacuum states $|\Omega\rangle$. We will explain how to deal with this problem in Section 4.

The WFU satisfies the Schrödinger equation,

$$i\hbar \frac{\mathrm{d}}{\mathrm{d}\eta} \Psi[\phi; \eta] - H(\phi(\eta; \mathbf{x}); \eta)\Psi[\phi; \eta] = \mathcal{D}\Psi[\phi; \eta] = 0 \, , \tag{3.72}$$

subject to the Bunch-Davies initial condition and can be parameterised as

$$\Psi[\phi; \eta] \equiv \langle \phi; \eta | \Omega \rangle = \exp \left\{ -\sum_{n=2}^{\infty} \int \prod_{a=1}^{n} \frac{\mathrm{d}^d k_a}{(2\pi)^d} \phi_{\mathbf{k}_a} \psi_n(\eta; \mathbf{k}_1, \ldots, \mathbf{k}_n) \delta^d \left( \sum \mathbf{k}_a \right) \right\} \, . \tag{3.73}$$

---

[28]See Appendix A of [13] for a pedagogical review of the WFU approach to the computation of cosmological correlators. This closely follow Weinberg's discussion of Path-Integral methods in [95] and an unpublished manuscript from Garrett Goon.

Despite not being directly observable themselves, it has become customary to discuss the role of symmetries and fundamental principles on the terms in this expansion, $\psi_n$, the so called coefficients of the WFU. The relations between the cosmological correlators $B_n$ and $\psi_n$ can be found in various parts of the literature, see e.g. [17,34,96]. In particular we have developed our understanding of Unitarity within the cosmological context in terms of its implications on these wavefunction coefficients. This is due to the discovery of the Cosmological Optical Theorem [13–15] where Unitarity of the time evolution operator was used to derive a constraint on contact wavefunction coefficients (Feynman-Witten amplitudes with only external lines):

$$\psi_n(\eta_0; |\mathbf{k}|, \mathbf{k}) + \psi_n^*(\eta_0; -|\mathbf{k}|, -\mathbf{k}) = 0 \,, \tag{3.74}$$

where this Unitarity constraint involves an analytic continuation of the magnitude of the *external* momenta (we will sometimes refer to the magnitudes as "energies") from the positive real axis through the lower half of the complex plane. However, for more complicated diagrams with internal lines the right hand side picks up a combination of lower point functions, i.e. in analogy with the Cutkosky cutting rules in flat space Unitarity in cosmology relates different orders in perturbation theory [15,97]. It was later shown in [54] that if one also analytically continues the *internal* energies (3.74) would hold for all tree-level contact and exchange diagrams with some caveats for certain types of *non-local* interactions. Unfortunately, these results are understood only perturbatively [15,97], whereas the symmetry perspective that we present here is valid non-perturbatively and, as such provides new insights on these objects.

By considering the *local Lagrangian symmetries*, in Section 4, we will show that a similar relationship holds for diagrams of arbitrary loop order when we analytically continue *both external and internal energies.* (Although we do not, in this paper, consider cutting rules, only constraints on individual Feynman-Witten diagrams.) This also provides insight into why certain types of non-local interactions may not satisfy (3.74). Through analogy with **RR** in flat space, we will interpret this as a limited consequence of Unitarity.

# 4 Local Lagrangian Symmetries

In this section we describe the notion of a local Lagrangian symmetry, and explain why this type of symmetry is needed to obtain a useful constraint on wavefunction coefficients in a single Poincaré patch.

## 4.1 Distinction from Spacetime Symmetries

In order to proceed, we must now make a distinction between two classes of symmetries:

1. A *spacetime symmetry*, which acts on the entire manifold $\mathcal{M}$ by moving points around by means of some map $m : x \to y$. (In addition to some local transformation of the fields $\phi(x)$, and possibly a complex conjugation of the amplitude.)

2. A *local Lagrangian symmetry*, which is an assertion that the Lagrangian $L$ at a point $p \in \mathcal{M}$ respects some symmetry. Here, the symmetry acts on the tangent space $T_p(\mathcal{M})$ (in addition to transforming the fields $\phi(p)$ and possibly complex conjugating).

The relation between these kinds of symmetry is somewhat subtle.

A spacetime symmetry can sometimes give rise to a local Lagrangian symmetry. This happens whenever there is a *fixed point* of the map $m$, that is a point $p$ for which $m : p \to p$. In this case there will be an induced map on the tangent space $T_p$, and this will normally[29] give rise to a symmetry of the local Lagrangian.

---

[29]The additional assumptions required are that the theory *has* a local Lagrangian, and somewhat more restrictively we are assuming that the symmetry preserves the Lagrangian and not just the action.

As translations in Minkowksi do not have fixed points, they do not give rise to any local Lagrangian symmetry by themselves. But they do, of course, ensure that if there is a local Lagrangian symmetry at one point $p$, it can be translated to any other point $q$.

There can also exist local Lagrangian symmetries which do not arise from any spacetime symmetry of $\mathcal{M}$. A classic example of this, is the standard assumption that a field theory has local Lorentz Invariance at each point $p$, even on a spacetime manifold which has no Killing symmetry. (This example also shows that the definition of "local" has nothing to do with gauge invariance, as we typically assume local Lorentz Invariance even when doing field theory on a fixed, non-gravitational background.) Another simple example would be a time-varying Hamiltonian $H(t)$, in a case where at each moment of time $t$ the Hamiltonian is time-reversal invariant, but there exists no time $t_0$ for which $H(t_0 + t) = H(t_0 - t)$.

Let us now consider the implications for the discrete symmetries in our $\mathbb{Z}_2 \times \mathbb{Z}_2$ group. In the case of Reflection Reality (**RR**), the fixed points include the entire *real* part of the Lorentzian manifold $\mathcal{M}$. Hence, if we are working in Lorentzian signature, we are automatically entitled to extend **RR** to a symmetry of the local Lagrangian at each point $p$.

The same is not true of an individual **CRT** symmetry, since in this case $t \to -t$ and not all points are fixed by a given **CRT**. But in a manifold with time translation symmetry (e.g. Minkowski) we can always translate the story to find *a* **CRT** symmetry at any given point. Alternatively, if we have local Lorentz Invariance, this can be Wick rotated to obtain a local 180° rotation, and then **RR** (which is always local by the argument above) implies local **CRT**.

## 4.2 Flat Space Wavefunction

Below, we will be interested in using discrete symmetries to constrain vacuum wavefunction coefficients. But, as we have alluded to, it will turn out that the local Lagrangian versions of this symmetry are much more useful than the spacetime versions.

Let us consider the spacetime version of the $\mathbb{Z}_2 \times \mathbb{Z}_2$ group, starting in flat spacetime. Recall that the vacuum wavefunction $\Psi$ at $\tau = it = 0$, is given by a path integral in the lower half of Euclidean space, that is it is a one-sided path integral over a spacetime with $\tau < 0$:

$$\Psi[\phi(x)] \equiv \langle \phi(x)|\Psi \rangle = \int_{\tau < 0} \mathrm{D}\phi \, e^{-I[\phi(x,\tau)]} \tag{4.1}$$

(This might be followed by some amount of Lorentzian time evolution, which will not affect the basic points below.)

There is no problem with identifying the constraint that **CRT** places on the flat space wavefunction. Assuming $\phi$ is real, and at fixed $t$, it is simply:

$$\mathbf{CRT} : \Psi[\phi(x)] = \Psi^*[\phi(-x)] \tag{4.2}$$

where $x$ is the single spatial coordinate that gets flipped. Nor does it matter much if we think of this as the spacetime or the local form of **CRT**, as long as it holds fixed the $t$-slice on which the wavefunction is evaluated. (Assuming time translation symmetry, $\Psi$ is independent of $t$.)

But what is the implication for **RR** on the wavefunction? Naively, there isn't one. **RR** tells you that correlators satisfy hermitian conjugation (2.18):

$$\mathbf{RR} : \langle \Psi| \, \phi_1 \, \phi_2 \ldots \phi_n \, |\Psi \rangle = \langle \Psi| \, \phi_n \ldots \phi_2 \, \phi_1 \, |\Psi \rangle^*, \tag{4.3}$$

but this holds for *every* possible value of $\Psi$ (that lives in a state space which admits a real inner product) regardless of the functional form of $\Psi[\phi]$. What went wrong? The problem is that the one-sided path

integral is done over negative $\tau$ values only. Hence, a spacetime symmetry which flips $\tau \to -\tau$ does not place a meaningful constraint on $\Psi$. Because it reverses the order of operators, it maps a ket state $|\Psi\rangle$ to a bra state $\langle\Psi|$, rather than placing any constraint on the ket itself.

A similar problem occurs for the 180° rotation, assuming we attempt to apply it in isolation and not as part of a bigger group of symmetries.[30]

The story above also applies, without any essential change, to global de Sitter where **CRT** allows us to write a concise non-perturbative constraint for the full de Sitter wavefunction at equal-$T$:

$$\Psi[T; \phi(\Phi_d)] = \mathbf{CRT}(\Psi[T; \phi(\Phi_d)]) \equiv \Psi^*[-T; \phi(\pi - \Phi_d)], \tag{4.4}$$

where we have used the fact that (as discussed in Section 3.1) for equal-time quantities, such as the de Sitter wavefunction, **CRT** simplifies to a complex conjugation, a flip in the global time coordinate and a rotation on the $d-1$-sphere. Note that at $T = 0$, **CRT** acts to constrain the wavefunction to be its own conjugate after a spatial reflection.

## 4.3 Poincaré Wavefunction of the Universe

In Poincaré de Sitter, things naively get even worse. As the Poincaré patch explicitly breaks time-reversal invariance, it seems that we can't even constrain the wavefunction using **CRT**. This is because the standard Poincaré **CRT** (described in Section 3.2) maps the future patch $P^+$ to the past patch $P^-$. But then it seems to place no constraints on $P^+$ considered in isolation.

However, this is too quick. In fact, as we shall see, discrete symmetries can place quite strong constraints on the wavefunction $\Psi$. This is so for two basic reasons:

- We can use the local Lagrangian form of discrete symmetries such as **CRT** and **RR**, and

- We can make use of the fact that $\Psi$, the Bunch Davies state, satisfies various analytic continuation properties.

Regarding the first point, let us consider a much more powerful form of local Lagrangian **CRT**:

**LL CRT**: For every point $p \in P^+$, there exists a **CRT** symmetry of its tangent space.

This is a much more powerful principle. In fact though, we can obtain it from Poincaré **CRT** if we also assume invariance under the full de Sitter group.

$$\text{Poincaré } \mathbf{CRT} + SO^+(d+1, 1) \implies \mathbf{LL\ CRT}.$$

Conversely, *if* we stipulate that all local Lagrangian symmetries must arise from spacetime symmetries, and also spatial translations, we find that the full de Sitter group is implied:

$$\mathbf{LL\ CRT} \text{ from spacetime + spatial translations} \implies SO^+(d+1, 1)$$

because the conjugation of translations by the local **CRT** symmetries can get you special conformals, and from there one generates the whole group. Of course, the Poincaré version of **CRT** is an easy corollary.

We therefore find, that in the context of homogeneous cosmology, local Lagrangian **CRT** can arise from spacetime symmetries only in the case of de Sitter. These wavefunction constraints will be considered in Sections 6.1–6.2 and (in more specific detail) in Section 7.

---

[30]If we have both **RR** and the 180°, we can obviously compose them to get **CRT** which then takes kets to kets again. Or, if we impose the full continuous Lorentz Invariance, we can check if $\Psi$ satisfies properties associated with Lorentz Invariance such as the Unruh effect.

On the other hand, local Lagrangian **RR**, which does *not* reflect a Lorentzian time direction, can be meaningfully applied in any real FLRW cosmology, regardless of the existence of any Killing symmetry. In the case where space is flat, we can use this to prove a more general version of the Cosmological Optical Theorem (in Section 6.3).

But before this, in the next Section 5, we will examine the close relationship between the local Lagrangian version of **RR**, and Unitarity in Lorentzian signature.

# 5 Accidental Unitarity

While **RR** is weaker than full Unitarity, it turns out it gets you a lot closer to full Unitarity than you might think! In this section, we will argue that, among the set of QFTs satisfying **RR**, a codimension 0 subset are also fully Unitary.

## 5.1 Perturbative Equivalence of Reflection Reality and Unitarity

Suppose we start with a unitary QFT, and consider deforming it by some small perturbation $\delta L$ to the Lagrangian density $L$.[31] Let us assume that this small perturbation does not change the number of configuration degrees of freedom in the theory.[32] It should then be possible to view it as a change to the Hamiltonian $\delta H$. If we consider all possible terms in the Hamiltonian (satisfying some set of symmetries) up to some degree of irrelevance, this should give us a finite-dimensional space of perturbations to the couplings, so $H \in \mathbb{C}^N$ for some finite dimension $N$. Now the subspace of perturbations that preserve Unitarity should be those satisfying:

$$\delta H = \delta H^\dagger \,, \tag{5.1}$$

where $\dagger$ is defined using the positive norm of the original theory's state space. This can be regarded as imposing a real structure on the space of couplings, i.e. it restricts you to a real subspace $\delta H \in \mathbb{R}^N \subset \mathbb{C}^N$ such that if a vector $v$ is in the space $\mathbb{R}^N$, $iv$ is not in the $\mathbb{R}^N$. In other words all of its basis vectors should be (complex)-linearly independent. Since Unitary implies Reflection Positivity and hence Reflection Reality, for any $v \in \mathbb{R}^N$, $\mathbf{RR}(v) = v$. Since **RR** is antilinear this means that $\mathbf{RR}(iv) = -iv$. We now know how **RR** acts on the entire space, and it is clear that the space of **RR** symmetric perturbations is no larger than the space of unitary ones. Hence, for small perturbations of this sort, **RR** and Unitarity are equivalent conditions.

For a bosonic field theory in Lorentzian signature, such perturbations are in one-to-one correspondence with *real* contributions to $\delta L$.[33] If however, we consider perturbations that involve derivative couplings, in Lorentzian signature there can also be (typically divergent) measure factors required to restore Unitarity, that are tantamount to fixing certain imaginary terms in the action. However, upon passing to Euclidean signature, such terms are fixed by **RR** alone, without using any additional feature of Unitarity. This is one way in which Euclidean field theory is nicer than Lorentzian field theory.

---

[31]Even if a QFT does not come from a Lagrangian, perturbations to the theory still do.

[32]If the perturbation involves higher derivative couplings, it may be necessary to assume it comes with powers of the UV cutoff, to prevent additional degrees of freedom from appearing. Throughout this section, we are working in the regime of naive QFT perturbation theory, where we assume the Hilbert space is independent of the choice of coupling constants. This is not really true, but we expect that the conclusions of this section would not be affected by a more sophisticated analysis.

[33]There is, indeed, one class of imaginary contributions to $\delta L$ that are compatible with Unitarity, namely imaginary total derivatives. These correspond to imaginary canonical transformations of the state space, which can be viewed as rescaling the definition of the inner product. But because the action appears inside of an exponential, such rescalings are multiplicative and thus cannot spoil Unitarity.

## 5.2 How to get Theories with Negative Norm States

Does this mean that all **RR** satisfying theories are also unitary? Clearly not, since there exist theories with negative norm states. But to get them you have to break one of the assumptions in the argument above. Since the argument above refers only to *continuous* variations of the Lagrangian, one way to obtain such theories is to make a bad *discrete* choice of field content, which is incompatible with Unitarity. One easy example of this is a negative-norm scalar field. Here we start with a field with negative propagator term, e.g:

$$L = -\tfrac{1}{2}\partial_\mu\phi\partial^\mu\phi + V(\phi)\,. \tag{5.2}$$

Although this action looks unbounded below in Euclidean signature, this problem can be resolved if we define a new field $\chi = i\phi$ and do our functional integrals along the real $\chi$ direction. But then $\chi^\dagger = -\chi$, so any terms in $\hat{V}(\chi) = V(i\phi)$ with odd powers of $\chi$ will have an unexpected imaginary sign, relative to the requirements of Unitarity. This is because the states with an odd number of $\chi$ quanta have negative norm.[34] This is an example of a bad discrete choice which leads to a **RR** but not Unitary theory. Another example of such a bad choice is a spin-statistics violating theory, e.g:

$$L = i\partial_\mu\psi\partial^\mu\overline{\psi}\,, \tag{5.3}$$

where $\psi$ and $\overline{\psi}$ are scalar fermions. In this case the 1-particle state space has indefinite signature and there is no way to extend **RR** to full Unitarity.

Finally we can also consider choices which change the number of degrees of freedom. An illuminating example is the Proca Lagrangian:

$$L = \tfrac{1}{4}F_{\mu\nu}F^{\mu\nu} + \tfrac{1}{2}m^2 A_\mu A^\mu\,. \tag{5.4}$$

In $D$ spacetime dimensions, the number of propagating degrees of freedom is $D-1$ when $m^2 \neq 0$, but only $D-2$ when $m^2 = 0$. The latter is, of course, due to the usual gauge symmetry of a massless photon field. Since the gauge theory has a bunch of null sates, which have 0 inner product with any other vector, it can be a boundary between a unitary theory, and a theory containing negative norm states. And in fact, in the tachyonic case where $m^2 < 0$, there is a propagating temporal mode which has negative norm. If then, you have a unitary theory with a gauge symmetry, and you want to deform it to another unitary theory, the safest thing to do is to restrict attention to only gauge-invariant perturbations to the Lagrangian.

Although the above counterexamples show that **RR** does not imply Unitarity, it is encouraging that we had to work hard to find such counterexamples—generic perturbations to a Lagrangian cannot do the trick. Put another way, once we impose **RR**, we have a decent chance of accidentally ending up with a unitary theory by pure luck. There is not going to be any need to fine-tune further parameters with infinite precision to end up in the Unitary case.

## 5.3 Unitarity from CRT

It is similarly possible to argue that Unitarity can accidentally follow from a local **CRT** invariance of the action.

If we restrict to locally Lorentz invariant bulk Lagrangians, all such covariant terms have a 180° rotation symmetry,[35] and hence **CRT** implies **RR** implies (for small perturbations as above) Unitarity. Thus **CRT** is also sufficient to accidentally imply Unitarity, in the absence of bad discrete choices for the field content.

---

[34]If $V$ is an even function, there exists an additional $\mathbb{Z}_2$ symmetry $\chi \to -\chi$, and the inner product can be redefined in a way that restores Unitarity. But this amounts to redefining **RR** using a different $\mathbb{Z}_2$ generator.

[35]An exception might occur in theories where Lorentz Invariance is spontaneously broken, e.g. if the action is defined relative to a unit timelike vector field $u^a$ as in Einstein-Aether theory [98, 99]. But even these theories can sometimes be accidentally unitary, so long as all the terms in the action are even powers of $u^a$, the 180° rotation symmetry remains.

It should be noted that the above argument refers to *local* **CRT** invariance of the Lagrangian $L$ at *each* point. In Section 3, we discussed a global **CRT** transformation that flips a future Poincaré-dS to the past one. This global **CRT** symmetry will not, by itself, suffice to prove local **CRT** invariance of the Lagrangian, as it only flips **CRT** around one particular static patch bifurcation surface. But, global **CRT** together with the full rotational symmetry $SO^+(d+1,1)$ implies local **CRT** at any point.

This helps to motivate the crucial importance of **CRT** in de Sitter as a check on holographic ideas. Suppose somebody hands you a claimed instance of dS/CFT and you have already checked that it is conformally invariant. But you want to know if it is dual to a bulk unitary theory. According to this argument, checking **CRT** invariance gets you nearly all the way to deciding Unitarity. As long as the low energy bulk fields don't violate spin-statistics or have negative-norm states, it is then reasonably likely that you have a Unitary theory on your hands. On the other hand, if you slap together some partition function without paying any regard to discrete symmetries, it would be infinitely unlikely that you will end up with a Unitary bulk theory. We will now explore how the symmetries described in Sections 3-4 impose constraints on the coefficients of the Wavefunction of the Universe.

# 6   Analytic Continuation in Cosmology

Similar to how the physical domain of Mandelstam variables in amplitudes is defined for $s, t \geq 0$ and crossing symmetry involves analytically continuing to $s, t \in \mathbb{C}$, we will now discuss how our discrete transformations in Section 3.2 will involve analytically continuing the conformal time outside of the physical domain of $\eta \leq 0$ and in Fourier space will also involve analytically continuing the magnitude of the momenta $k = \sqrt{\mathbf{k} \cdot \mathbf{k}} \geq 0$.

## 6.1   Nonperturbative Analytic Continuation

To perform the analytic continuation explained in Section 3.2 more carefully, let us do QFT on a fixed spacetime Poincaré patch metric.[36] We take the metric to have the slightly more general form:

$$\mathrm{d}s^2 = (A \, \mathrm{d}\zeta)^2 + (B \, e^{H\zeta} \, \mathrm{d}y_i)^2 = A^2 \, \mathrm{d}\zeta^2 + B^2 e^{2H\zeta} \, \mathrm{d}y_i \, \mathrm{d}y_i \,, \tag{6.1}$$

where $A$ and $B$ are constants. It should be noted that because a typical matter action has a factor of $\sqrt{g}$ out front, the sign of $A$ and $B$ matters, not just $A^2$ and $B^2$. Some special cases of this metric are:

- $A = B = +1$: Normal Euclidean hyperbolic space $H^0$, with $\zeta = \tau$;

- $A = +i$, $B = +1$: Future dS-Poincaré $P^+$, with $\zeta = t$;

- $A = -i$, $B = +1$: Past dS-Poincaré $P^-$, with $\zeta = -t$;

- $A = B = +i$: Funky Euclidean hyperbolic space $H^+$ with all minus signature.

The funky hyperbolic space $H^+$ does not in general have the same properties as a normal field theory: massive fields become tachyons, and (depending on the theory) it need not satisfy Reflection Reality.

---

[36]Note as we previously explained that in global de Sitter there is no need for an analytic continuation as the **CRT** transformation is well-defined within those coordinates. AT thanks Carlos Duaso Pueya and Ciaran McCulloch for fruitful discussions regarding analytic continuations in cosmology.

**Correct path via Wick rotation of $g_{tt}$:** From this it can be seen that a *correct* way to analytically continue from $P^+$ to $P^-$ is to hold $B$ fixed and take $A$ along the path:

$$A = i^{1-2q} = e^{(1-2q)i\pi/2}, \qquad q \in [0,1] \tag{6.2}$$

where $q = 0$ corresponds to $H^+$ and $q = 1$ corresponds to $H^-$. The halfway point, $q = \frac{1}{2}$, corresponds to normal Euclidean hyperbolic space $H_0$. The advantage of this path is that a well-behaved QFT (e.g. an arbitrary $p$-form field) has an action whose real part is bounded below everywhere in the interior of the $q$-interval. As usual the Lorentzian path integral, being oscillatory, is on the edge of convergence. (If we define the initial condition by taking $A$ to have a small positive part at early times, we obtain the Bunch-Davies state; hence, the initial condition is independent of the choice of $q \in [0,1]$.)

**Geometric path via $\eta$ continuation:** However, the correct path does not have any obvious geometric interpretation in terms of analytically continuing the time coordinate; that is, the path does not lie on the complexification of $P^+$. If we choose instead to rotate the $\eta$ coordinate, this corresponds to starting in $H^+$ and shifting $Ht \to Ht + i\pi$. This time we are holding $A$ fixed and taking $B$ along the path:

$$B = i^{2q} = e^{i\pi q}, \qquad q \in [0,1] \tag{6.3}$$

At $q = \frac{1}{2}$, this contour passes through funky hyperbolic space $H^+$ where the QFT need not converge. Worse still, at $q = 1$, it ends up at $A = +i$, $B = -1$ which is not the same as $P^-$!

However, the two paths are very similar. For each value of $q$, they differ only by an overall rescaling of the metric. That is, the geometrical path can be converted back into the correct path if, for each value of $q \in [0,1]$, we additionally rotate the metric by

$$g_{ab} \to \Omega^2 g_{ab}, \qquad \Omega = (-i)^{2q} = e^{-i\pi q} \tag{6.4}$$

This looks very similar to an imaginary Weyl transformation, although we do not rescale any of the matter fields. At $q = \frac{1}{2}$, this puts us back to normal hyperbolic space $H^0$, and at $q = 1$, it yields $P^-$, after applying $\Omega = -1$. Then, $\Omega^2 = +1$ and we have simply rotated the metric $g_{ab}$ a full cycle in the complex plane. However, in general the amplitudes become singular as $g_{ab} \to 0$, and hence we have rotated around a singularity. So we are on a new Riemann sheet, and we do not necessarily get the same physics as if we had not done the Weyl rescaling!

If we write our amplitudes as an explicit function of $\Omega$, we can thus conclude that we can continue from Bunch-Davies in $P^+$ (evolving forwards in time) to Bunch-Davies in $P^-$ (evolving backwards in time) by doing the transformation:

$$\mathcal{A}^-(\eta; y_1, \ldots, y_d; \Omega) = \mathcal{A}^+(e^{-i\pi}\eta; y_1, \ldots, y_d; e^{-i\pi}\Omega), \tag{6.5}$$

where in this expression, the use of $e^{i\pi}$ rather than $-1$ is a mnemonic indicating in which direction one should rotate in the complex plane to reach $-1$ (all such rotations are to be done simultaneously, in this case holding the ratio of $\eta$ and $\Omega$ fixed). But $P^+$ and $P^-$ are also related by **CRT** symmetry, which tells us that $P^+$ and $P^-$ are also related by sending $y_i \to -y_i$ and complex conjugating the amplitude. Hence, invariance under **CRT** also requires $P^+$ and $P^-$ to also be related by the transformation:

$$\mathcal{A}^-(\eta; y_1, \ldots, y_d; \Omega) = \mathcal{A}^+(-\eta; -y_1, \ldots, -y_d; \Omega)^*. \tag{6.6}$$

where a minus sign that is written out explicitly, represents a mere reflection of the coordinate (no analytic continuation). Combining these together, it follows that, in a **CRT** invariant theory, the wavefunction coefficients in $P^+$ must obey:

$$\mathbf{CRT} : \psi_n^*(\eta; y_1, \ldots, y_d; \Omega) = \psi_n(e^{-i\pi}\eta; -y_1, \ldots, -y_d; e^{i\pi}\Omega). \tag{6.7}$$

In momentum space we would have almost the same relation:

$$\mathbf{CRT} : \psi_n^*(\eta; k_1, \ldots, k_d; \Omega) = \psi_n(e^{-i\pi}\eta; k_1, \ldots, k_d; e^{-i\pi}\Omega) \,, \tag{6.8}$$

apart from the fact that complex conjugation in position space reverses the sign of $k$, cancelling out the reflection of each spatial coordinate.

**Geometric path via $y$ continuation:** There is another way to achieve a similar result, by analytically continuing the $y$ variables rather than the $\eta$ variable. If we analytically continue $y_i \to e^{-i\pi q}y_i$ (for all of the $y$ coordinates at once, and for $q \in [0,1]$ as before), then to hold the metric fixed, we end up doing the exact same rotation of $B$ as in (6.3). As before, to convert this to the correct contour for the $A$ parameter, we will need to rotate $\Omega$ to obtain equivalence to the correct path. Also, as $\eta$ keeps the same sign, we will need to invoke some discrete symmetry which remains within $P^+$. The obvious choice here is $\mathbf{RR}$.

The simplest way to write down a correct constraint on the wavefunction coefficient is to start with **CRT** in the form (6.7), and also act with a 180° rotation. As the latter is an analytic continuation of the $SO^+(1,1)$ scaling symmetry, it takes the form:

$$\mathbf{D}_{-1}^{\pm} : \psi_n(\eta; y_1, \ldots, y_d; \Omega) = \psi_n(e^{\pm i\pi}\eta; e^{\pm i\pi}y_1, \ldots, e^{\pm i\pi}y_d; \Omega) \,, \tag{6.9}$$

where the $\pm$ sign depends on the direction of the 180° rotation. When considering wavefunction coefficients, it is no longer an *a priori* truth that the 360° rotation is a trivial transformation. This enhances the symmetry group to $\mathrm{Aut}(\mathbb{Z})$, see the discussion surrounding Table 2 in Section 7.2. Hence, the combined statement (which should follow from Reflection Reality) implies the following result for the $y$-continued wavefunction coefficient:

$$\mathbf{RR} : \psi_n^*(\eta; y_1, \ldots, y_d; \Omega) = \psi_n(\eta; -e^{i\pi}y_1, \ldots, -e^{i\pi}y_d; e^{-i\pi}\Omega) \,, \tag{6.10}$$

where, similarly to previous expressions, multiplication by $-e^{i\pi}$ is shorthand for an analytic continuation followed by a reflection, and cannot validly be replaced with $+1$!

In momentum space, these relations become:

$$\mathbf{D}_{-1}^{\pm} : \psi_n(\eta; k_1, \ldots, k_d; \Omega) = (-1)^{\pm d}\psi_n(e^{\pm i\pi}\eta; e^{\mp i\pi}k_1, \ldots, e^{\mp i\pi}k_d; \Omega) \,, \tag{6.11}$$

and

$$\mathbf{RR} : \psi_n^*(\eta; k_1, \ldots, k_d; \Omega) = (-1)^d\psi_n(\eta; e^{-i\pi}k_1, \ldots, e^{-i\pi}k_d; e^{i\pi}\Omega) \,. \tag{6.12}$$

where the prefactor of $(-1)^d$ arises because of the scaling of the overall delta function $\delta^d\left(\sum_a^n \mathbf{k}_a\right)$ due to momentum conservation, which transforms like $k^{-d}$, and which we (purely conventionally) exclude from our definition of $\psi_n$. (We would not need to explicitly include this factor if we were manipulating $\psi_n'$, which is defined to include the delta function.) In cases involving fractional dimensions (e.g. dim. reg.), this prefactor becomes $e^{i\pi d}$ in the case of $\mathbf{RR}$ and $e^{\pm i\pi d}$ in the case of $\mathbf{D}_{-1}^{\pm}$, transforming in the opposite direction as the $k$ argument.

To summarise, we obtain the following non-perturbative results for the wavefunction coefficients in position space:

$$\mathbf{CRT} : \psi_n^*(\eta; y_1, \ldots, y_d; \Omega) = \quad \psi_n(e^{-i\pi}\eta; -y_1, \ldots, -y_d; e^{-i\pi}\Omega) \,; \tag{6.13}$$

$$\mathbf{D}_{-1}^{\pm} : \psi_n(\eta; y_1, \ldots, y_d; \Omega) = \quad \psi_n(e^{\pm i\pi}\eta; e^{\pm i\pi}y_1, \ldots, e^{-\pm i\pi}y_d; \Omega) \,; \tag{6.14}$$

$$\mathbf{RR} : \psi_n^*(\eta; y_1, \ldots, y_d; \Omega) = \quad \psi_n(\eta; -e^{i\pi}y_1, \ldots, -e^{i\pi}y_d; e^{-i\pi}\Omega) \,. \tag{6.15}$$

and in momentum space:

$$\mathbf{CRT}: \psi_n^*(\eta; k_1, \ldots, k_d; \Omega) = \psi_n(e^{-i\pi}\eta; k_1, \ldots, k_d; e^{-i\pi}\Omega) \, ; \tag{6.16}$$

$$\mathbf{D}_{-1}^{\pm}: \psi_n(\eta; k_1, \ldots, k_d; \Omega) = e^{\pm i\pi d}\,\psi_n(e^{\pm i\pi}\eta; e^{\mp i\pi}k_1, \ldots, e^{\mp i\pi}k_d; \Omega) \, ; \tag{6.17}$$

$$\mathbf{RR}: \psi_n^*(\eta; k_1, \ldots, k_d; \Omega) = e^{i\pi d}\,\psi_n(\eta; e^{-i\pi}k_1, \ldots, e^{-i\pi}k_d; e^{-i\pi}\Omega) \, . \tag{6.18}$$

To avoid confusion, in the expressions above that involve complex conjugation, the arguments $\eta, k, \Omega$ on the LHS should be taken to be real, so that analytic continuation occurs only on the RHS.[37]

Similar to the constraint (4.4) on the global de Sitter wavefunction, **CRT** also allows us to write a concise non-perturbative constraint for the full de Sitter wavefunction in the Poincaré patch:

$$\Psi[\eta; \Omega; \phi(\mathbf{y})] = \mathbf{CRT}\,(\Psi[\eta; \Omega; \phi(\mathbf{y})]) \equiv \Psi[e^{-i\pi}\eta; e^{-i\pi}\Omega; \phi(-\mathbf{y})]^* \, , \tag{6.19}$$

$$\Psi[\eta; \Omega; \phi(\mathbf{k}_a)] = \mathbf{CRT}\,(\Psi[\eta; \Omega; \phi(\mathbf{k}_a)]) \equiv \Psi[e^{-i\pi}\eta; e^{-i\pi}\Omega; \phi(\mathbf{k}_a)]^* \, , \tag{6.20}$$

where $\eta$, $\Omega$ and $\phi$ are real, and it is important that the complex conjugation on the RHS is the *final step*, after evaluating the rest of the expression. Note that at $\eta = 0$, **CRT** acts to constrain the wavefunction to be its own conjugate after the spatial reflection and Weyl rotation.

As all of the above analytic continuations are equivalent to the correct Wick rotation, which should work for any fixed background QFT, Eqs. (6.7)–(6.12) will always be valid *non-perturbatively*, at least in the non-gravitational coupling constants.[38] In the next section, we will see how to rewrite these results perturbatively. In that context, the **RR** conditions (6.10) and (6.12) will become the statement of Hermitian analyticity, and (as expected from past work [13, 14, 54]) they are actually valid for *all* FLRW spacetimes, to all orders in perturbation theory (and for Bunch-Davies). This makes sense because unlike the other two transformations, **RR** does not require the existence of a map from $P^+$ to $P^-$.

## 6.2 Perturbative Discrete Symmetries

The transformations in Section 6.1 are somewhat obscure, due to their explicit dependence on the Weyl factor $\Omega$. If, however, we are doing *unrenormalised* perturbation theory around a free, covariant, parity-even bosonic theory, a remarkable simplification appears. This is because the action takes the form:

$$I = \int \mathrm{d}^D x \, \sqrt{g}\, \mathcal{L}_0[\phi, g_{ab}] \, , \tag{6.21}$$

where, apart from the overall factor of $\sqrt{g}$, only *integer* powers of the metric appear in $\mathcal{L}$.[39] In this section, we are defining $I$ with the conventions of the Euclidean action, so that the amplitude always goes like $e^{-I}$, even in Lorentzian signature; thus the Lorentzian action is imaginary.

---

[37]If we interpret the $*$ on the LHS as global complex conjugation, we would get a contradiction as it is not possible to equate an anti-holomorphic function on the LHS with a holomorphic function on the RHS, unless the functions are constant. Hence we may only take the arguments on the LHS to be complex, if we define the conjugate function $f^*(z)$ as the *holomorphic* extension of the complex conjugate acting on the real domain $f : \mathbb{R} \to \mathbb{C}$, i.e. taking the complex conjugate of the Taylor coefficients of $f$. This would need to be distinguished from both the anti-holomorphic extension of the original function $f(z^*)$, or the global complex conjugate $f(z)^* = f^*(z^*)$. This is analogous to the notation for the $\dagger$ discussed in footnote 20.

[38]In the case of gravity, at least two additional problems present themselves: (1) the conformal mode problem, which means that the path integral is not convergent for any value of $q$, and (2) we cannot simply write $g_{ab} = g_{ab}^0 + h_{ab}$ and treat $h_{ab}$ like any other field, because the gauge conditions of the graviton no longer make sense unless we allow $\Omega$ to act on $h_{ab}$ as well. Furthermore, as $H^0$ reverses the sign of $\Lambda$, there is reason to expect that the result here cannot make sense nonperturbatively, as in many approaches to quantum gravity (e.g. the string landscape), $\Lambda$ is not a freely adjustable parameter. That said, when working perturbatively in the gravitational coupling, in all the cases we have checked we get the same phases as in the case of a scalar field.

[39]For spinor fields, the propagator term contains a factor of $\gamma_\mu$ which scales like the square-root of a metric. So in theories with fermions, we expect an additional power of $(-1)$ in **CRT** for each fermion propagator, in any dimension $D$.

It follows that when we analytically continue $\Omega \to -\Omega$ (regardless of which direction we go around $\Omega = 0$), the sole effect is to multiply the $\sqrt{g}$ factor by a minus sign for each spacetime dimension: $(-1)^D$. This is then equivalent to acting on the Lagrangian with the transformation:

$$\mathcal{L}(-\Omega) = (-1)^D \mathcal{L}(\Omega) \,. \tag{6.22}$$

So when $D = d + 1$ is even, there is actually no difference between the correct path and the paths where we analytically continue $\eta$ or $y_i$. We can just (in unrenormalised perturbation theory) ignore the $\Omega \to -\Omega$ instruction.

On the other hand, when $D = d + 1$ is odd, the Lagrangian changes sign: $\mathcal{L} \to -\mathcal{L}$. In any Feynman-Witten diagram, this provides an extra minus sign for every vertex $V$, and also every internal edge $I$ (because the sign of the propagator term also reverses). In any connected diagram, the Euler formula $V - I = 1 - L$ allows us to write the change of any unrenormalised Feynman-Witten amplitude $\mathcal{A}$ in terms of the number of loops $L$:

$$\mathcal{A}^{(L)}(-\Omega) = e^{i\pi(d+1)(L-1)} \mathcal{A}^{(L)}(\Omega) \,, \tag{6.23}$$

where $\mathcal{A}^{(L)}$ denotes a perturbative amplitude computed at some loop order $L$ (e.g. $L = 0$ corresponds to tree-level and $L = 1$ corresponds to 1-loop order), and we allow $d$ to be non-integer. The reason we keep saying "unrenormalised", is that this pattern does not continue to hold when doing renormalisation theory. The reason is that the UV cutoff $\epsilon$ has units of length, and so it implicitly depends on the value of $g_{ab}$ and hence $\Omega$. If, for example, one has a log divergence of the form $C \log(\epsilon)$, then when $\Omega \to e^{-i\pi}\Omega$, this will also send $\epsilon \to e^{-i\pi}\epsilon$, thus producing the shift:

$$C \log\!\big(e^{i\pi}\epsilon\big) = C\big[\log(\epsilon) - i\pi\big] \,, \tag{6.24}$$

so that any log-divergent term produces an imaginary shift in the corresponding finite term (this shift has no explicit dependence on $D$). Of course, this effect does not matter for tree-level diagrams, as these are UV-finite and thus do not require renormalisation. Nor does it affect any loop diagrams that happen to be finite.[40] Also, the coefficient of the log divergence itself still satisfies the phase rule (except when considering a diagram with higher powers of the $\log(\epsilon)$, in which case this remark would apply to the leading order log divergence).[41]

A conceptually similar effect comes if we are doing bulk perturbation theory, not around a free theory, but around an interacting CFT where various operators have non-integer anomalous dimensions. In this case, the conformal perturbation theory can have essentially arbitrary non-integer powers of the metric determinant $g$, and so $\Omega \to -\Omega$ can introduce strange phases even when $D = d + 1$ is even. To say it as a slogan, the **CRT** transformation takes the simple form (6.23) only for amplitudes that involve no bulk quantum anomalies.

In cases without such quantum anomalies, the only extra factor to worry about is $(-1)^{D(1-L)}$, and so, combining our results with those of the previous section, we therefore obtain the following perturbative results for the wavefunction coefficients in position space:

$$\textbf{CRT}: \quad \psi_n^{(L)*}(\eta; y_1, \ldots, y_d) = e^{i\pi(d+1)(L-1)} \, \psi_n^{(L)}\big(e^{-i\pi}\eta; -y_1, \ldots, -y_d\big) \,; \tag{6.25}$$

$$\textbf{D}_{-1}^{\pm}: \quad \psi_n^{(L)}(\eta; y_1, \ldots, y_d) = \psi_n^{(L)}\big(e^{\pm i\pi}\eta; e^{\pm i\pi}y_1, \ldots, e^{\pm i\pi}y_d, \big) \,; \tag{6.26}$$

$$\textbf{RR}: \quad \psi_n^{(L)*}(\eta; y_1, \ldots, y_d) = e^{i\pi(d+1)(L-1)} \, \psi_n^{(L)}\big(\eta; -e^{i\pi}y_1, \ldots, -e^{i\pi}y_d\big) \,, \tag{6.27}$$

---

[40]In the case of power law divergences, one tends to obtain, for any divergent loop, a power $\epsilon^{-D+2n}$ where $n$ is an integer, and hence when $\epsilon \to -\epsilon$ there is an extra factor of $(-1)^D$ for each such divergence. This is actually good since it means that counterterm needed to renormalise a loop diagram has the same sign as the tree level Lagrangian. But, we can also just choose to use a regulator scheme which automatically eliminates power law divergences, and not worry about them.

[41]A similar looking imaginary shift will appear in Section 7.5 for IR log divergent quantities that go like $\log(-\eta)$, when we take the $\eta \to 0$ limit of **CRT** at future infinity.

where $\psi_n^{(L)}$ denotes a perturbative wavefunction coefficient computed at some loop order $L$. Having these constraints in position space may prove to be quite powerful in a position-space cosmological bootstrap program, which is yet to be explored in the literature but could be more well-motivated in terms of connecting to cosmological observations, since imprints of physical principles, such as locality, in the late-time non-Gaussianity is obscured in Fourier space [100].[42] In momentum space we obtain:

$$\textbf{CRT}: \quad \psi_n^{(L)*}(\eta; k_1, \ldots, k_d) \;=\; e^{i\pi(d+1)(L-1)} \; \psi_n^{(L)}(\, e^{-i\pi}\eta; k_1, \ldots, k_d)\,; \tag{6.28}$$

$$\textbf{D}_{-1}^{\pm}: \quad \psi_n^{(L)}(\eta; k_1, \ldots, k_d) \;=\; e^{\pm i\pi d} \; \psi_n^{(L)}(e^{\pm i\pi}\eta; e^{\mp i\pi}k_1, \ldots, e^{\mp i\pi}k_d,)\,; \tag{6.29}$$

$$\textbf{RR}: \quad \psi_n^{(L)*}(\eta; k_1, \ldots, k_d) \;=\; e^{i\pi((d+1)L-1)} \psi_n^{(L)}(\, \eta\,; e^{-i\pi}k_1, \ldots, e^{-i\pi}k_d)\,. \tag{6.30}$$

As this section considers only the case of a parity-even Lagrangian, technically the arguments in this section don't care about whether we reflect the $y$-coordinates in **CRT** or **RR**. However, we have put the signs in the correct place for parity-odd theories as well, as can be deduced in Section 6.3 where we do not restrict to parity-even theories.

The analytic continuation of the conformal time coordinate $\eta$ may result in an interesting connection to the recent work in [101], where the in-in correlators were related to an in-out prescription involving evolution through a pair of Poincaré patches stitched together by their $\eta = 0$ surfaces. The authors of [101] refer to the transformation on the Feynman-Witten diagrams used in this construction as a time reversal which is not quite correct as it is really a **CRT** transformation.

## 6.3 Perturbative Cosmological Optical Theorem for FLRW

Next, we show how to prove (6.27) (and thus (6.30)) perturbatively, for a general spatially-flat FLRW cosmology with arbitrary scale factor $a(\zeta)$:

$$\mathrm{d}s^2 = (A\,\mathrm{d}\zeta)^2 + (B\,a(\zeta)\,\mathrm{d}y_i)^2\,, \tag{6.31}$$

and we continue to define $P^\pm$ as before, even though in the general case there is no way to go from $P^+$ to $P^-$ by an extension of the *real* spacetime manifold. So here, $P^-$ is more of an abstraction representing what would happen if you solve the same QFT but on a contracting spacetime instead of an expanding one.

We now drop the assumptions that the Lagrangian is Lorentz-covariant, or that it is invariant with respect to time or space reversal. This may be done by allowing the action to depend covariantly on the 1-index *purely temporal* and $d$-index *purely spatial* permutation symbols:

$$\boldsymbol{\epsilon}^t\,, \qquad \boldsymbol{\epsilon}^{i_1 i_2 \ldots i_d}\,, \tag{6.32}$$

where we define each permutation symbol so that it is invariant under all real coordinate transforms in (time / space) except those for which det Jacobian $= -1$, in which case the (time / space) permutation symbol flips sign.[43] However, for any kind of analytic continuation, the $\boldsymbol{\epsilon}$'s have to remain the same sign throughout, as the analytic continuation of 1 is 1. Now let a given amplitude be labelled $T = \pm 1$ according to whether it depends on $\boldsymbol{\epsilon}^t$ an (even/odd) number of times, and let it be labelled $X = \pm 1$ according to whether it depends on $\boldsymbol{\epsilon}^{i_1 i_2 \ldots i_d}$ an (even/odd) number of times.

Reflection Reality (**RR**) may now be stated as the principle that on the Euclidean cosmology, each term in the Lagrangian satisfies $\mathcal{L} = T\mathcal{L}^*$, i.e. the $\tau$-reversal even terms are real and the $\tau$-reversal odd

---

[42]AT thanks Carlos Duaso Pueyo for making him aware of this potential application of the position-space constraints.

[43]If a Lorentz covariant parity-odd theory is desired, one simply requires the epsilon symbols to come together in the form of the usual spacetime permutation symbol: $\boldsymbol{\epsilon}^{a_0 a_1 \cdots a_d} = (d+1)\boldsymbol{\epsilon}^{[a_0}\boldsymbol{\epsilon}^{a_1 a_2 \cdots a_d]}$.

terms are imaginary. From this it follows that, if we do the $y$-rotation described in the previous section, we obtain:

$$\psi_n^*(\eta; y_1, \ldots, y_d; \Omega) = T\psi_n(\eta; e^{i\pi}y_1, \ldots, e^{i\pi}y_d; e^{-i\pi}\Omega), \tag{6.33}$$

but note that we are not yet reflecting the $y$-coordinates here. Doing that reflection in each coordinate gives us an additional factor when $d$ and the parity are both odd:[44]

$$\psi_n(\eta; -y_1, \ldots, -y_d; \Omega) = X^d \psi_n(\eta; y_1, \ldots, y_d; \Omega). \tag{6.34}$$

Furthermore, since $\boldsymbol{\epsilon}^t$ substitutes for a factor of $\sqrt{g_{tt}}$ in the Lagrangian $\mathcal{L}$, and $\boldsymbol{\epsilon}^{i_1 i_2 \cdots i_d}$ substitutes for a factor of $\sqrt{g^{(d)}}$ in $\mathcal{L}$—this is equivalent saying you don't need the metric to integrate a $p$-form in $p$ dimensions—the $\Omega$ continuation is also modified from (6.22) and becomes:[45]

$$\psi_n^{(L)}(\eta; y_1, \ldots, y_d; e^{i\pi}\Omega) = TX^d(-1)^{D(L-1)}\psi_n^{(L)}(\eta; y_1, \ldots, y_d; \Omega) = TX^d e^{i\pi D(L-1)}\psi_n^{(L)}(\eta; y_1, \ldots, y_d; \Omega). \tag{6.35}$$

By combining together Eqs. (6.33)–(6.35), we observe that the dependence on $T$ and $X$ cancels out, and so we verify the relation from the previous section:

$$\mathbf{RR}: \qquad \psi_n^{(L)*}(\eta; y_1, \ldots, y_d) = e^{i\pi(d+1)(L-1)}\psi_n^{(L)}(\eta; -e^{i\pi}y_1, \ldots, -e^{i\pi}y_d). \tag{6.36}$$

In momentum space we thus recover (6.30). This is the same as the *Hermitian Analyticity* component of the Cosmological Optical Theorem (COT), see e.g [54], i.e. the part of the COT which does not involve cutting rules.

# 7 The Cosmological CPT Theorem

Our "Cosmological CPT theorem" makes the following three claims:

$$\text{Discrete Scale Invariance} + \mathbf{RR} \text{ invariance} \implies \mathbf{CRT} \text{ invariance} \ ;$$

$$\mathbf{CRT} \text{ invariance} + \text{Discrete Scale Invariance} \implies \mathbf{RR} \text{ invariance} \ ;$$

$$\mathbf{CRT} \text{ invariance} + \mathbf{RR} \text{ invariance} \implies \text{Discrete Scale Invariance} \ ,$$

where we remind the reader that Reflection Reality is a discrete symmetry which **ALL** unitary theories have to satisfy non-perturbatively, as well as to all orders in perturbation theory.

In Section 6, we already outlined the necessary analytic continuations needed to define these three symmetries {**CRT**, **RR**, **D**}. In Sections 7.1–7.3, we will rederive the perturbative form of these symmetries, from a more concrete and detailed perspective. In Section 7.4 we will take the limit as $\eta \to 0$ in order to derive the form of the symmetries at the boundary, including the phase formula. Section 7.5 will then check this result by comparing it to specific calculations in the literature.

---

[44] We do not attempt to extend this formula to non-integer $d$. It is problematic to write down parity-odd terms in non-integer dimensions, as $\boldsymbol{\epsilon}^{i_1 i_2 \cdots i_d}$ would have a non-integral number of indices and therefore cannot be fully contracted with normal tensors. If we instead consider the effect of flipping *all* spatial dimensions, this is equivalent to a rotation in $SO(d)$ when $d =$ even, and so one expects it have no effect on the correlator for rotationally invariant theories that are defined in general $d$.

[45] Some readers may be wondering why the $X$ and $T$ factors in (6.35) do not depend on the number of loops $L$, even though our parity-even result did depend on the number of loops. But that was because flipping the sign of the Lagrangian changes both the interactions and the propagators, leading to an invocation of the Euler formula. On the other hand, $T$ and $X$ just count the individual number of odd items in the Feynman-Witten diagram, without regard to whether the items are vertices or edges. In an action where *all* terms were odd under $X$ or $T$, this would provide a factor that also depends on the number of loops, but we make no such assumption here, as in general it is more natural for there to be both odd and even terms in $L$.

## 7.1 Unitarity and the Bunch-Davies Vacuum

In this section we discuss our approach to enforcing the Bunch-Davies vacuum in a way that is consistent with Unitarity. Establishing such consistency is particularly important here as it enforces the direction in which we must rotate the spacetime coordinates through the complex plane. This extends the discussion in Section 3.3 where we defined the correlation functions in terms of the vacuum of the full theory, $|\Omega\rangle$. In the presence of interactions perturbation theory requires that we rewrite this state in terms of the free (Bunch-Davies) vacuum. It is standard to equate these two states in the early time limit by introducing a small imaginary contribution to the time, $\eta_i = -\infty(1 - i\varepsilon)$, rotating the contour off the negative real axis and turning off the interactions. To see this consider the free vacuum expanded in the basis of energy eigenstates of the full theory (see [102] for a more pedagogical derivation),

$$|0\rangle = \sum_N |N\rangle\langle N|0\rangle. \tag{7.1}$$

We then consider the evolution of this state in the full theory,

$$e^{-iH(\eta-\eta_i)}|0\rangle = \sum_n e^{-iE_n(\eta-\eta_i)}|n\rangle\langle n|0\rangle \tag{7.2}$$

$$= e^{-iE_\Omega(\eta-\eta_i)}|\Omega\rangle\langle\Omega|0\rangle + \sum_{N\neq\Omega} e^{-iE_N(\eta-\eta_i)}|N\rangle\langle N|0\rangle \tag{7.3}$$

$$= e^{-iE_\Omega(\eta-\eta_i)}\left(|\Omega\rangle\langle\Omega|0\rangle + \sum_{N\neq\Omega} e^{-i(E_N-E_\Omega)(\eta-\eta_i)}|N\rangle\langle N|0\rangle\right). \tag{7.4}$$

The vacuum is, by definition, the lowest energy state in the theory and so $E_N - E_\Omega > 0$. Therefore, in the limit $\eta_i \to -\infty(1 - i\varepsilon)$ only the first term remains and

$$e^{-iH(\eta-\eta_i)}|0\rangle = e^{-iH(\eta-\eta_i)}|\Omega\rangle\langle\Omega|0\rangle. \tag{7.5}$$

This is just the Hamiltonian time evolution operator and so

$$\langle\mathcal{O}(\eta)\rangle = \langle 0|U_I^\dagger(\eta)\mathcal{O}_I(\eta)U_I(\eta)|0\rangle \tag{7.6}$$

However, as was noted in [103, 104], this approach spoils the Unitarity of the time evolution operator,

$$U_I^{-1}(\eta) = \overline{\mathcal{T}}\exp\left[-i\int_{-\infty(1-i\varepsilon)}^\eta d\eta H_{int}[\phi,\phi']\right] \neq U_I^\dagger(\eta) = \overline{\mathcal{T}}\exp\left[-i\int_{-\infty(1+i\varepsilon)}^\eta d\eta H_{int}[\phi,\phi']\right]. \tag{7.7}$$

Therefore, following [103], we instead introduce the shifted Hamiltonian,

$$H^\varepsilon = He^{\varepsilon\eta}, \tag{7.8}$$

which was shown to produce the same results perturbatively as the more standard rotation. Furthermore, it has the same effect of turning off the interactions in the infinite past so that interacting and free vacua agree thus we remain in the Bunch-Davies vacuum state. The reason for this is that, in both cases the $\varepsilon$-prescription contributes a negative real part to the exponent of the Hamiltonian in the infinite past thereby ensuring that it vanishes,

$$\lim_{\eta\to-\infty} H^\varepsilon \sim \lim_{\eta\to-\infty} e^{ik_T\eta+\varepsilon\eta} = \lim_{\eta\to-\infty} e^{i(k_T-i\varepsilon)\eta} = 0. \tag{7.9}$$

From (7.9) we can see that this $\varepsilon$-prescription is equivalent to introducing a small negative imaginary part to the energies. In terms of this new Hamiltonian the expectation value is

$$\langle\mathcal{O}(\eta)\rangle = \langle 0|U_\varepsilon^\dagger(\eta)\mathcal{O}_I(\eta)U_\varepsilon(\eta)|0\rangle, \quad U_\varepsilon(\eta) = \mathcal{T}\exp\left[-i\int_{-\infty}^\eta d\eta H_{int}^\varepsilon[\phi,\phi']\right], \tag{7.10}$$

which is now a well defined, unitary operator.

When we act with each of our three transformations it is important to ensure that we remain in the same vacuum state (else they cannot be symmetries of the theory). This will require that we rotate each of our coordinates in a particular direction through the complex plane when moving from the positive to the negative real axis (or vice versa). To see this we first consider $\mathbf{RR}$, which involves a rotation of the momenta through the negative half of the complex plane,

$$\mathbf{k} \to e^{-i\theta}\mathbf{k}\,. \tag{7.11}$$

From these momenta we generate a set of energies[46], $k = \sqrt{\mathbf{k} \cdot \mathbf{k}}$, which are the variables that are typically used to express both correlation functions and wavefunction coefficients. Under this rotation the energy transforms as

$$k \to \sqrt{\mathbf{k} \cdot \mathbf{k}\, e^{-2i\theta}} = \sqrt{\mathbf{k} \cdot \mathbf{k}}\sqrt{e^{-2i\theta}}\,, \tag{7.12}$$

where the final line follows as $k$ is real. To ensure the single-valuedness of $k$ during this rotation we need to continuously follow a Riemann sheet of the square root (i.e. we will not add any factors of $2\pi$ to the exponent). Therefore, we have that

$$k \to e^{-i\theta}k\,. \tag{7.13}$$

In order for the interactions to vanish in the early time limit and thus to remain in the Bunch-Davies vacuum we must ensure that

$$\mathrm{Re}(ik\eta + \varepsilon\eta) < 0 \Rightarrow \lim_{\eta \to -\infty} e^{ik\eta + \varepsilon\eta} = 0\,. \tag{7.14}$$

Under this rotation in the energies this condition becomes,

$$\mathrm{Re}(ike^{-i\theta}\eta + \varepsilon\eta) = k\sin(\theta)\eta + \varepsilon\eta > 0 \tag{7.15}$$

which is true for positive $\theta$ (thus retroactively justifying our decision to rotate into the lower half of the complex plane). The direction of this rotation (as well as the fact that the factor $\varepsilon$ is left unchanged) was noticed in the discussion of the Cosmological Optical Theorem [13] where the $i\varepsilon$ prescription was included on the energy, $k \to k - i\varepsilon$. In that case it was necessary to complex conjugate the energy when sending $k \to -k^*$ to keep the sign of $\varepsilon$ constant. Although here the $\varepsilon$ is treated independently of $k$ the same behaviour is observed, $k$ is reflected and $\varepsilon$ remains unchanged.

Next, we consider the $\mathbf{CRT}$ transformation acting on this Hamiltonian. In this case the momentum is unchanged and we just rotate the time coordinate, $\eta \to \eta e^{-i\theta}$ so that,

$$\mathrm{Re}(ik\eta e^{-i\theta} + \varepsilon\eta e^{-i\theta}) = k\sin(\theta)\eta + \varepsilon\eta\cos(\theta)\,. \tag{7.16}$$

This is a linear combination of sines and cosines and so could equivalently be represented as a shifted sine function which is non-zero at $\theta = 0$. Therefore, this function is guaranteed to change sign over the range $0 \leq \theta \leq \pi$ (or $-\pi \leq \theta \leq 0$) and we must also rotate $\varepsilon$ alongside $\eta$, $\varepsilon \to \varepsilon e^{i\theta}$, so that the exponent is

$$\mathrm{Re}(ik\eta e^{-i\theta} + \varepsilon\eta) = k\sin(\theta)\eta + \varepsilon\eta \tag{7.17}$$

which is negative just as for the rotation in $k$.

Finally, we must consider the simultaneous rotation of $k$ and $\eta$ this comes from the analytic continuation of the scaling transformation and so when we send $\eta \to \lambda\eta$ we must simultaneously send $k \to \lambda^{-1}k$. Therefore, the two rotate in opposite directions through the complex plane such that the combination $k\eta$

---

[46]Although there is an absence of time translation symmetry in cosmology the literature refers to the magnitude of a spatial momentum vector as "energy", as it plays an analogous role to energy in flat space.

is unchanged. As a result of this the exponent in the infinite time limit of the Hamiltonian has a real part given by,

$$\text{Re}(ik\eta + \varepsilon\eta e^{i\theta}) = \varepsilon\eta\cos(\theta). \tag{7.18}$$

Just as before this will change sign and so we must rotate $\varepsilon$ in the same direction as $k$. This doesn't enforce a particular direction to the rotation of $k$ or $\eta$ however.

## 7.2 The Loop Momentum Integral

In Section 6.2 we established the transformation properties of the wavefunction coefficients from the analytic continuation of the coordinates and metric. Here we present an alternative perspective to reach the same conclusions by looking at the explicit calculation of the wavefunction coefficients.

In order to achieve this goal we first break up the wavefunction into distinct parts that are easier to tackle on their own. As explained in [35] the wavefunction coefficients can be calculated by performing a series of nested time integrals plus, potentially, some loop integrals,

$$\psi_n'^{(L)}(\eta_0; \mathbf{k}; \varepsilon) = i^V \int^{\eta_0} \prod_v^V a_v^{d+1} \, d\eta_v e^{\varepsilon\eta_v} \prod_l^L d^d\mathbf{P}_l \prod_a^n K_{k_a}(\eta) \prod_i^I G_{p_i}(\eta, \eta') \delta^d \left( \sum_a^n \mathbf{k}_a \right), \tag{7.19}$$

where the prime in $\psi_n'^{(L)}$ is used to explicitly highlight that these wavefunction coefficients contain the $\delta$-function whilst those without the $\delta$-function will be denoted by $\psi_n^{(L)}$. The external energies are indicated by $k_a = |\mathbf{k}_a|$ whilst the $p_i$ are the internal energies which are built from the $\mathbf{k}_a$ plus potentially some of the loop momenta, $\mathbf{P}_l$. Note that here we have made the $\varepsilon$ prescription explicit and that there are no derivatives acting on any of our propagators. This is purely for the sake of notational simplicity and adding any derivatives will not add any technical complications.

We do not explicitly consider UV divergences in this section (which can lead to some additional corrections with a relative $i$ sign, as discussed in Section 6.2). However, as our results are valid in general dimensions, it is straightforward to renormalise these divergences by dimensional regularisation and this will be discussed explicitly in Section 7.5.

The first component we will look at is the time integral which runs from $-\infty$ to $\eta_0$ when we have a transformation involving the time coordinate it is these limits that pick up a minus sign. To remove this we redefine the variables inside the integral with $\eta_v \to -\eta_v$[47] The effect of this on the measure factor is then straightforward as everything is just some power of $\eta$.

The loop integral is slightly more involved. To treat this we first change variables so that rather than doing the loop integral over some $d$ dimensional vectors we instead perform it over some set of loop energies. This requires a change of basis which was elucidated for arbitrary loops in [105]. For our purposes, the most important fact about this determinant is that it is related to the volume of a simplex with edges that are the lengths of squares of energies. Therefore, its zeros and functional form are unchanged under a transformation which sends $k \to -k$. Explicitly, for a single loop, the integration measure becomes

$$\int d^d\mathbf{P}_l = \frac{N}{\sqrt{\text{Vol}^2(s)}} \int_\Gamma \prod_i^{I_l} 2p_i \, dp_i \left[ \frac{\text{Vol}^2(p,s)}{\text{Vol}^2(s)} \right]^{\frac{d-I_l-1}{2}}, \tag{7.20}$$

where $I_l$ is the number of internal lines in the loop, $l$; $\mathbf{P}_l$ is the momentum running through this loop; and $N$ accounts for the integral over the $d - I_L$ sphere plus an extra, unimportant, proportionality constant.

---

[47] Note that whether this minus sign is $e^{i\pi}$ or $e^{-i\pi}$ will be important but, as it can rotate in a different way depending on which transformation we are considering, we will keep things vague at this point. When we write explicit expressions for the transformations we will be careful to make clear in which direction the rotation goes.

Under the rescalling $p \to \lambda p$ and $s \to \lambda s$ this transforms as

$$\lambda^d \int \mathrm{d}^d \mathbf{P}_l = \frac{N\lambda^{2I_l}}{\sqrt{\lambda^{2I_l-2}}\sqrt{\mathrm{Vol}^2(s)}} \int_\Gamma \prod_i^{I_l} 2p_i \, \mathrm{d}p_i \left[ \frac{\lambda^{2I_l}\mathrm{Vol}^2(p,s)}{\lambda^{2I_l-2}\mathrm{Vol}^2(s)} \right]^{\frac{d-I_l-1}{2}} . \tag{7.21}$$

For positive $\lambda$ we can see that the scaling on each side agrees. Naively, on the left hand side continuing $\lambda \to -\lambda$ would appear to have no impact. However, from the right hand side we can see how our analytic continuation in the energies alters this conclusion, instead recovering a factor of $(-1)^d$. To see this, we first note that the region, $\Gamma$, depends only on the squares of the energies and so is unchanged. However, each of our square roots of energies does pick up a factor of $-1$. As we now have a potentially non-integer exponent in the dimension of the integral $d$ we will need to be careful with $e^{i\pi d}$ vs $e^{-i\pi d}$. For **RR**, in accordance with the discussion in Section 7.1, all rotations in the energy must be taken in the lower half plane and so $\lambda = \lambda e^{-i\pi}$. Therefore we have,

$$(-\lambda)^d \int \mathrm{d}^d \mathbf{P}_l = \frac{Ne^{-i\pi d}\lambda^d}{\sqrt{\mathrm{Vol}^2(s)}} \int_\Gamma \prod_i^{I_l} p_i \, \mathrm{d}p_i \left[ \frac{\mathrm{Vol}^2(p,s)}{\mathrm{Vol}^2(s)} \right]^{\frac{d-I_l-1}{2}} . \tag{7.22}$$

We next explore the transformations of the propagators. To do this we consider the differential equation,

$$\mathcal{O}_\mathbf{k}(\eta) = a^{d-1}\frac{\partial^2}{\partial\eta^2} + (d-1)a'a^{d-2}\frac{\partial}{\partial\eta} + m^2 a^{d+1} + a^{d-1}k^2 \Rightarrow \begin{cases} \mathcal{O}_\mathbf{k}(\eta)G_k(\eta,\eta') &= -i\delta(\eta-\eta'), \\ \mathcal{O}_\mathbf{k}(\eta)K_k(\eta) &= 0. \end{cases} \tag{7.23}$$

The propagators additionally satisfy the boundary conditions

$$\lim_{\eta\to\eta_0} K_k(\eta) = 1, \qquad\qquad \lim_{\eta\to-\infty} K_k(\eta,\eta') \propto e^{ik\eta}, \tag{7.24}$$

$$\lim_{\eta,\eta'\to\eta_0} G_k(\eta,\eta') = 0, \qquad\qquad \lim_{\eta,\eta'\to-\infty} G_k(\eta,\eta') \propto e^{ik\eta}. \tag{7.25}$$

These boundary conditions at early time will ensure that our $i\varepsilon$ prescription will cause the early time integrand to vanish. Let us first consider three relationships that our differential operator satisfies. These will later be linked to symmetries of the wavefunction and are labeled accordingly,

$$\mathbf{RR} : \mathcal{O}_\mathbf{k}^*(\eta) = \mathcal{O}_{e^{-i\pi}\mathbf{k}}(\eta), \tag{7.26}$$

$$\mathbf{CRT} : \mathcal{O}_\mathbf{k}^*(\eta) = e^{-i\pi(d+1)}\mathcal{O}_\mathbf{k}(e^{-i\pi}\eta), \tag{7.27}$$

$$\mathbf{D}_{-1}^\pm : \mathcal{O}_\mathbf{k}(\eta) = e^{\pm i\pi(d+1)}\mathcal{O}_{e^{\mp i\pi}\mathbf{k}}(e^{\pm i\pi}\eta). \tag{7.28}$$

Here, the direction of the rotations in $\mathbf{k}$ and $\eta$ are dictated by the asymptotic time behaviour, preserving the Bunch-Davies vacuum discussed in Section 7.1 and the $\pm$ sign in $\mathbf{D}_{-1}^\pm$ represents the ambiguity in the direction of rotation of $\eta \to e^{\pm i\pi}\eta$[48].

We start by considering Reflection Reality,

$$\mathcal{O}_\mathbf{k}^*(\eta)K_k^*(\eta) = 0, \qquad \lim_{\eta\to\eta_0} K_k^*(\eta) = 1, \qquad \lim_{\eta\to-\infty} K_k^*(\eta) \propto e^{-ik\eta}, \tag{7.29}$$

$$\mathcal{O}_{e^{-i\pi}\mathbf{k}}(\eta)K_{e^{-i\pi}k}(\eta) = 0, \qquad \lim_{\eta\to\eta_0} K_{e^{-i\pi}k}(\eta) = 1, \qquad \lim_{\eta\to-\infty} K_{e^{-i\pi}k}(\eta) \propto e^{-ik\eta}. \tag{7.30}$$

In light of (7.26) these two differential equations are identical to each other as are the boundary conditions therefore,

$$\mathbf{RR} : K_k^*(\eta) = K_{e^{-i\pi}k}(\eta). \tag{7.31}$$

---

[48] Note that the ambiguity in the sign of the Scale Invariance transformation is meaningless in integer dimension but in non-integer dimension it introduces a seemingly meaningful choice which we will discuss more later.

Similarly, the bulk-bulk propagator satisfies

$$-\mathcal{O}_{\mathbf{k}}^*(\eta)G_k^*(\eta,\eta') = -i\delta(\eta-\eta'), \qquad \lim_{\eta\to\eta_0} -G_k^*(\eta,\eta') = 0, \qquad \lim_{\eta\to-\infty} -G_k^*(\eta,\eta') \propto e^{-ik\eta}, \quad (7.32)$$

$$\mathcal{O}_{e^{-i\pi}\mathbf{k}}(\eta)G_{e^{-i\pi}k}(\eta,\eta') = -i\delta(\eta-\eta'), \quad \lim_{\eta\to\eta_0} G_{e^{-i\pi}k}(\eta,\eta') = 0, \quad \lim_{\eta\to-\infty} G_{e^{-i\pi}k}(\eta,\eta') \propto e^{-ik\eta}, \quad (7.33)$$

which gives a very similar expression for this propagator,

$$\mathbf{RR} : G_k^*(\eta,\eta') = -G_{e^{-i\pi}k}(\eta,\eta'). \tag{7.34}$$

Notice, that we are free to introduce a minus sign in the early time limit in (7.32), this is because the boundary condition when we introduce $\varepsilon$ is that the propagator vanishes, so it is just the sign of the imaginary exponent that needs to be fixed.

The **CRT** transformation acts similarly,

$$\mathcal{O}_{\mathbf{k}}^*(\eta)K_k^*(\eta) = 0, \qquad \lim_{\eta\to\eta_0} K_k^*(\eta) = 1, \qquad \lim_{\eta\to-\infty} K_k^*(\eta) \propto e^{-ik\eta}, \tag{7.35}$$

$$\mathcal{O}_{\mathbf{k}}(e^{-i\pi}\eta)K_k(e^{-i\pi}\eta) = 0, \qquad \lim_{\eta\to\eta_0} K_k(e^{-i\pi}\eta) = 1, \qquad \lim_{\eta\to-\infty} K_k(e^{-i\pi}\eta) \propto e^{-ik\eta}. \tag{7.36}$$

The additional phase introduced to the differential operator can simply be dropped when considering the bulk-boundary propagator as the right hand side of the differential equation is zero and so,

$$\mathbf{CRT} : K_k^*(\eta) = K_k(e^{-i\pi}\eta). \tag{7.37}$$

For the Green's function this is not the case, but the boundary conditions can have an arbitrary phase introduced,

$$\mathcal{O}_{\mathbf{k}}^*(\eta)G_k^*(\eta,\eta') = i\delta(\eta-\eta'), \qquad\qquad \mathcal{O}_{\mathbf{k}}(e^{-i\pi}\eta)G_k(e^{-i\pi}\eta,e^{-i\pi}\eta') = i\delta(\eta-\eta'), \tag{7.38}$$

$$\lim_{\eta\to\eta_0} G_k^*(\eta,\eta') = 0, \qquad\qquad \lim_{\eta\to\eta_0} e^{i\pi(d+1)}G_k(e^{-i\pi}\eta,e^{-i\pi}\eta') = 0, \tag{7.39}$$

$$\lim_{\eta\to-\infty} G_k^*(\eta,\eta') \propto e^{-ik\eta}, \qquad\qquad \lim_{\eta\to-\infty} e^{i\pi(d+1)}G_k(e^{-i\pi}\eta,e^{-i\pi}\eta') \propto e^{-ik\eta}, \tag{7.40}$$

where the time reversal has introduced an extra minus sign into the delta function due to its action on the limits of the integral. To see this consider,

$$\int_{-\infty}^{\eta'} \mathrm{d}\eta\, \delta(\eta-\eta') \to \int_{\infty}^{-\eta'} \mathrm{d}\eta\, \delta(\eta+\eta') = -\int_{-\infty}^{\eta'} \mathrm{d}\eta\, \delta(-\eta+\eta'). \tag{7.41}$$

Once again, we can see that (7.27) allows us to extract a relationship between the complex conjugate of the bulk-bulk propagator and its analytic continuation,

$$\mathbf{CRT} : G_k^*(\eta,\eta') = e^{i\pi(d+1)}G_k(e^{-i\pi}\eta,e^{-i\pi}\eta'). \tag{7.42}$$

Next we consider dilatations. This symmetry simply introduces a phase to both equations whilst leaving the boundary conditions unchanged. Therefore,

$$\mathbf{D}_{-1}^{\pm} : K_k(\eta) = K_{e^{\mp i\pi}k}(e^{\pm i\pi}\eta), \text{ and } G_k(\eta,\eta') = e^{\mp i\pi d}G_{e^{\mp i\pi}k}(e^{\pm i\pi}\eta,e^{\pm i\pi}\eta'). \tag{7.43}$$

Thus, we see that the bulk-boundary propagators pick up very simple relationships from each of our symmetries. The bulk-bulk propagators have similar relationships but additionally introduce a dimension dependent phase.

Finally, we must consider the transformation of the delta function. This will behave similarly to the right hand side of the differential equation for the bulk-bulk propagator (7.41),

$$\delta^d \left( e^{i\theta} \sum_a^k \mathbf{k}_a \right) = e^{-i\theta d} \delta^d \left( \sum_a^k \mathbf{k}_a \right). \tag{7.44}$$

This is unlike what we expect for the delta function which, for real numbers, transforms as

$$\delta(ax) = \frac{1}{|a|} \delta(x). \tag{7.45}$$

However, just as for the loop and time integrals, the analytic continuation that we perform will also rotate the integrals over the external momenta by $e^{i\theta d}$. This is then matched by the analytic continuation of the delta function.

We now have all the ingredients to understand how our three transformations work on a wavefunction coefficient. The first symmetry we will consider is Reflection Reality which relates the complex conjugate to the rotation of all momenta through the lower half of the complex plane,

$$\mathbf{RR} : \left[ \psi_n'^{(L)}(\eta_0; \mathbf{k}; \varepsilon) \right]^* \tag{7.46}$$

$$= (-i)^V \int^{\eta_0} \prod_v^V a_v^{d+1} \, d\eta_v e^{\varepsilon \eta_v} \prod_l^L d^d \mathbf{P}_l \prod_a^n K_{k_a}^*(\eta) \prod_i^I G_{p_i}^*(\eta, \eta') \delta^d \left( \sum_a^n \mathbf{k}_a \right) \tag{7.47}$$

$$= (-i)^V \int^{\eta_0} \prod_v^V a_v^{d+1} \, d\eta_v e^{\varepsilon \eta_v} \prod_l^L d^d \mathbf{P}_l \prod_a^n K_{e^{-i\pi} k_a}(\eta) \prod_i^I -G_{e^{-i\pi} p_i}(\eta, \eta') \delta^d \left( \sum_a^n \mathbf{k}_a \right) \tag{7.48}$$

$$= e^{i\pi(I - V + d(L-1))} i^V \int^{\eta_0} \prod_v^V a_v^{d+1} \, d\eta_v e^{\varepsilon \eta_v} \prod_l^L d^d \left( e^{-i\pi} \mathbf{P}_l \right) \prod_a^n K_{e^{-i\pi} k_a}(\eta) \prod_i^I G_{e^{-i\pi} p_i}(\eta, \eta') \delta^d \left( e^{-i\pi} \sum_a^n \mathbf{k}_a \right) \tag{7.49}$$

$$= e^{i\pi(1+d)(L-1)} \psi_n'^{(L)}(\eta_0; e^{-i\pi} \mathbf{k}; \varepsilon). \tag{7.50}$$

where in the penultimate line we have employed the Feynman-Euler relationship, $V - I + L = 1$ and the rotation on $\mathbf{k}$ applies to all internal and external energies. Next we investigate Scale Invariance which rotates all momenta magnitudes as well as the conformal time,

$$\mathbf{D}_{-1}^{\pm} : \psi_n'^{(L)}(\eta_0; \mathbf{k}; \varepsilon) = i^V e^{\pm i\pi d(V + L - I - 1)} \int^{\eta_0} \prod_v^V e^{\mp i\pi d} a_v^{d+1} \, d\eta_v e^{\varepsilon \eta_v} \prod_l^L d^d \left( e^{\mp i\pi} \mathbf{P}_l \right) \tag{7.51}$$

$$\times \prod_a^n K_{e^{\mp i\pi} k_a}(e^{\pm i\pi} \eta) \prod_i^I G_{e^{\mp i\pi} p_i}(e^{\pm i\pi} \eta, e^{\pm i\pi} \eta') \delta^d \left( e^{\mp i\pi} \sum_a^n \mathbf{k}_a \right) \tag{7.52}$$

$$= i^V \int^{e^{\pm i\pi} \eta_0} \prod_v^V a_v^{d+1} \, d\eta_v e^{\mp i\pi \varepsilon \eta_v} \prod_l^L d^d \left( e^{\mp i\pi} \mathbf{P}_l \right) \prod_a^n K_{e^{\mp i\pi} k_a}(\eta) \prod_i^I G_{e^{\mp i\pi} p_i}(\eta, \eta') \delta^d \left( e^{\mp i\pi} \sum_a^n \mathbf{k}_a \right) \tag{7.53}$$

$$= \psi_n'^{(L)}(e^{\pm i\pi} \eta_0; e^{\mp i\pi} \mathbf{k}; e^{\mp i\pi} \varepsilon). \tag{7.54}$$

The final transformation we will consider is **CRT** which flips the conformal time with an overall complex

conjugation of the wavefunction coefficient,

$$\mathbf{CRT} : \left[\psi_n'^{(L)}(\eta_0; \mathbf{k}; \varepsilon)\right]^* \tag{7.55}$$

$$= (-i)^V \int^{\eta_0} \prod_v^V a_v^{d+1} \, \mathrm{d}\eta_v e^{\varepsilon \eta_v} \prod_l^L \mathrm{d}^d \mathbf{P}_l \prod_a^n K_{k_a}^*(\eta) \prod_i^I G_{p_i}^*(\eta, \eta') \delta^d \left(\sum_a^n \mathbf{k}_a\right) \tag{7.56}$$

$$= (-i)^V e^{-i\pi dV} \int^{\eta_0} \prod_v^V e^{i\pi d} a_v^{d+1} \, \mathrm{d}\eta_v e^{\varepsilon \eta_v} \prod_l^L \mathrm{d}^d \mathbf{P}_l \prod_a^n K_{k_a}(e^{-i\pi}\eta) \prod_i^I e^{i\pi(d+1)} G_{p_i}(e^{-i\pi}\eta, e^{-i\pi}\eta') \delta^d \left(\sum_a^n \mathbf{k}_a\right) \tag{7.57}$$

$$= e^{i\pi(I-V)(d+1)} i^V \int^{-\eta_0} \prod_v^V a_v^{d+1} \, \mathrm{d}\eta_v e^{e^{-i\pi}\varepsilon \eta_v} \prod_l^L \mathrm{d}^d \mathbf{P}_l \prod_a^n K_{k_a}(\eta) \prod_i^I G_{p_i}(\eta, \eta') \delta^d \left(\sum_a^n \mathbf{k}_a\right) \tag{7.58}$$

$$= e^{i\pi(1+d)(L-1)} \psi_n'^{(L)}(e^{-i\pi}\eta_0; \mathbf{k}; e^{i\pi}\varepsilon) \,. \tag{7.59}$$

|       | 1     | CRT   | RR           |
| ----- | ----- | ----- | ------------ |
| 1     | 1     | CRT   | RR           |
| CRT   | CRT   | 1     | $\mathbf{D}_{-1}^+$ |
| RR    | RR    | $\mathbf{D}_{-1}^-$ | 1 |

Table 2: Table showing how the transformations combine for $\psi_n^{(L)}$. In this table the first transformation is in the column heading and then the row heading second. We see that **CRT** and **RR** each square to 1. Furthermore, they combine to give $\mathbf{D}_{-1}^\pm$. Since *a priori* (before imposing dilation symmetry) the operations $\mathbf{D}_{-1}^+$ and $\mathbf{D}_{-1}^-$ act differently, we no longer have the $\mathbb{Z}_2 \times \mathbb{Z}_2$ structure as the dilatation operator doesn't square to itself. Instead this group has the same structure as the group $\mathrm{Aut}(\mathbb{Z})$ of automorphisms of the integers.

An interesting observation one can make is that for $\psi_n'^{(L)}$, **CRT** and **RR** introduce equal phases, and in even spacetime dimensions all three transformations leave any real $\psi_n'^{(L)}$ invariant at any loop order, provided $\psi_n'^{(L)}$ is UV-finite. This also highlights the fact that in the original contact Cosmological Optical Theorem [13,14] the minus sign picked up between $\psi_n^{(L)}$ and its reflected counterpart can be understood as an artefact of the fact that the $\delta$-function has been stripped off. Indeed, when we consider $\psi_n^{(L)}$ without this delta function stripped off we find

$$\mathbf{CRT} : \left[\psi_n^{(L)}(\eta_0; \mathbf{k}; \varepsilon)\right]^* = e^{i\pi(d+1)(L-1)} \psi_n^{(L)}(e^{-i\pi}\eta_0; \mathbf{k}; e^{i\pi}\varepsilon) \,, \tag{7.60}$$

$$\mathbf{D}_{-1}^\pm : \psi_n^{(L)}(\eta_0; \mathbf{k}; \varepsilon) = e^{\pm i\pi d} \psi_n^{(L)}(e^{\pm i\pi}\eta_0; e^{\mp i\pi}\mathbf{k}; e^{\mp i\pi}\varepsilon) \,, \tag{7.61}$$

$$\mathbf{RR} : \left[\psi_n^{(L)}(\eta_0; \mathbf{k}; \varepsilon)\right]^* = e^{i\pi((d+1)L-1)} \psi_n^{(L)}(\eta_0; e^{-i\pi}\mathbf{k}; \varepsilon) \,. \tag{7.62}$$

These relationships each imply an operator that leaves the wavefunction coefficient unchanged up to an overall phase in the presence of the corresponding symmetry,

$$\mathbf{CRT}\left(\psi_n^{(L)}\right) = \left[\psi_n^{(L)}(e^{-i\pi}\eta_0, \mathbf{k}, e^{i\pi}\varepsilon)\right]^* = e^{i\pi(d+1)(L-1)} \psi_n(\eta_0; \mathbf{k}; \varepsilon) \,, \tag{7.63}$$

$$\mathbf{D}_{-1}^\pm\left(\psi_n^{(L)}\right) = \psi_n^{(L)}(e^{\pm i\pi}\eta_0; e^{\mp i\pi}\mathbf{k}; e^{\mp i\pi}\varepsilon) = e^{\mp i\pi d} \psi_n(\eta_0; \mathbf{k}; \varepsilon) \,, \tag{7.64}$$

$$\mathbf{RR}\left(\psi_n^{(L)}\right) = \left[\psi_n^{(L)}(\eta_0; e^{-i\pi}\mathbf{k}; \varepsilon)\right]^* = e^{i\pi((d+1)L-1)} \psi_n(\eta_0; \mathbf{k}; \varepsilon) \,, \tag{7.65}$$

where, for clarity, we have kept the $\varepsilon$-dependence to explicitly show how it rotates. Although, in the final wavefunction coefficient $\psi_n^{(L)}$ we take $\varepsilon \to 0$ so that this rotation, ultimately, will not influence

these transformations[49]. These symmetry transformations can then be combined, as presented in Table 2. Just as in the flat space case the combination of **CRT** and **RR** generates the other transformations and they square to themselves. However, here we do not recover the $\mathbb{Z}_2 \times \mathbb{Z}_2$ group but instead the group $\text{Aut}(\mathbb{Z})$ of automorphisms of the integers. It is important to note that we need both $\mathbf{D}^\pm_{-1}$ for this group to close as (in arbitrary dimensions) they are not generically self inverse. This is not an obstacle as both transformations are implied by the Scale Invariance of the theory due to the fact that the Bunch-Davies condition is preserved by both rotations (this condition is what restricts us from considering rotations in the other directions for either **CRT** or **RR**). Interestingly, the ambiguity in this rotation direction appears absent from $\psi'^{(L)}_n$ where $\mathbf{D}^\pm_{-1}$ doesn't introduce a phase. However, this is merely an artifact of the dilatation symmetry, without imposing it the two transformations act differently. Furthermore, to connect **RR** to **CRT** we still need to be able to rotate in both directions using our dilatation operator. To see this consider how $\mathbf{D}^+_{-1}$ acts on an already transformed wavefunction coefficient,

$$\mathbf{D}^+_{-1}\left(\mathbf{RR}\left(\psi^{(L)}_n\right)\right) = \mathbf{D}^+_{-1}\left(e^{i\pi(-1+(d+1)L)}\psi^{(L)}_n\right) = e^{i\pi(d+1)(L-1)}\psi^{(L)}_n = \mathbf{CRT}\left(\psi^{(L)}_n\right), \qquad (7.66)$$

$$\mathbf{D}^+_{-1}\left(\mathbf{CRT}\left(\psi^{(L)}_n\right)\right) = \mathbf{D}^+_{-1}\left(e^{i\pi(d+1)(L-1)}\psi^{(L)}_n\right) = e^{i\pi((d+1)L-2d-1)}\psi^{(L)}_n \neq \mathbf{RR}\left(\psi^{(L)}_n\right). \qquad (7.67)$$

We can see that this agrees with the group structure in Table 2,

$$\mathbf{CRT} \cdot \mathbf{RR} = \mathbf{D}^+_{-1} \Rightarrow \left\{ \begin{array}{l} \mathbf{D}^+_{-1} \cdot \mathbf{RR} = \mathbf{CRT}\,, \\ \mathbf{D}^+_{-1} \cdot \mathbf{CRT} = \mathbf{CRT} \cdot \mathbf{RR} \cdot \mathbf{CRT} \neq \mathbf{RR}\,, \end{array} \right. \qquad (7.68)$$

where the second implication follows from the fact that the **CRT** transformation squares to one.

We will now move on to discuss how these transformations can be used to identify constraints of the $\mathcal{I}^+$ boundary correlators where $\eta = 0^-$ which will be of particular importance for dS/CFT!

## 7.3   Spinning Fields

The analysis above also applies to spinning fields. To see this it is convenient to use the following notations and conventions. In $d+1$-dimensional spacetime, for traceless[50], integer spin fields we use the free action developed in [106] and discussed in the context of the cosmological bootstrap in [15, 107]:

$$S = \int \mathrm{d}^{d+1}x \, [a(\eta)]^{d-1} \frac{1}{2s!} \left[ \left(\sigma'_{i_1\ldots i_s}\right)^2 - c_s^2 \left(\partial_j \sigma_{i_1\ldots i_s}\right)^2 - \delta c_s^2 \left(\partial^j \sigma_{j i_2\ldots i_s}\right)^2 - m^2 a^2 \left(\sigma_{i_1\ldots i_s}\right)^2 \right]. \qquad (7.69)$$

The totally-symmetric, traceless tensor $\sigma_{i_1\ldots i_s}$ has spatial indices $i_1 = 1, \ldots, d$ which span the $d$-dimensional spacelike hypersurface orthogonal to the $\eta$ coordinate.[51] $\sigma_{i_1\ldots i_s}$ has $(2s+1)$ components, which each create states ("particles") with helicities $0, \pm 1, \ldots, \pm s$, and we have enforced invariance under dilatations by including inverse factors of the scale factor for each coordinate derivative. Following [15, 107] we Fourier transform and diagonalise this using the helicity modes, $\sigma_h$, defined by:

$$\sigma_{i_1\ldots i_s}(\eta; x) = \int_{\mathbf{k}} e^{i\mathbf{k}\cdot\mathbf{x}} \sum_{h=-S}^{S} \mathbf{e}^h_{i_1\ldots i_s}(\mathbf{k}) \sigma_h(\eta; \mathbf{k})\,, \qquad (7.70)$$

---

[49]Of course, $\varepsilon$ is important in these transformations as it dictates how $\mathbf{k}$ and $\eta_0$ transform but when considering the final wavefunction it can be safely dropped if we keep this rotation direction in mind.

[50]To consider fields with a non-zero trace, one can subtract the trace and treat it as an additional scalar field.

[51]The Effective Field Theory of Inflation (EFToI) [108] is derived by considering a theory which is only invariant under spatial diffeomorphisms, for which there is a preference for the co-ordinate choice used in unitary gauge, where the time coordinate is chosen to coincide with the surfaces of constant value of the field $\sigma_{i_1\ldots i_s}$. The EFToI thus encapsulates generic class of models of inflation where spatial diffeomorphisms are preserved. (7.69) can be written in a covariant way by using the Goldstone boson $\pi$ of time translations to upgrade the spatial tensor $\sigma_{i_1\ldots i_s}$ to a covariant spacetime tensor. The coupling of $\sigma_{i_1\ldots i_s}$ to $\pi$ is also dictated by this constructions but we will not need this here.

These helicity tensors are defined as an outer product of helicity vectors,

$$\mathbf{e}^h_{i_1\dots i_s} = \mathbf{e}^{h_1}_{i_1}\dots\mathbf{e}^{h_s}_{i_s}, \tag{7.71}$$

which satisfy the following relations:

$$\mathbf{e}^h_i(\mathbf{k})\left[\mathbf{e}^{h'}_i(\mathbf{k})\right]^* - 4\delta_{hh'} = 0 \qquad \text{(orthogonality and normalisation)}, \tag{7.72}$$

$$\left[\mathbf{e}^h_i(\mathbf{k})\right]^* - \mathbf{e}^h_i(-\mathbf{k}) = 0 \qquad\qquad (\sigma_{i_1\dots i_s}(x) \text{ is real}). \tag{7.73}$$

Note that these fields are not assumed to be transverse, $h$ is allowed to take $d$ different values including $0$ where $\mathbf{e}^0$ is proportional to the momentum. The contributions from the other helicity modes are therefore transverse by the orthogonality condition.

We parameterise the WFU, $\Psi$, at conformal time $\eta_0$ in terms of the helicities of the integer spin field as

$$\Psi[\eta_0;\sigma(\mathbf{k})] = \exp\left[-\sum_{n=2}^{\infty}\frac{1}{n!}\sum_{h_i=\pm}\int_{\mathbf{k}_1,\dots,\mathbf{k}_n}\psi_n^{h_1\dots h_n}(\eta_0;\mathbf{k})(2\pi)^d\delta^d\left(\sum\mathbf{k}_a\right)\sigma_{h_1}(\eta_0;\mathbf{k}_1)\dots\sigma_{h_n}(\eta_0;\mathbf{k}_n)\right]. \tag{7.74}$$

Spatial translations and spatial rotations ensure that wavefunction coefficients can be written as a product of a *helicity factor*, which is an $SO(d)$ invariant function of helicity vectors and spatial momenta, multiplied by a *trimmed wavefunction coefficient* which is only a function of the magnitudes of the momenta in the literature:

$$\psi_n^{h_1\dots h_n} = (\text{tensor structure}) \times (\text{trimmed wavefunction coefficient}). \tag{7.75}$$

We take all coefficients appearing in the tensor structure to be real and therefore include any factors of $i$ that might appear when converting to momentum space, or simply as part of the Feynman rules, in the trimmed part which we will denote as $\psi_n$ for brevity. For a general $\psi_n^{h_1\dots h_n}$, we have

$$\psi_n^{h_1\dots h_n} = \left[\mathbf{e}^{h_1}(\mathbf{k}_1)\dots\mathbf{e}^{h_n}(\mathbf{k}_n)\,\mathbf{k}_1^{\alpha_1}\dots\mathbf{k}_n^{\alpha_n}\right]\psi_n, \tag{7.76}$$

for some integer $\alpha_i$. Note that we can choose for *both parity-even and parity-odd* tensor structures to be invariant with respect to the discrete symmetries of **CRT**, **RR** and **D** which we discuss in this paper.[52] Hence, all our results extend directly from the scalar $\psi_n$ discussed in Section 7.2 to the tensor case.

## 7.4 Constraints on Boundary Wavefunction Coefficients

All of the results that we have presented in this section so far rely on an analytic continuation, either in momenta or time. However, by considering the wavefunction of the universe at the future boundary of dS, where the time dependence often trivialises, we can derive direct constraints on the wavefunction coefficients. This future boundary is where the reheating surface is understood to live in the inflationary paradigm. Therefore, such constraints are of particular cosmological relevance.

In the late time limit the bulk fields take the form

$$\lim_{\eta\to 0^-}\phi(\eta,\mathbf{y}) = \overline{\phi}_+(\mathbf{y})\eta^{\Delta^+} + \overline{\phi}_-(\mathbf{y})\eta^{\Delta^-}, \tag{7.77}$$

$$= \overline{\phi}_+(\mathbf{y})\eta^{\Delta} + \overline{\phi}_-(\mathbf{y})\eta^{d-\Delta}, \tag{7.78}$$

---

[52]Had we used another convention where the tensor structures scaled in non-trivial way with Dilatations **D**, then the scalar component of the field would have to scale in the inverse way, in order to leave the field $\sigma_{i_1\dots i_s}$ invariant.

where for the case of scalars, the mass $m^2/H^2 = m^2\ell^2 = \Delta(d - \Delta)$ is related to the conformal dimension in the usual way $\Delta = d/2 + \nu$, $\nu = \sqrt{d^2/4 - m^2/H^2}$.[53] For massive spin-$s$ fields the mass and spin are related to the conformal dimension by $\Delta = d/2 + \mu$, $\mu = \sqrt{(d + 2s - 4)^2/4 - m^2/H^2}$. We define $\overline{\phi}_+$ with $\Delta^+ \equiv \Delta$ and $\overline{\phi}_-$ (with $\Delta^- = d - \Delta$). For heavy scalar fields ($m > dH/2$), $\Delta$ is complex and so these boundary operators do not represent a self adjoint basis[54]. We will therefore restrict our discussion to light fields and leave a more comprehensive examination, including heavy fields, to future work. In this case, $\Delta$ is real and positive so the $\overline{\phi}_-$ term dominates. Starting from the expansion of the wavefunction in terms of its coefficients, 3.71, we can similarly define a "boundary" wavefunction[55] which is a functional of these boundary fields $\overline{\phi}_-$ and has no explicit $\eta$-dependence

$$\overline{\Psi}[\overline{\phi}_-(\mathbf{k}_a)] = \exp\left\{-\sum_{n=2}^{\infty} \int \left[\prod_{a=1}^{n} \frac{\mathrm{d}^d k_a}{(2\pi)^d} \overline{\phi}_-(\mathbf{k}_a)\right] \overline{\psi}_n(\mathbf{k})\delta^d\left(\sum \mathbf{k}_a\right)\right\}, \tag{7.79}$$

where we have introduced boundary wavefunction coefficients[56] denoted by $\overline{\psi}_n$ to distinguish them from the bulk wavefunction coefficients $\psi_n$.[57] The choice to Taylor expand the wavefunction in the field operator with dimension $d - \Delta$ rather than using the seemingly more sensible choice of labeling this operator's dimension $\Delta$ is purely conventional. It is standard to refer to $\Delta^+$ as *the dimension* of a field in cosmology (for example a massless field has dimension $\Delta = d$) but it turns out that this is not the part of the field that survives on the boundary.

Let us now count the scalings of each term in the exponent one by one to see how $\overline{\psi}_n(\mathbf{k}_a)$ scales under $\mathbf{k} \to \lambda\mathbf{k}$ which in turn will scale the internal and external energies $k \to \lambda k$. The $d$-dimensional $\mathrm{d}^d k$ measures each scale with volume and thus they will scale with an overall factor of $\lambda^{nd}$. This cancels with an equivalent scaling of the Fourier transformed $\overline{\phi}_-(\mathbf{k}_a)$ which give an overall scaling of $\lambda^{-nd}$.[58] Each boundary field must contribute an extra scaling like $\eta^{\Delta-d}$ to ensure the Scale Invariance of the bulk field (7.78). The result of this is that in addition to the factor coming from the Fourier Transform the fields contribute a further $\lambda^{dn-\sum_\alpha \Delta_\alpha}$. Finally, the $\delta^d(\mathbf{k}_1 + \ldots + \mathbf{k}_n)$ scales with inverse volume $\lambda^{-d}$.[59] We know that the exponent of (7.79) must be dimensionless, thus by dimensional analysis, any IR-finite boundary wavefunction coefficient will scale as

$$\overline{\psi}_n(\lambda\mathbf{k}) = \lambda^d \lambda^{-nd} \lambda^{\sum_\alpha \Delta_\alpha} \overline{\psi}_n(\mathbf{k}) = \lambda^{d(1-n)+\sum_\alpha \Delta_\alpha} \overline{\psi}_n(\mathbf{k}). \tag{7.81}$$

We will now look at the constraints from **CRT**, **RR** and **D$_{-1}$** on the boundary wavefunction coefficients $\overline{\psi}_n$, where this dimensional analysis exercise will prove to be useful. Let us look at each transformation individually.

---

[53]Note for AdS this relation is $m^2\ell^2_{AdS} = \Delta(\Delta - d)$. In AdS $\Delta^\pm \in \mathbb{R}$ for scalar fields of any mass. In dS for scalar fields with mass $m^2/H^2 > d^2/4$, $\Delta^\pm \in \mathbb{C}$ which is known as the principal series; fields with mass $m^2/H^2 \leq d^2/4$, $\Delta^\pm \in \mathbb{R}$ which is known as the complementary series.

[54]Of course, we could have made the choice to expand the fields in a self-adjoint way but this would sacrifice boundary conformal invariance as such an expansion would necessarily mix terms of different weights. See e.g. [55–64] for discussions regarding the principal series.

[55]Although this is referred to as a "boundary" wavefunction in the literature, this is not really a wavefunction living at the $\mathcal{I}^+$ where $\eta = 0^-$, since from (7.77) it is clear that only massless fields would survive in the limit of $\eta = 0^-$. The more accurate description is that the coefficients of the wavefunction have had their $\eta$-dependence stripped off.

[56]These can be interpreted as the correlators of the operators of some conjectured CFT living at the boundary, i.e. dS/CFT.

[57]To obtain the boundary wavefunction coefficient we differentiate the boundary wavefunction $\overline{\Psi}$ with respect to the sources $\overline{\phi}_-$ which have dimension $d - \Delta$

$$\overline{\psi}_n(\mathbf{k}) \equiv \lim_{\overline{\phi}_- \to 0} \frac{\delta^n}{\delta^n \overline{\phi}_-} \overline{\Psi}[\overline{\phi}_-(\mathbf{k}_a)]. \tag{7.80}$$

[58]This is in the exact same way that in (3.73) the original bulk field $\phi_{\mathbf{k}_a}$ scaled as $\lambda^d$ resulting in the bulk wavefunction coefficient scaling inversely to the $\delta^d(\sum \mathbf{k}_a)$.

[59]Usually in the context of computations in de Sitter, the $\delta(\mathbf{k}_1 + \ldots + \mathbf{k}_n)$-function is omitted but we remind the reader of its importance in terms of arguments involving Scale Invariance, as it scales with inverse volume, as well as its necessity for ensuring momentum conservation which comes from translation invariance in position space.

**RR** does not care about the time at which the wavefunction is evaluated and so the resulting constraint is unchanged:[60]

$$\mathbf{RR}: \left[\overline{\psi}_n^{(L)}(\mathbf{k})\right]^* = e^{i\pi[(d+1)L-1]}\overline{\psi}_n^{(L)}(e^{-i\pi}\mathbf{k}). \tag{7.82}$$

Scale Invariance in the bulk requires that

$$\psi_n^{(L)}(\eta/\lambda; \lambda\mathbf{k}) = \lambda^d \psi_n^{(L)}(\eta; \mathbf{k}). \tag{7.83}$$

As was derived in (7.81), Scale Invariance of the boundary theory at $\mathcal{I}^+$ where $\eta = 0^-$ tells us that the wavefunction coefficients must scale as

$$\overline{\psi}_n^{(L)}(\lambda\mathbf{k}) = \lambda^{d(1-n)+\sum_\alpha \Delta_\alpha}\overline{\psi}_n^{(L)}(\mathbf{k}), \tag{7.84}$$

where $\Delta_\alpha$ are the dimensions of the external fields. The Cosmological CPT theorem tells us that the $SO^+(1,1)$ boost which is the important continuous symmetry in the flat space CPT theorem acts in the Poincaré patch as dilatations $\mathbf{D}$, i.e. Scale Invariance $\lambda \in \mathbb{R}^+$, which we can analytically continue to $SO(2)$, provided we have a Hamiltonian bounded from below. Under analytic continuation of $\lambda \in \mathbb{R}^+$ through the complex plane $\lambda \in \mathbb{C}$,[61] we can then access $\lambda = e^{-i\pi}$, to conclude that:

$$\overline{\psi}_n^{(L)}(e^{-i\pi}\mathbf{k}) = (-1)^{d(1-n)+\sum_\alpha \Delta_\alpha}\overline{\psi}_n^{(L)}(\mathbf{k}), \tag{7.85}$$

where for cases with fractional $d$ and $\Delta$ we have

$$\mathbf{D}_{-1}^\pm: \overline{\psi}_n^{(L)}(e^{\mp i\pi}\mathbf{k}) = e^{\mp i\pi[d(1-n)+\sum_\alpha \Delta_\alpha]}\overline{\psi}_n^{(L)}(\mathbf{k}). \tag{7.86}$$

We need to take into account that $\overline{\psi}_n^{(L)}$ is analytic through the lower half-plane $\mathbb{C}^{-i}$. Given that the **RR** (7.82) transformation rotates $\overline{\psi}_n^{(L)}$ counter-clockwise, we must use the $\mathbf{D}_{-1}^+$ in (7.86) to land on

$$\mathbf{D}_{-1}^+ \cdot \mathbf{RR}: \left[\overline{\psi}_n^{(L)}(\mathbf{k})\right]^* = e^{i\pi[(d+1)L-1-d(1-n)-\sum_\alpha \Delta_\alpha]}\overline{\psi}_n^{(L)}(\mathbf{k}). \tag{7.87}$$

We can also derive the same expression independently using **CRT**, which tells us directly that

$$\left[\psi_n^{(L)}(\eta; \mathbf{k})\right]^* = e^{i\pi(d+1)(L-1)}\psi_n^{(L)}(e^{-i\pi}\eta; \mathbf{k}), \tag{7.88}$$

$$\left[\overline{\psi}_n^{(L)}(\mathbf{k})\right]^* = e^{i\pi[(d+1)(L-1)+dn-\sum_\alpha \Delta_\alpha]}\overline{\psi}_n^{(L)}(\mathbf{k}), \tag{7.89}$$

where for (7.88) we have to continue $1/\eta \propto k \equiv |\mathbf{k}|$ in the lower-half plane $\mathbb{C}^{-i}$, and as previously explained the factor of $\eta^{\sum_\alpha d - \Delta_\alpha}$ comes from the fields scaling as $\eta^{\Delta-d} \propto \lambda^{d-\Delta}$ at the boundary. We have now identified the discrete symmetries which the boundary wavefunction coefficients $\overline{\psi}_n^{(L)}$ satisfy:

$$\mathbf{CRT}: \quad \overline{\psi}_n^{(L)}(\mathbf{k}) = e^{-i\pi[(d+1)(L-1)+dn-\sum_\alpha \Delta_\alpha]}\left[\overline{\psi}_n^{(L)}(\mathbf{k})\right]^*, \tag{7.90}$$

$$\mathbf{D}_{-1}^\pm: \quad \overline{\psi}_n^{(L)}(\mathbf{k}) = e^{\pm i\pi[d(1-n)+\sum_\alpha \Delta_\alpha]}\overline{\psi}_n^{(L)}(e^{\mp i\pi}\mathbf{k}), \tag{7.91}$$

$$\mathbf{RR}: \quad \overline{\psi}_n^{(L)}(\mathbf{k}) = e^{-i\pi[(d+1)L-1]}\left[\overline{\psi}_n^{(L)}(e^{-i\pi}\mathbf{k})\right]^*, \tag{7.92}$$

---

[60]We remind the reader that the conventions we use in cosmology where the conformal time on conformally flat Poincaré slicing $\eta \in [-\infty, 0^-]$ is negative in the expanding branch and increases from $-\infty$ in the infinite past to $\mathcal{I}^+ = 0^-$.

[61]Given that $\overline{\psi}_n^{(L)}$ is only analytic in the lower half-place $\overline{\psi}_n^{(L)} \in \mathbb{C}^{-i}$, we can see from the group structure in Table 2 we must act with $\mathbf{D}_{-1}^+$ after **RR** in order to obtain **CRT**; or alternatively act with **RR** after $\mathbf{D}_{-1}^-$.

where the last condition from **RR** holds for boundary correlators in any flat FLRW spacetime with a future conformal boundary. We can use (7.90) to solve for the phase of $\overline{\psi}_n^{(L)}$ directly, obtaining the following phase formula for the boundary wavefunction coefficients:

$$e^{i\arg(\overline{\psi}_n^{(L)})} \equiv \frac{\overline{\psi}_n^{(L)}(\mathbf{k})}{|\overline{\psi}_n^{(L)}(\mathbf{k})|} = \pm\sqrt{\frac{\overline{\psi}_n^{(L)}(\mathbf{k})}{\overline{\psi}_n^{(L)*}(\mathbf{k})}} = \pm(-i)^{(d+1)(L-1)+dn-\sum_\alpha \Delta_\alpha}\,, \tag{7.93}$$

where there is a $\pm$ out front because **CRT** cannot determine the overall real sign, because obtaining the phase involves taking a square root.[62] Hence, we obtain the result (1.10) quoted in Section 1.2:

$$\arg(\overline{\psi}_n^{(L)}) = -\frac{\pi}{2}\left((d+1)(L-1)+dn-\sum_\alpha \Delta_\alpha\right) + \pi\mathbb{N}\,. \tag{7.94}$$

Remarkably (7.93) and (7.94) will hold for any $\overline{\psi}_n^{(L)}$ computed in cosmology provided that:

- spacetime is de Sitter (possibly with boost-breaking terms);
- the Lagrangian is locally **CRT**-invariant;
- the amplitude is UV and IR finite;
- and involves fields in representations with integer spin and real $\Delta$ (no spinors or principal series).

Furthermore, the phase of $\overline{\psi}_n^{(L)}$ has no dependence on the details of the bulk interactions, e.g. derivative couplings will have the same phase, provided the Feynman-Witten diagrams have the same external legs with the same $\Delta$'s.

The steps leading up to this phase formula implicitly assumed that the amplitudes were IR and UV-finite. We have already discussed the $-i\pi$ shift due to UV log divergences in Section 6.2; we now make the corresponding remark for IR divergences. For a typical IR-divergent amplitude, there is a term like $C\log(-\eta)$ which, upon rotating $\eta$ from $0^-$ to $0^+$, becomes $C\log(-e^{-i\pi}\eta)$ This provides an extra $-i\pi C$ shift that adjusts the IR-finite piece of the amplitude, which does not conform to (7.94). But, the leading order IR-divergence will continue to satisfy (7.94). An example of such a calculation will be quoted in Section 7.5 below.

## 7.5 Explicit Checks of the Phase Formula

We will now provide a list of non-trivial checks of the phase formula (7.93) and (7.94) for various specific Feynman-Witten diagrams in cosmology.

The first five calculations **i**–**v** below all involve conformally coupled scalars (denoted by $\varphi$), while the remaining two **vi** and **vii** involve massless fields (denoted by $\phi$). For the calculations in **iii**–**vii**, we work in non-integer dimensions for the purposes of dimensional regularisation (dim-reg) using the prescription described in Appendix C of [109]. In this approach, the mass of the field is also renormalised to keep the order of the Hankel function $\nu$ fixed, which ensures that the integrals can be computed analytically in the dim-reg parameter $\delta$.[63]

---

[62]Although, for the 2-point function of a field which is weakly coupled in the bulk, this sign is fixed by normalisability.

[63]An alternative prescription proposed in [110] does not renormalise the mass of the field, resulting in the order of the Hankel function for the case of massless fields in $d = 3+\delta$-dimensions becoming $\nu = d/2 = (3+\delta)/2$. However, this prevents the integrals from being computed analytically. Consequently, the authors carry out an expansion in $\delta$ prior to computing the integrals which introduces non-trivial corrections at various points in the calculation. This makes the overall answer non-analytic in $\delta$ and prevents a direct comparison with the phase formula in (7.93) and (7.94).

### i. Conformal tree-level IR-finite coefficient $\overline{\psi}_{3,\varphi\varphi\tilde{\sigma}}^{\text{tree}}$

The phase of the contact three-point wavefunction coefficient $\overline{\psi}_{3,\varphi\varphi\tilde{\sigma}}^{\text{tree}}$, generated via a simple cubic interaction $\varphi\varphi\tilde{\sigma}$ in $3+1$-dimensions, involving two conformally-coupled scalars (with $\Delta = 2$) and a massive scalar $\tilde{\sigma}$ with mass $m < \sqrt{2}H$ (corresponding to $2 < \Delta \leq 3$), in order for the time integral to converge in the IR (i.e. when $\eta = 0^-$), is found in Appendix B of [13] to be

$$\arg(\overline{\psi}_{3,\varphi\varphi\tilde{\sigma}}^{\text{tree}}) = \frac{\pi}{2}\left(\nu + \frac{1}{2}\right). \tag{7.95}$$

This agrees with (7.93) and (7.94) as it correctly predicts the generically complex phase of a wavefunction coefficient involving a light field with a mass $m < \sqrt{2}H$.

### ii. Conformal tree-level IR-divergent coefficient $\overline{\psi}_{3,\varphi\varphi\varphi}^{\text{tree}}$

The contact three-point wavefunction coefficient $\overline{\psi}_{3,\varphi\varphi\varphi}^{\text{tree}}$ for the interaction involving three conformally-coupled scalars, generated via a simple cubic interaction $\varphi\varphi\varphi$ in $3+1$-dimensions, where the time integral diverges in the IR (i.e. when $\eta = 0^-$), is found to be

$$\overline{\psi}_{3,\varphi\varphi\varphi}^{\text{tree}} = \frac{i(\log(-i(k_1 + k_2 + k_3)\eta_0) + \gamma)}{(\eta_0)^3 H^4} + O\left(\frac{1}{\eta_0^2}\right), \tag{7.96}$$

i.e. it has a $\log(-\eta)$ IR-divergence. However, as we previously explained (7.93) and (7.94) continue to hold for the phase of the coefficient of the leading $\log(-\eta)$ IR-divergence and thus we find correctly that there is an $i$ in front of the $\log(-\eta)$ term. This is another non-trivial check of the phase formula as it correctly predicts the imaginary phase of the $\log(-\eta)$ in the IR-divergent wavefunction coefficient.

### iii. Conformal tree-level IR-finite coefficient in non-integer dimension $d = 3 + \delta$, $\overline{\psi}_{6,\varphi^6}^{\text{tree}}$

The phase of the contact six-point wavefunction coefficient $\overline{\psi}_{6,\varphi^6}^{\text{tree}}$ for the interaction involving six conformally-coupled scalars, generated via a simple 6th-order interaction $\varphi^6$ in non-integer $d = 3 + \delta$ dimensions (which rescales the conformal dimension of the conformally coupled scalar $\Delta = d/2 + 1/2 = (4 + \delta)/2$), was first computed in [111]. However, an incorrect phase was found, which will be corrected in [72]. For this particular Feynman-Witten diagram one finds

$$\overline{\psi}_{6,\varphi^6}^{\text{tree}} = -\frac{e^{-i\pi\delta}\Gamma(3 + 2\delta)}{H^{4+\delta}\left(k_T^{(6)}\right)^{3+2\delta}}. \tag{7.97}$$

Once again (7.93) and (7.94) correctly predict the generically complex phase of the wavefunction coefficient in non-integer dimensions $d = 3 + \delta$.

### iv. Conformal 1-loop UV-finite coefficient in non-integer dimension $d = 3 + \delta$, $\overline{\psi}_{4,\varphi^6}^{(L=1)}$

The phase of the 1-site 1-loop four-point wavefunction coefficient $\overline{\psi}_{4,\varphi^6}^{(L=1)}$ for the interaction involving four conformally-coupled scalars, generated via a 6th-order interaction $\varphi^6$ in non-integer $d = 3 + \delta$-dimensions (which, as previously explained, rescales the conformal dimension of the conformally coupled scalar $\Delta = d/2 + 1/2 = (4 + \delta)/2$), was first computed in [111]. However, an incorrect phase was found, which will be corrected in [72]. For this particular Feynman-Witten diagram one finds

$$\overline{\psi}_{4,\varphi^6}^{(L=1)} = \frac{\lambda e^{-i\pi\delta}}{2^{5+\delta}\pi^{\frac{4+\delta}{2}}H^2\left(k_T^{(4)}\right)^{1+\delta}}\Gamma(1 + \delta)\Gamma\left(\frac{2 + \delta}{2}\right). \tag{7.98}$$

Since $\overline{\psi}_{4,\varphi^6}^{(L=1)}$ comes from a UV-finite diagram we see that is is analytic in the $k_T$-energy pole as $\delta \to 0$. As predicted we find $\lim_{\delta\to 0} \overline{\psi}_{4,\varphi^6}^{(L=1)} \in \mathbb{R}$. Additionally (7.93) and (7.94) correctly predicts the generically complex phase of $\overline{\psi}_{4,\varphi^6}^{(L=1)}$ when $\delta$ is kept finite. This will be important when considering the next examples, which are UV-divergent.

### v. Conformal $1$-loop UV-divergent coefficient in non-integer dimension $d = 3 + \delta$, $\overline{\psi}_{n,\varphi^{(n+4)/2}}^{(L=1)}$

The 2-site, 1-loop de Sitter wavefunction coefficient for arbitrary number $n$ of conformally coupled fields in external legs in non-integer $d = 3 + \delta$-dimensions will be computed in upcoming work [112][64] where the author generalises the recently developed differential equation approach [113,114] to computing flat FLRW cosmological wavefunction coefficients to loop diagrams. The result for the phase of the UV-divergent term is

$$\arg\left(\overline{\psi}_{n,\varphi^{(n+4)/2}}^{(L=1)}\right) = -\frac{n\pi(2+\delta)}{4} + \pi\mathbb{N}\,, \tag{7.99}$$

which is again compatible with our phase formula. Note that the phase for $n = 4$ matches the phase of (7.98) because the phase formula depends only on the external legs and not the internal structure of the Feynman-Witten diagram.

### vi. Massless $1$-loop coefficient in non-integer dimension $d = 3 + \delta$, $\overline{\psi}_{2,\phi'^3}^{(L=1)}$

As previously discussed, in cases with a logarithmic UV-divergence, one can renormalise these using dim-reg. In dim-reg, the logarithmic UV-divergence turns into a $1/\delta$ divergence (for IR-finite 1-loop diagrams), which can be canceled by expanding the phase using the simple fact that any generic complex number $A$ can be expressed as $A = |A|e^{i\arg(A)}$. For IR-finite 1-loop diagrams we thus find

$$\lim_{\delta\to 0}\left[|\overline{\psi}^{(L=1)}|e^{i\arg(\overline{\psi}^{(L=1)})}\right] \sim \frac{1}{\delta}(1 \pm i\pi\delta + O(\delta^2)) = \frac{1}{\delta} \pm i\pi + O(\delta)\,. \tag{7.100}$$

This phenomenon found in [111], where the authors computed a parity-odd contribution to the scalar trispectrum using the in-in formalism. We have demonstrated that this is in fact a generic feature of UV-divergent loop diagrams and will be explored further in [67].

In [97], the 1-loop four-point wavefunction coefficient $\overline{\psi}_{2,\phi'^3}^{(L=1)}$ for the interaction involving three massless scalars, generated via a boost-breaking derivative interaction $\phi'^3$ in non-integer $d = 3+\delta$-dimensions (which rescales the conformal dimension of the massless scalar $\Delta = d/2 + 3/2 = (6 + \delta)/2$), was computed with an incorrect phase that will be corrected in [72]. For this particular Feynman-Witten diagram one finds

$$\overline{\psi}_{2,\phi'^3}^{(L=1)} = H^2 \frac{S_{1+\delta}}{(2\pi)^{3+\delta}} k^{3+\delta}(-iH)^\delta I(\delta)\,, \tag{7.101}$$

where $S_{d-2}$ is the surface area of the $(d-2)$-dimensional unit sphere (i.e. $S_1 = 2\pi$) and $I(\delta)$ is a dimensionless quantity with no contribution to the phase defined in [97]. As we stated in Sections 7.2 and 7.4 the phase formula still holds for the case of derivative interactions; the phase is also important for the consistency of the cutting rules derived in [97].

### vii. Graviton 2-pt function in integer dimension $\quad \psi_2^{(L)} = \langle T_{ij}(\mathbf{k})T_{lm}(-\mathbf{k})\rangle$

This graviton 2-point function in cosmology is equivalent to calculating the 2-point function of the stress tensor in dS/CFT. Our phase rule also gives the correct prediction in this case. The answer depends on the dimension $d$ and loop order $L$ as follows:

---

[64]We thank Tom Westerdijk for kindly sharing this result from his upcoming work with us.

- In even $d + 1$-spacetime dimensions, $\langle T_{ij}(\mathbf{k})T_{lm}(-\mathbf{k})\rangle \in \mathbb{R}$ to ALL loop order;

- In odd $d + 1$-spacetime dimensions, $\langle T_{ij}(\mathbf{k})T_{lm}(-\mathbf{k})\rangle \in i\mathbb{R}$ at tree level and even loop order (i.e. when $L \in 2\mathbb{Z}$);

- In odd $d + 1$-spacetime dimensions, $\langle T_{ij}(\mathbf{k})T_{lm}(-\mathbf{k})\rangle \in \mathbb{R}$ at odd loop order (i.e. when $L \in 2\mathbb{Z} + 1$).

Hence for the case of $d = 2$ we expect the central charge, which is equivalent to the coefficient of the 2-point stress tensor correlator, to be imaginary at tree level and real at 1-loop order and so on. This matches what has been found in the literature, for a list of previous results see Section 1.3.

## 8 Discussion

In this paper we have provided a new perspective on the CPT theorem, which naturally enables us to make converse statements. Our analysis highlights the profound interplay between Unitarity, Lorentz Invariance, and the discrete symmetries. By demonstrating that any two of the $\mathbb{Z}_2$ symmetries in the set $\{\mathbf{RR}, \mathbf{CRT}, 180°\}$ imply the third, we have established a more unified and fundamental understanding of these fundamental principles, in both flat and de Sitter spacetimes.

Importantly, as shown in Section 5, the $\mathbf{RR}$ symmetry alone is often enough to ensure full Unitarity over a codimension-0 domain of generic couplings in many field theories. This means that assuming $\mathbf{CRT}$ together with covariance is often sufficient to obtain a physically realistic QFT. This is very convenient from a practical point of view, since the detailed statement of Unitarity in Lorentzian spacetimes is quite finicky — for example it often requires fine-tuning the measure factors in the path integral — if you try to impose it directly. This perspective is even more valuable in cosmology, where the dynamical nature of the spacetime makes it tricky to use the (manifestly unitary but not manifestly covariant) Hamiltonian formalism.

In the cosmological context, we used discrete symmetries to derive non-perturbative constraints on both cosmological correlators, and wavefunction coefficients. This allowed us to extend the tree-level results for Hermitian Analyticity [13, 14, 54]. Our non-perturbative $\mathbf{RR}$ result requires analytically continuing the momenta $\mathbf{k}$ and the bulk Weyl factor $\Omega$. We also derived a somewhat more explicit perturbative statement of the result, at arbitrary loop order, which like [13, 14, 54] involves an analytic continuation of the momenta $\mathbf{k}$ alone. This result holds for all flat FLRW cosmologies.

In the more restrictive context of Poincaré de Sitter, one can instead invoke $\mathbf{CRT}$ by analytically continuing the $\eta$ coordinate. These results reveal that the $\mathbf{CRT}$ constraint is powerful enough to determine the phase of wavefunction coefficients at future infinity. Surprisingly, this does not require any analytic continuation or comparison to past infinity; the constraints can just be read off directly at the level of the wavefunction coefficients.

It should be noted, however, that in $d =$ even spatial dimensions, the phase does depend on whether the number of bulk loops is even or odd. Also, in any dimension, when the bulk theory has a logarithmic UV or IR divergence, the associated finite term gets an imaginary shift relative to the coefficient of the log divergence. Most of our equations have been written for general real dimension $d$, making it easy to apply our results to calculations using dimensional regularisation.

Our phase rule has important implications for the dS/CFT correspondence, providing a new tool to determine the phases of correlators in the holographic dual conformal field theory, thereby advancing our understanding of quantum gravity in cosmological settings. As dS/CFT theories are non-unitary—in the sense that the boundary theory, thought of as a Euclidean field theory, does not itself satisfy a **boundary-RR**, they do not satisfy the reality conditions of normal unitary theories. It is therefore critical to identify

the (presumably equally constraining) properties that a dS/CFT must satisfy to be dual to a unitary bulk. This article provides substantial progress towards this goal, but further work is needed to clarify what this constraint means from the perspective of the boundary field theory. It would also be useful if this reality constraint can be promoted to a more restrictive positivity constraint on dS/CFT, similar to how Reflection Positivity is more restrictive than Reflection Reality in unitary theories. Put another way, our reality conditions are *necessary* for bulk unitarity, but it would be even nicer to identify a necessary and *sufficient* condition for bulk unitarity.

Our **CRT** results also have some applicability to inflation cosmology, due to the slow roll approximation which makes the spacetime approximately described by Poincaré de Sitter. The constraints from **CRT** are compatible with a breaking of de Sitter isometries (namely the breaking of de Sitter boosts but preservation of Discrete Scale Invariance) during inflation, so long as **CRT** still manifests as a symmetry of the local Lagrangian, as discussed in Section 4. However, in general the existence of a nonzero spectral tilt $n_s - 1$ (related to the slow-roll parameter) implies that this form of **CRT** is slightly broken, which should lead to small corrections to our **CRT** result for wavefunction coefficients.

There are a number of other directions in which this work can be usefully extended. Our results lead fairly directly to a no-go theorem for certain specific cosmological correlators of bosonic fields [67] derived from the phase formula in this paper. An interesting direction which should certainly be explored is using a similar formalism to constrain spinors using discrete symmetries, which is yet to be done in the cosmology literature. To derive such constraints it will be necessary for us to use projective representations of the group $\mathbb{Z}_2 \times \mathbb{Z}_2$ with the appropriate double cover of the group for half-integer spin fields being the dihedral group $D_4$. Further work should also certainly be done on extending the analysis of the boundary wavefunction coefficients to the case of heavy fields in the principal series (discussed by [55–64]). In this case, the two dimensions $\Delta = d/2 \pm i\nu$ correspond to equally-leading and oscillatory modes as $\eta \to 0^-$, leaving to subtle questions about the best way to define the boundary conditions at $\mathcal{I}^+$. Presumably, the same analytic continuation in $\eta$ which led to the phase rule, will now lead to an enhancement of one mode over the other by an amount exponential in $\nu$.

Another interesting direction would be to see the implications of our discrete symmetry constraints and the phase rule for the recently proposed differential equation method for computing wavefunction coefficients in general flat FLRW cosmologies [113, 114]. It might also be worth seeing if there is a phase rule for the de Sitter S-matrix, constructed along the lines of [115, 116], or to the in-out formalism of [101]. It could also be that our phase rule provides useful constraints on flat space amplitudes, via the fact that going on the total "energy" pole $k_T \to 0$ (which is equivalent to going to the infinite past $\eta \to -\infty$), i.e. the "flat-space" limit, the Bunch-Davies vacuum looks like the flat-space vacuum.

Finally, it should be noted that in the derivation of our phase rule, we have implicitly assumed that the de Sitter vacuum is stable, and hence not subject to decay. In the string theory community, it is generally believed that de Sitter vacua in the string landscape will always be metastable. The probability that *no* vacuum decay event occurs anywhere, should fall off as an exponential of the total elapsed spacetime volume at late times. Although this correction looks badly divergent, it seems to have the right form to be renormalised by a $d$-dimensional cosmological constant counterterm in $\log Z_{\text{CFT}}$. However, this counterterm has a sign which is real, unlike the usual holographic counterterms in dS/CFT which are imaginary [44, 48, 71, 117]. As vacuum decay would lead to branches of the wavefunction disappearing into other possibilities, the assumption that vacuum decay does not occur is effectively a nonunitary postselection rule. Thus, we do not expect that our phase rule is necessarily valid for such processes. It would be interesting to study such vacuum transitions in more detail.

## Acknowledgements

It is a pleasure to thank Santiago Agüí Salcedo, Gonçalo Araujo Regado, Dionysios Anninos, Tarek Anous, Nima Arkani-Hamed, Paolo Benincasa, Calvin Chen, Chandramouli Chowdhury, Nick Cooper, Mang Hei Gordon Lee, Hofie Sigridar Hannesdottir, Daniel Harlow, Ted Jacobson, Rohit Kalloor, Rifath Khan, Juan Maldacena, Ciaran McCulloch, Prahar Mitra, Sebastian Mizera, Enrico Pajer, Guilherme Pimentel, Bharathkumar Radhakrishnan, Adel Rahman, Muthusamy Rajaguru, Alan Rios Fukelman, Vladimir Schaub, Vasudev Shyam, David Stefanyszyn, Leonard Susskind, Marija Tomašević, Xi Tong, Erik Verlinde, Tom Westerdijk, Edward Witten, and Yuhang Zhu for useful discussions. AT would like to especially emphasise the importance of the initial collaboration with Carlos Duaso Pueyo, who was the first to encourage him to continue pursuing this research program.

HG was supported jointly by the Science and Technology Facilities Council through a postgraduate studentship and the Cambridge Trust Vice Chancellor's Award. HG is supported by a Postdoctoral Fellowship at National Taiwan University funded by the Ministry of Education (MOE) NTU-112L4000-1. AT is supported by the Bell Burnell Graduate Scholarship Fund and the Cavendish (University of Cambridge). AT and AW are supported by the AFOSR grant FA9550-19-1-0260 "Tensor Networks and Holographic Spacetime". AW was also supported by the STFC grant ST/P000681/1 "Particles, Fields and Extended Objects", an Isaac Newton Trust Early Career grant, and NSF PHY-2309135 to the Kavli Institute for Theoretical Physics (KITP). AW was also partly supported by NSF grant PHY-2207584 while finishing this paper during his sabbatical at the IAS.

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
