# Peer review of "The Cosmological CPT Theorem"

_SciPost Physics_

## Round 2 · Referee Report · Anonymous (Referee 1) · 2025-10-26

Report

The paper examined the CPT theorem and related topics in a cosmological background, mostly focusing on the dS spacetime. The discussion highlights the interplay among the unitarity, the CPT, and a generalized version of dilatation, which I found interesting and clear. The authors also discussed the implications of CPT/CRT invariance on the wavefunction coefficients, which may be relevant to realistic cosmological tests of these fundamental principles. Overall, I think that the paper is worthy of publication in SciPost Physics after the authors address the following questions.

Requested changes

  1. In Eq. 3.53, both the operator $O(\eta)$ and the state $\Psi(\eta)$ depend on time $\eta$, which I find a bit confusing. Did the authors work with the Schrödinger picture or the Heisenberg picture here? Also, when saying $|\Psi\rangle=|\Omega\rangle$ below 3.53, did the authors mean that the universe was in the real physical vacuum at some fixed moment, or that the state $|\Psi\rangle=|\Omega\rangle$ is time independent? Please clarify.

  2. Most of the discussions in the paper assumed the dS background, except Sec. 6.3, where the authors considered a general spatially flat FLRW background. According to the authors, the analytical continuation in this case can be easily achieved by rotating spatial instead of temporal coordinates, which sounds pretty general. Thus, I wonder if the discussions in the rest of the paper can be generalized to FLRW spacetime as well.

  3. In Sec. 7, the authors derived and checked many phase formulas for wavefunction coefficients. I wonder whether these phase formulas could be tested using real cosmological observables. Can authors give some explicit examples in terms of observables instead of wavefunction coefficients? This will be helpful for a real test of CPT invariance in a cosmological setup.

Recommendation

Ask for major revision

  • validity: top
  • significance: high
  • originality: top
  • clarity: top
  • formatting: excellent
  • grammar: perfect

Author:  Ayngaran Thavanesan  on 2025-11-05  [id 5994]

(in reply to Report 1 on 2025-10-26)
Category:
remark
answer to question
correction

Please see the attached file for our response to the referee report, or the text from the LaTeX for our response to the referee report below.

\documentclass{article}
\usepackage{amsmath}
\usepackage[margin=1.5in]{geometry}
\usepackage{hyperref}

\begin{document}
First of all, we would like to thank the referee for their comments about the paper; they helped us identify some points where our discussion was lacking clarity and also noticed some further research directions that we agree are interesting. In the following, we quote the different parts of the report and answer them in turn.

\begin{itemize}
\item \emph{In Eq. 3.53, both the operator $O(\eta)$ and the state $\Psi(\eta)$ depend on time $\eta$, which I find a bit confusing. Did the authors work with the Schrödinger picture or the Heisenberg picture here?}

As we mention below this equation all our analysis was done in the interaction picture, and so both states and operators have time dependence. However, the author is correct that in (3.53) we are not yet in this picture, and so we have changed our notation to the Heisenberg picture where the states do not have any time dependence.
\item \emph{Thus, I wonder if the discussions in the rest of the paper can be generalised to FLRW spacetime as well.}

We agree that this is an interesting question to ask and two of the authors addressed this in a \href{https://arxiv.org/pdf/2510.21701}{recent paper}. Actually, in the case of the RR symmetry, we already in this paper make arguments that are valid for a range of flat FLRW spacetimes (as long as they start in a period of initial inflation with a Bunch-Davies vacuum). The main obstacle to generalising the other symmetries, is that it is not clear that we should impose some kind of CRT condition in generic spacetimes. In the case of de Sitter, the CRT symmetry is inherited from the global extension of the spacetime, which we don't have in other cases. Also, a typical cosmology will not have a scale-invariance symmetry $\eta \to \lambda\eta$, $y \to \lambda y$.

\item \emph{I wonder whether these phase formulas could be tested using real cosmological observables. Can authors give some explicit examples in terms of observables instead of wavefunction coefficients?}

Again, this question is indeed an important one to address, and this was precisely the angle taken in a \href{https://arxiv.org/pdf/2501.06383}{follow-up paper} by one of the authors, where it was shown that this symmetry argument imposes constraints on the parity-odd part of the correlators. We felt that these results were best separated firstly to appropriately assign credit for the work to the author and secondly as we viewed this paper as a look at the formal aspects, with the second paper there for anyone just interested in the phenomenology. We do, however, address that these results could be used in this way in the conclusions.

In general, a constraint on the \emph{phase} of a wavefunction coefficient is not usually easy to observe, because it requires looking at correlators involving time derivatives $\pi$ (e.g.~at least one of $\langle \phi \pi \rangle$ or $\langle \pi \pi \rangle$), which are subleading compared to correlators involving the field $\phi$ itself (see e.g.~\href{http://arxiv.org/abs/2110.01635}{this paper by Șengör and Skordis} for the case of the two-point correlator). It is only at special values of $\Delta$, such as the one addressed by the \href{https://arxiv.org/pdf/2501.06383}{recent paper}, that there is a no-go result on certain correlators becoming large. For generic values of $\Delta$, it is more feasible to view our result as a way to constrain the unobservable part of the wavefunction, based on the observable part.

The Discussion has be edited to clarify this point. The new text begins with the phrase: "In general, a constraint on the phase".
\end{itemize}
In addition to the points raised by the referee we also made the following changes that we felt further improved the presentation of our results.
\begin{itemize}
\item In the introduction to Section 6, we have noted with appropriate references that Charlotte Sleight and collaborators have worked with similar, yet distinct, analytic continuations when relating EAdS correlators to dS correlators.
\item In Section 6, we have corrected some confusing notation where the argument of the wavefunction was the vector components of a single position or momentum vector, so that now it is more explicit that we are rotating all of the vectors upon which the function depends. We have left the notation in Section 7 unchanged for compactness, with this notation defined below (7.19).
\item In Section 6, we correct footnote 47 as we noted that our results do seem to hold for the case of principal series (heavy) fields, if one imposes self-adjoint boundary conditions.
\item In Section 7, we correct footnote 71 as we noted that not all $c=26$ two-dimensional CFTs can be duals to pure gravity in $\text{dS}_3$
.
\item Other minor corrections including added references and correcting typos.
\end{itemize}
These changes will be appearing in v3 on the arXiv within the next few days.

\end{document}

Attachment:

SciPost_ResponsetoRefereeReport.pdf

---

## Round 2 · Referee Report · Anonymous (Referee 2) · 2025-12-2

Strengths

  1. The paper describes a novel way of understanding unitarity for cosmological correlators.
  2. The results seems well grounded and in particular the thoroughly study the case of scalar fields.
  3. Their results are valid a all orders and they write concise formula that determines the phase of late-time wavefunction coefficients—even in the presence of the leading UV/IR divergences—giving a useful input for cosmological bootstrap approaches.

Weaknesses

See below

Report

The paper addresses some important issues of QFT in cosmological contexts which are relevant to current discussions In partocular they investigate how discrete symmetries, unitarity, and analyticity constrain quantum field theories in both flat and cosmological settings. Rather than treating CPT as an isolated theorem, the authors recast it within a minimal algebraic framework that cleanly separates the roles of unitarity, spatial transformations, and charge–parity–time reversal. This reinterpretation becomes useful when applied in the context of de Sitter spacetimes, where the absence of global time reversal or a past boundary makes CPT-type statements nontrivial.

Within this framework the authors identify simple and robust condition (Reflection Reality, BD analyticity, and a controlled analytic continuation) that together impose a universal constraint on the phase of late-time wavefunction coefficients. This leads to a compact, nonperturbative relation that applies to all UV/IR-finite contributions and the leading divergent parts of cosmological correlators. The result provides a structural boundary condition for de Sitter wavefunction coefficients that is highly relevant to cosmological bootstrap methods and to understanding the analytic structure of correlators in expanding universes.

I do however have some questions that I'd like the authors address before recommending the paper for publication,

  1. In the discussion of “accidental unitarity,” could the authors clarify the assumptions under which Reflection Reality effectively determines the unitary subspace? In particular, does this require holding the spectrum and number of propagating degrees of freedom fixed? 2 Does Reflection Reality, perhaps together with locality, imply the full set of Schwinger–Keldysh unitarity relations (see eg. 1805.09331), or is it meant to capture only part of the this structure as ? 3 In realistic inflationary models the background spontaneously breaks time translations, so the late-time mode functions deviate from pure $ \eta^{\Delta-d}$ scaling and acquire slow-roll and logarithmic corrections, hence the map $(\eta,\mathbf x)\to(-\eta,-\mathbf x)$ is no longer a symmetry of the background. In this situation, do you expect any controlled remnant of your cosmological CPT / phase-fixing relation to survive (e.g. as an expansion in slow-roll parameters), or should your result be understood as strictly confined to exact de Sitter and not directly applicable to generic inflationary EFTs?

4 What conditions guarantee that the analytic continuation $(\eta,k)\to(e^{i\pi}\eta,e^{i\pi}k)$ is valid?, and is there a natural extension of the phase relation to principal-series fields or to theories with more complicated singularity structure?

5 In the loop examples treated with dimensional regularisation, the coefficient of the $1/\epsilon$ divergence clearly satisfies your phase relation, and you note that the associated finite term acquires an imaginary shift relative to this divergent piece. Should one expect the full renormalised finite part of a loop diagram to obey any generalised version of your phase constraint, or is the result intended to apply strictly to the divergent coefficient (and its induced shift) while the remaining finite contributions are left unconstrained?

Recommendation

Ask for minor revision

  • validity: high
  • significance: high
  • originality: high
  • clarity: good
  • formatting: good
  • grammar: perfect

Author:  Ayngaran Thavanesan  on 2025-12-19  [id 6165]

(in reply to Report 2 on 2025-12-02)
Category:
remark
answer to question
validation or rederivation

Please see the attached file for our response to the referee report, or the text from the LaTeX for our response to the referee report below.

\documentclass{article}
\usepackage{xcolor}
\usepackage{amsmath}
\usepackage{amsfonts}
\usepackage[margin=1.5in]{geometry}
\usepackage{hyperref}

\begin{document}
First of all, we would like to thank the referee for their thoughtful comments. They have helped us identify places where the exposition can be clarified and have also highlighted several interesting directions for future work. In the following, we quote each point of the report in turn and respond to them.

\begin{itemize}

\item \emph{In the discussion of “accidental unitarity,” could the authors clarify the assumptions under which Reflection Reality effectively determines the unitary subspace?}

In general, Reflection Reality is a necessary but not sufficient condition for Unitarity. This can be seen from the fact that RR implies the existence of a preserved norm on the state space, where the norm is hermitian but might be indefinite.

But, we claim that a large generic class of RR-satisfying theories are also unitary, without the need to fine-tune any parameters in the Lagrangian. In particular, if you take a unitary theory and perturb the Lagrangian by an RR-satisfying term, then the resulting theory will still be unitary.

\emph{In particular, does this require holding the spectrum and number of propagating degrees of freedom fixed?}

Yes, this argument requires that the number of fundamental degrees of freedom propagating in the Lagrangian does not change. The example in 5.4 of the Proca Lagrangian was given to illustrate this point. The Proca Lagrangian (Eq. 5.4) is unitary when $m^2 > 0$, but it has a negative-norm propagating mode in the tachyonic case $m^2 < 0$. These domains are separated by the $m^2 = 0$ Maxwell case, where there is a gauge-symmetry and hence there is 1 fewer propagating degree of freedom. Perturbations which change the sign of $m^2$ are therefore not allowed perturbations for purposes of the above argument. (More generally, continuity requires that one cannot obtain a negative-norm state without passing through a case with a zero-norm state, implying the existence of an additional gauge symmetry.)

It is not clear to us what the referee means by the ``spectrum". If this refers to the spectrum of the Hamiltonian $H$ (for simplicity, we can consider a non-cosmological context, where $H$ is independent of time), then of course perturbations to the Lagrangian $L$ can change the energy eigenvalues, without thereby modifying the unitary condition $H^{\dagger}=H$. This is precisely what happens for a typical RR-satisfying perturbation to a unitary $H$. (Conversely, if we decline to impose RR, a generic complex perturbation to $H$ will make the energy eigenvalues become complex, which is a symptom of the fact that Unitarity is now violated.)

\item \emph{Does Reflection Reality, perhaps together with locality, imply the full set of Schwinger–Keldysh unitarity relations (see eg. 1805.09331), or is it meant to capture only part of this structure?}

RR by itself captures only part of the full Schwinger–Keldysh structure. As discussed around (2.15), RR requires only the existence of a preserved norm and therefore enforces the analogue of the SK constraints associated with Hermiticity of the Hamiltonian. In the language of 1805.09331, this is equivalent to enforcing the counterparts of eqs. (2.20) and (2.23), but not the analogue of (2.25), which encodes the largest-time equation and cutting rules. Thus RR should be viewed as a weaker constraint than full SK unitarity.

Of course, in any field theory where we happen to accidentally obtain full Unitarity, all other implications of Unitarity will also hold.

\item \emph{In realistic inflationary models the background spontaneously breaks time translations, so the late-time mode functions deviate from pure $\eta^{\Delta-d}$ scaling and acquire slow-roll and logarithmic corrections. In this situation, do you expect any controlled remnant of your cosmological CPT / phase-fixing relation to survive, or should your result be understood as strictly confined to exact de Sitter and not directly applicable to generic inflationary EFTs?}

Our derivation applies strictly to the $\epsilon=0$ de Sitter limit. However, as explained in the discussion below (3.50), and emphasised in the following discussions in the rest of the paper, the relation is robust for a large class of boost-breaking Lagrangians, including those appearing in the Effective Field Theory of Inflation 0709.0293. In the Discussion we noted that slow-roll corrections induce only small deviations (of order a few per cent) to the late-time scaling behaviour, and hence we expect that a controlled remnant of our relation should survive as an expansion in slow-roll parameters. A detailed analysis of these corrections would be an interesting direction for future work.

\item \emph{What conditions guarantee that the analytic continuation $(\eta,k)\to(e^{i\pi}\eta,e^{i\pi}k)$ is valid, and is there a natural extension of the phase relation to principal-series fields or to theories with more complicated singularity structure?}

The analytic continuation is valid whenever the mode functions admit an analytic branch through the negative real axis and no singularities obstruct the contour deformation. This is satisfied for both complementary- and principal-series fields in the bulk. Our restriction to the complementary series was not due to a limitation of the continuation itself, but because for principal-series fields the boundary behaviour is not characterised by a definite scaling weight. As a result, the phase relation for the wavefunction coefficients is not directly meaningful: in holographic language, principal-series bulk fields source operators with complex conformal dimension, and CRT relates such operators to their complex conjugates. Thus it maps $\psi_n \equiv \langle \mathcal{O}_1\ldots\mathcal{O}_n\rangle$ to $\tilde{\psi}_n \equiv \langle \mathcal{O}^{\dagger}_1\ldots\mathcal{O}^{\dagger}_n\rangle$ rather than to $\psi_n^{\ast}$.

\item \emph{In the loop examples treated with dimensional regularisation, the coefficient of the $1/\epsilon$ divergence satisfies your phase relation, and the associated finite term acquires an imaginary shift relative to this divergent piece. Should one expect the full renormalised finite part of a loop diagram to obey any generalised version of your phase constraint, or is the result intended to apply strictly to the divergent coefficient?}

The phase relation applies directly to the regulated value at fractional $d$, and also to the coefficient of the log divergence. The finite part in the $d \to \text{integer}$ limit is generically renormalisation-scheme dependent, and counterterms may shift the (renormalised) mass or scaling dimension of the field. In such cases, the naïve phase relation need not hold. However, if one works with fields whose mass is protected, or for which the renormalised mass is known, then one can formulate an analogous phase relation for the renormalised correlators. A full investigation of this generalisation would be interesting but goes beyond the scope of the present work but was explored to some extent in 2501.06383.

\end{itemize}

\end{document}

Attachment:

RefereeReport2_Response.pdf

---

## Editorial Decision

resubmitted